# Large-area integration of two-dimensional materials and their heterostructures by wafer bonding

Arne Quellmalz [1✉], Xiaojing Wang[1], Simon Sawallich [2,3], Burkay Uzlu[3,4], Martin Otto [4], Stefan Wagner [4], Zhenxing Wang [4], Maximilian Prechtl [5], Oliver Hartwig [5], Siwei Luo[5], Georg S. Duesberg [5], Max C. Lemme [3,4], Kristinn B. Gylfason[1], Niclas Roxhed [1], Göran Stemme[1] & Frank Niklaus [1✉]

Integrating two-dimensional (2D) materials into semiconductor manufacturing lines is essential to exploit their material properties in a wide range of application areas. However, current approaches are not compatible with high-volume manufacturing on wafer level. Here, we report a generic methodology for large-area integration of 2D materials by adhesive wafer bonding. Our approach avoids manual handling and uses equipment, processes, and materials that are readily available in large-scale semiconductor manufacturing lines. We demonstrate the transfer of CVD graphene from copper foils (100-mm diameter) and molybdenum disulfide ($MoS_2$) from $SiO_2$/Si chips (centimeter-sized) to silicon wafers (100-mm diameter). Furthermore, we stack graphene with CVD hexagonal boron nitride and $MoS_2$ layers to heterostructures, and fabricate encapsulated field-effect graphene devices, with high carrier mobilities of up to $4520 \, cm^2V^{-1}s^{-1}$. Thus, our approach is suited for backend of the line integration of 2D materials on top of integrated circuits, with potential to accelerate progress in electronics, photonics, and sensing.

[1] Division of Micro and Nanosystems, School of Electrical Engineering and Computer Science, KTH Royal Institute of Technology, Stockholm, Sweden. [2] Protemics GmbH, Aachen, Germany. [3] Chair of Electronic Devices, Faculty of Electrical Engineering and Information Technology, RWTH Aachen University, Aachen, Germany. [4] AMO GmbH, Advanced Microelectronic Center Aachen (AMICA), Aachen, Germany. [5] Institute of Physics, EIT 2, Faculty of Electrical Engineering and Information Technology, Universität der Bundeswehr München, Neubiberg, Germany. ✉email: arne.quellmalz@eecs.kth.se; frank.niklaus@eecs.kth.se

The astonishing properties of two-dimensional (2D) materials have aroused tremendous interest in the semiconductor industry. These new types of atomically thin materials promise to continue the trend of shrinking transistors and better sensors. For graphene, an atomically thin layer of carbon atoms, the ultrahigh charge carrier mobility, and its strong light absorption enabled the demonstration of high-frequency analog electronics[1,2], flexible electronics[3], high-sensitivity Hall sensors[4,5], and high-speed photodetectors for telecommunication and imaging technologies[6–8]. Transition metal dichalcogenides are often semiconducting and promise advancements for high- and low-power transistors[9], infrared photodetectors[10], and emerging device concepts such as memristors[11–13] and single-photon emitters for optical quantum communication[14–16]. The scaling limit of transistors is set by short-channel effects that arise from intrinsic semiconductor properties. These effects degrade the off-state leakage current and are the limiting factor for silicon transistors with sub-10 nm gate length. On this scale, molybdenum disulfide ($MoS_2$) transistors are expected to achieve more than two orders of magnitude lower leakage current than silicon transistors[17]. Furthermore, the carrier mobility in $MoS_2$ transistors degrades less with decreasing channel thickness than in silicon transistors[18,19]. Hence, 2D semiconductors may be the ultimate channel material for ultra-scaled MOSFETs[20]. Vertical stacking of 2D materials forms van der Waals heterostructures, a novel class of materials with new material properties that result from synergetic effects between the stacked layers at their interfaces[21–26]. Integrating 2D materials into conventional semiconductor fabrication lines is critically important to exploit their material properties in commercial devices while benefitting from established silicon-based infrastructure with low-cost and high-volume device manufacturing on large[20] substrates[20,27]. Yet, there are several bottlenecks that have prevented this last decisive step so far. The synthesis of 2D materials by chemical vapor deposition (CVD) is scalable to large areas, and the best contemporary processes provide close to intrinsic material quality[28–31]. However, the high process temperatures required to obtain high material quality rules out direct growth on preprocessed silicon electronics substrates, and hence mandates a material transfer from dedicated growth substrates to the device substrate. Commonly used wet transfer approaches rely on an intermediate polymeric carrier, typically poly(methyl methacrylate) (PMMA)[32–34] or polycarbonate[35,36], which mechanically supports the 2D material during its removal from the growth substrate and the transfer to the target substrate by scooping from the surface of a liquid. This process may be implemented at various stages of the device fabrication and allows the placement of the 2D material directly on a target substrate of choice, including complementary metal oxide semiconductor electronic wafers. For graphene, wet transfers degrade the material properties by causing defects, wrinkles, and strain in the transferred layer[37]. In addition, residuals of the polymeric carrier layer remain on the surface of the 2D material and degrade its electronic transport properties[38,39]. Attempts to minimize these adverse effects of wet transfer include bubble delamination[40], advanced cleaning procedures[41], and replacing poly(methyl methacrylate) with paraffin as carrier polymer[42]. Nevertheless, they do not eliminate the noted issues entirely. Dry transfer approaches using rollers, laminators, and hot presses enable the reuse of metallic growth substrates and avoid submersion of the target substrate in liquids, which is beneficial, for example, for the integration of 2D materials with suspended devices[33,43–50]. However, dry-transferred layers often suffer from microcracks, wrinkles, reduction in charge carrier mobility, and contamination by residuals from the sacrificial carrier layers. On the micrometer scale, cleaning techniques can mechanically manipulate the

location of encapsulated contaminants and restore intrinsic material properties[51]. Delamination of single-crystal CVD graphene from copper foil by strong van der Waals forces with a stamp of exfoliated hexagonal boron nitride (hBN) yields almost intrinsic graphene properties, but is limited in size to areas of a few hundred micrometers[30]. Most importantly, neither of the reported transfer methods is compatible with industry routines for large-scale manufacturing while still preserving the high quality of the 2D material as on the growth substrate, which is required for many applications[52]. Here, we present a versatile approach for the transfer and stacking of 2D materials to heterostructures by adhesive bonding with bisbenzocyclobutene (BCB) and commercial wafer bonding equipment. BCB is a hydroxyl-free dielectric material that was developed for the semiconductor industry and is being used as an interlayer dielectric, for heterogeneous system integration, and as an organic gate dielectric layer since it has low fixed-charge and trap densities[53–57]. Our approach avoids manual handling of the 2D material and does not require sacrificial carrier layers that may contaminate the surface of the 2D material. The proposed method only utilizes equipment, processes and materials that are readily available in large-scale semiconductor production lines, which are crucial advantages for the integration into the semiconductor ecosystem[58,59]. We demonstrated the transfer of monolayer graphene from copper foils to 100-mm-diameter silicon wafers, the transfer and stacking of multilayer hBN, and monolayer graphene to form graphene/hBN heterostructures, and the stacking of two graphene layers to form double-layer graphene. We also fabricated field-effect graphene devices to demonstrate the utility of our methodology for wafer-level device manufacturing with conventional semiconductor processes. Furthermore, we demonstrated the transfer of multilayer $MoS_2$, as a representative for transition metal dichalcogenides, from a centimeter-sized silicon chip to a 100-mm-diameter silicon wafer. All transferred layers and heterostructures were of high quality, that is, they featured uniform coverage and little strain in the transferred 2D materials. These findings indicate that our proposed integration approach preserves similar mechanical properties of the 2D materials as present on the growth substrate while minimizing degradation of the transferred layers through the introduction of wrinkles or excessive strain.

## Results

**Method for transfer of 2D materials and heterostructures**. Our method for transferring 2D materials from their growth substrate to a target wafer comprises four consecutive steps (Fig. 1): first, the target wafer is spin-coated with an adhesive layer of thermosetting BCB. A softbake removes solvents and solidifies the adhesive layer (Fig. 1a (1)). Next, the 2D material on its growth substrate is placed on top of the target wafer such that the 2D material is facing the adhesive layer (Fig. 1a (2)). The stack is then loaded in a commercial wafer bonder (Fig. 1a (3)). Inside the tool, heating temporarily decreases the viscosity of the adhesive layer while the bond chuck applies a uniform force to the wafer stack. Thus, the adhesive layer molds against the 2D material and forms a stable bond to the target wafer while replicating the surface topography of the growth substrate without exerting excessive pressure on the 2D material. This characteristic is beneficial, since it minimizes potential damage, wrinkles, or strain in the transferred 2D material. After wafer bonding, the growth substrate is removed (Fig. 1a (4)) by either etching, delamination, or permeation of liquids into the interface between the 2D material and the growth substrate, which leaves the 2D material transferred on the target wafer (Fig. 1a (I)). Since BCB is a thermosetting polymer, heating partially cross-links the polymer chains in the adhesive layer, which form a network with high

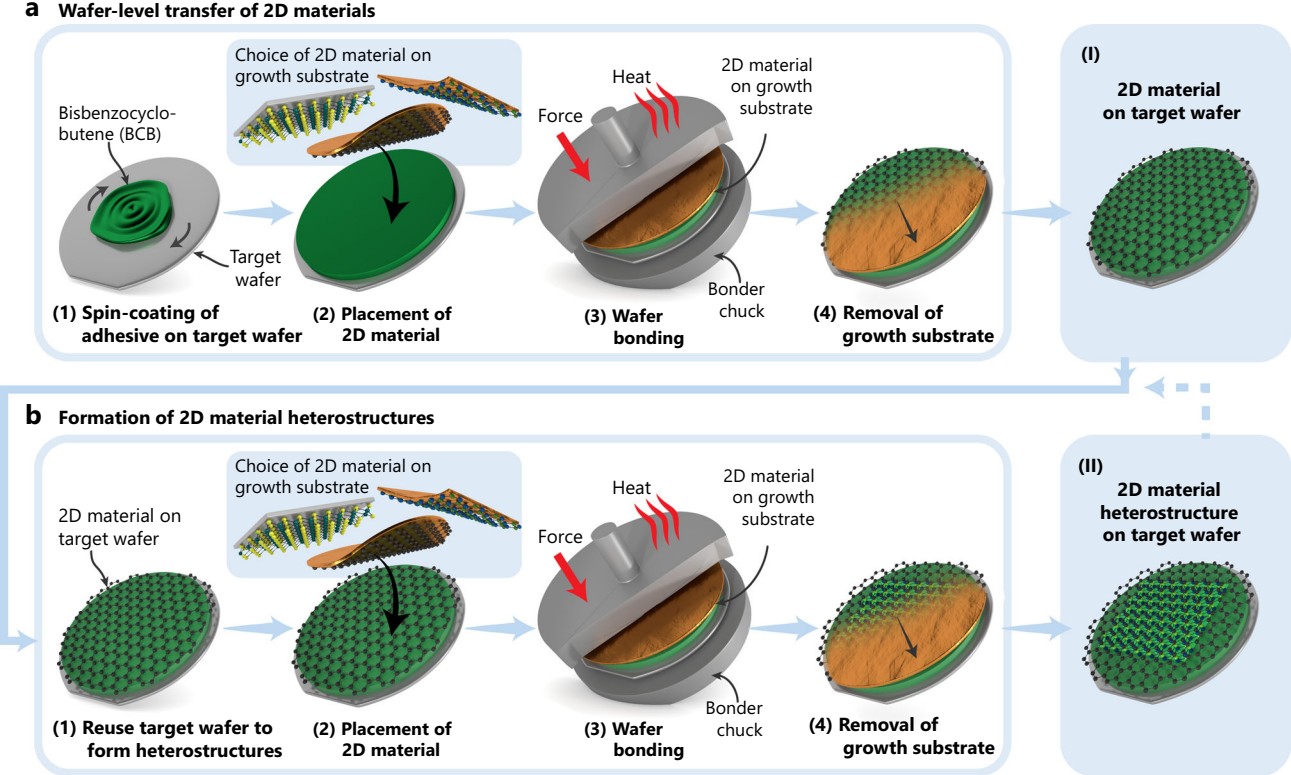

**Fig. 1 Schematics illustration of the methodology for wafer-level transfer of two-dimensional (2D) materials and formation of 2D material heterostructures. a** Wafer-level transfer of materials. (1) Spin coating and soft baking of thermosetting bisbenzocyclobutene (BCB) as an adhesive layer on the target wafer. (2) Placement of the 2D material on its growth substrate on top of the target wafer, with the 2D material facing the target wafer. (3) Adhesive wafer bonding by applying heat and force to the wafer stack using a commercial semiconductor wafer bonding tool, thereby forming a stable bond between the 2D material on its growth substrate and the target wafer. (4) Removal of the growth substrate. (I) Transferred 2D material on the target wafer. **b** Formation of 2D material heterostructures. (1) Reusing the target wafer from step (I) without additional treatment. (2) Placement of a second 2D material on its growth substrate on top of the target wafer, with the 2D material facing the previously transferred 2D material. (3) Wafer bonding as in a) step (3). (4) Removing the growth substrate of the second 2D material. (II) Transferred 2D material heterostructure on the target wafer.

chemical stability. If the degree of cross-linking is kept low, by appropriate choice of bonding temperature and time, the BCB adhesive layer allows repeated molding. Hence, reusing the same target wafer for another transfer assembles heterostructures by vertical stacking of 2D materials—without coating any additional adhesive, simply by reusing the existing adhesive layer below the previously transferred 2D material (Fig. 1b (1–4) and (II)). Thus, van der Waals heterostructures form by repeating the three steps of material placement (Fig. 1b (2)), wafer bonding (Fig. 1b (3)), and removal of the growth substrate (Fig. 1b (4)). We note that in this process neither the interface between the first and second 2D material nor the surface of the second 2D material is exposed to any polymer carrier or adhesive that may potentially degrade the 2D material properties by contamination. Partially cross-linked BCB is chemically stable and sustains conventional wafer processing. This feature enables not only structuring of the 2D material to devices after the transfer (using the wafers (I) and (II) in Fig. 1a, b, respectively), but also the integration of metal contacts on top of the BCB layer before transfer (Supplementary Fig. 1). These contacts could be used to interface integrated circuits (ICs) that may be embedded on the target wafer. Moreover, membranes of 2D materials can be suspended over cavities, which were etched into the adhesive layer before transfer (Supplementary Fig. 2). In both cases, the 2D material in the contact and membrane area is not in contact with the adhesive at any time (see Supplementary Notes 1 and 2 for demonstration).

**Wafer-level transfer of graphene/hBN heterostructures.** To demonstrate the viability of our methodology for wafer-level transfer of 2D materials and formation of large-area van der Waals heterostructures, we integrated graphene/hBN heterostructures (see "Methods" for details of the process and materials). We characterized the transferred layers using van der Pauw (vdP) devices, noncontact terahertz (THz) near-field spectroscopy, and Raman spectroscopy (Fig. 2). In short, consecutive transfers of multilayer hBN from copper foil and monolayer graphene from copper foil, both synthesized by CVD, formed a graphene/hBN heterostructure on a 100-mm-diameter silicon wafer. First, we attached the hBN on copper foil (2.5 cm × 2.5 cm) to the adhesive on the spin-coated target wafer by wafer bonding. After etching away the copper in $FeCl_3$ solution and rinsing in deionized water, we placed the monolayer graphene on a copper foil (100 mm diameter) on top of the target wafer and performed a second bonding process in a vacuum atmosphere. Finally, etching in $FeCl_3$ uncovered the transferred layers. Due to the difference in the size of the growth substrates, the graphene/hBN heterostructure in the center of the wafer is surrounded by a monolayer of graphene on BCB. Note that in the heterostructure, neither the graphene nor the top surface of the hBN was in contact with polymers or adhesives at any time. Moreover, the second transfer was performed in a vacuum and at elevated temperature, which reduces the amount of water and gas molecules that may get trapped between the layers. Residuals of copper

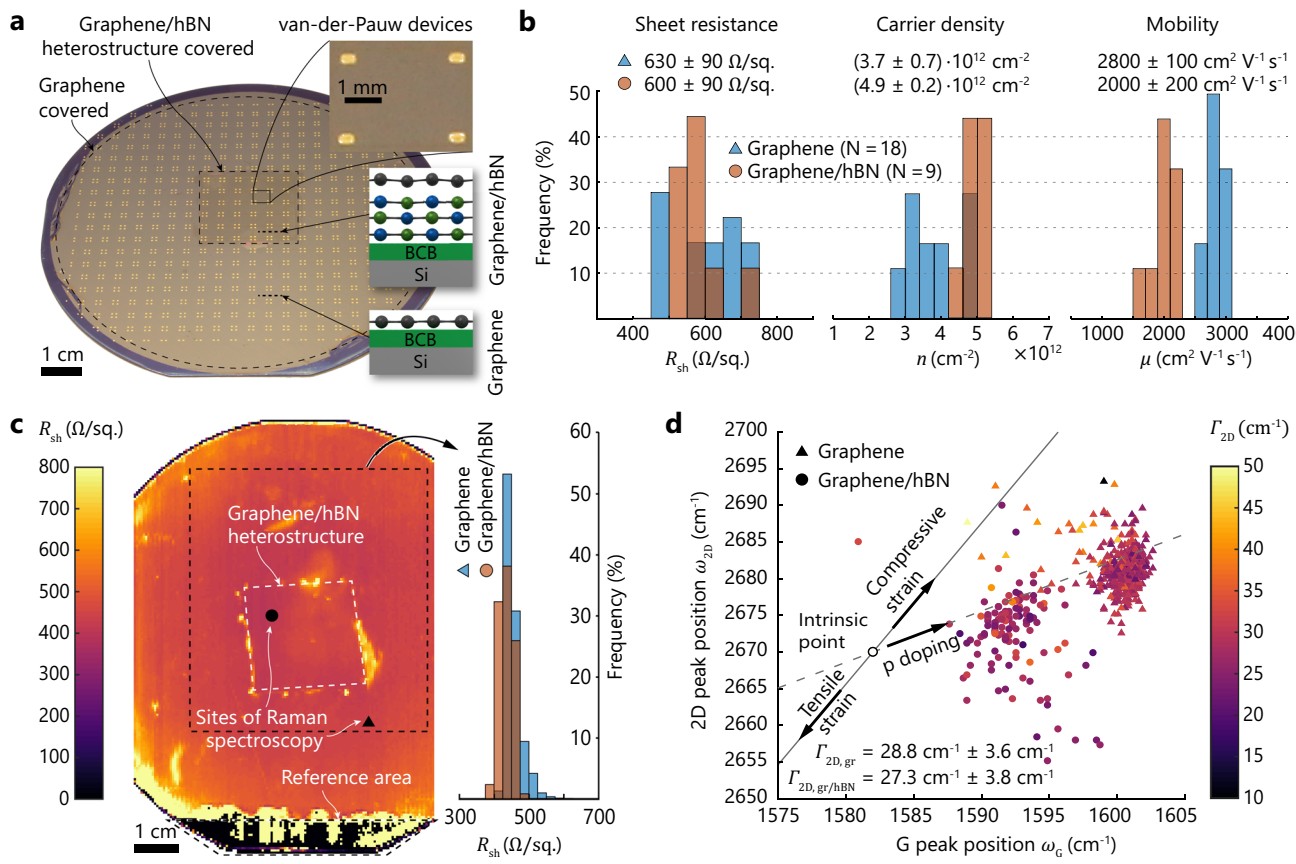

**Fig. 2 Characterization of transferred CVD graphene and graphene/hexagonal boron nitride (hBN) heterostructures. a** Photograph of a 100-mm-diameter silicon wafer with van der Pauw (vdP) devices. The graphene sheet covers the entire wafer and lies on top of multilayer CVD hBN in the marked region. **b** Extracted sheet resistance ($R_{sh}$), carrier density ($n$), and charge carrier mobility ($\mu$) from Hall measurements of vdP devices at room temperature (number of devices: 18 graphene (blue); 9 graphene/hBN (red)). **c** Spatially resolved map of $R_{sh}$ extracted from noncontact terahertz near-field spectroscopy, indicating a uniform coverage of graphene on the entire high resistive silicon wafer (average within the marked region: $450 \pm 50 \, \Omega \, \text{sq}^{-1}$). **d** Correlation map of the Raman G and 2D peak position ($\omega_G$ and $\omega_{2D}$, respectively) of the transferred graphene (triangles) and the graphene/hBN heterostructure (circles). The colormap represents the extracted full-width at half-maximum of the 2D peak ($\Gamma_{2D}$) of the transferred graphene ($\Gamma_{2D,gr}$) and the graphene/hBN heterostructure ($\Gamma_{2D,gr/hBN}$). The open circle indicates the peak positions of intrinsic graphene[74]. The total number of spectra for graphene on bisbenzocyclobutene (BCB) and graphene on hBN is 430 and 124, respectively. Note, **a**, **b** are from the same sample, whereas **c**, **d** are performed on a second sample.

and the copper etchant are still potential contaminants at the interlayer interface, which potentially can affect the properties of the heterostructure[60]. Next, we integrated Ti/Au electrodes and patterned the 2D material heterostructure to vdP devices, which allow electrical material characterization on the target wafer (Fig. 2a). Four-point probe measurements of a total of 27 vdP devices yielded the graphene sheet resistance ($R_{sh}$), carrier density ($n$), and mobility of majority charge carrier ($\mu$) of the transferred graphene layer, both for graphene resting on BCB and for graphene resting on hBN (Fig. 2b) (see "Methods"). In both cases, all measured devices were functioning and we found a similar $R_{sh}$ ($630 \pm 90$ and $600 \pm 90 \, \Omega \, \text{sq}^{-1}$, respectively), whereas the $n$ and $\mu$ showed differences for graphene resting on BCB and on hBN, respectively. The carrier density in the graphene on BCB was lower than for graphene on hBN (($3.7 \pm 0.7) \times 10^{12}$ and $(4.9 \pm 0.2) \times 10^{12} \, \text{cm}^{-2}$), while the mobility was higher for graphene on BCB than for the graphene on hBN ($2800 \pm 100$ and $2000 \pm 200 \, \text{cm}^2 \, \text{V}^{-1} \, \text{s}^{-1}$). For the transferred graphene monolayer resting on BCB, the extracted mobilities are in the higher range of previously reported values for polycrystalline CVD graphene transferred on large areas from polycrystalline copper substrates by various methods, including wet transfer with polymeric carrier

layers[42,61], face-to-face transfer by bubble formation[62] and lamination[50]. The mobility in our graphene/hBN heterostructure is similar to values reported by Pandey et al.[63] and Shautsova et al.[64], who encapsulated graphene in CVD hBN using wet and dry transfers. In accordance with these works, our results indicate that current large-area CVD hBN substrates typically do not increase the mobility in polycrystalline graphene, thus contrasting several studies on single crystals of graphene and hBN[29,30,65,66]. We emphasize that for a meaningful comparison, the initial graphene quality before the transfer is of crucial importance. Our measurements suggest slightly lower mobility in the graphene/hBN heterostructure than for graphene on BCB. Since we performed the measurements on the same wafer containing graphene from the same growth substrate, which was exposed to the same fabrication processes, we conclude that the substrate below the graphene (hBN and BCB, respectively) is mainly responsible for the discrepancy in material characteristics. For our materials, the microscopic surface roughness of hBN on Cu foil is ~50% higher than for graphene on Cu foil (root mean square (RMS): 264 and 176 nm, respectively, see Supplementary Fig. 3). This difference might cause local strain variations in the covering graphene layer, which reduces charge carrier mobility[67]. In

addition, large topography of the transferred materials may potentially cause problems for very high-resolution lithography. Using growth substrates with lower surface topography may mitigate this issue[68–70].

We used terahertz time-domain spectroscopy (THz-TDS) to further characterize the electrical properties and homogeneity of the wafer-level transferred graphene and large-area graphene/hBN heterostructure. The THz-TDS measurements are recorded in transmission mode using a photoconductive near-field detector[71,72] (see "Methods"), which records the amplitude and phase information of a THz signal traveling through the sample. Scanning across the sample surface yields spatially resolved images of the graphene sheet resistance on wafer scale. For our measurements, we set the spatial resolution to 500 μm by adjusting the distance of the near-field detector to the sample and the pitch between acquisition sites. The spatial resolution is significantly larger than the intrinsic length scale that is probed by the THz-TDS. This transport length is estimated by the distance $l_D$, which a charge carrier transverses during one cycle of the alternating THz field[73]. For our sample and setup, we estimate $l_D \approx 60 - 100$ nm (see Supplementary information for calculation). The map of the graphene sheet resistance on a 100-mm-diameter wafer after transfer and formation of the graphene/hBN heterostructure indicates a uniform coverage and successful transfer on the entire wafer (Fig. 2c). The sheet resistance in the marked region averages to $450 \pm 50\,\Omega\,sq^{-1}$, and is similar for both underlying materials ($450 \pm 20$ and $440 \pm 20\,\Omega\,sq^{-1}$ for BCB and hBN, respectively), which is in line with the measurements of the vdP devices shown in Fig. 2b. These values are similar to the values reported for polycrystalline graphene that was transferred from polycrystalline copper substrates using various methods[42,50,63]. Rahimi et al.[61] reported a higher mean value of $2600\,\Omega\,sq^{-1}$, which originated from lower residual doping. Note that the THz near-field measurement in Fig. 2c was performed on a different sample with high-resistivity Si than the vdP measurements in Fig. 2a, b to avoid absorption of the THz signal by the substrate. At the lateral transition between the graphene/hBN heterostructure and the monolayer graphene (i.e., the outer edges of the heterostructure), the graphene sheet resistance is increased and less uniform. This variation was most likely caused by surface inhomogeneities in the BCB layer that resulted from a partly detached edge region of the hBN on copper foil in the first transfer. This effect is known to occur in wafer bonding of compliant substrates where the contact area of the wafer chuck is smaller than the substrate. Precise adjustment of the contact area between the wafer bonder and the donor substrate to the size of the copper foil should improve the lateral interface between regions with different layer stacks.

We performed Raman measurements to extract detailed information about the local graphene quality, including the relationship between doping and nanometer-scale strain variations, by analyzing the positions of the Raman G ($\omega_G$) and 2D ($\omega_{2D}$) peaks and the full-width at half-maximum of the 2D peak ($\Gamma_{2D}$)[74–76]. In particular, stress variations are of interest since they limit the charge carrier mobility in high-quality graphene, a crucial parameter for high-performance graphene devices[67]. In Fig. 2d, we plot the correlation map of $\omega_G$ and $\omega_{2D}$ for the acquired Raman spectra of the transferred graphene placed on BCB (triangles) and on hBN (filled circles), respectively. The location of the acquisition sites on the wafer is marked in Fig. 2c. The single open circle represents the G and 2D peak positions of intrinsic graphene, which is neither doped nor strained ($\omega_G = 1582$ cm$^{-1}$, $\omega_{2D} = 2670$ cm$^{-1}$)[74]. Hole doping of graphene shifts $\omega_G$ and $\omega_{2D}$ with a relative slope of 0.7 (dashed line), while strain results in a relative shift with a slope of 2.2 (solid line)[75]. For both substrate

materials on our sample, the peak locations indicate slightly tensile strain and predominant hole doping. For graphene on BCB, the charge carrier doping is more pronounced than for graphene on hBN, while the peak locations are less scattered and more confined within a smaller area. Our observation of hole doping and reduced doping level of the graphene when placed on hBN compared to placement on polymeric substrate materials agrees with literature[77]. However, multilayer hBN grown by CVD does not have the same strong effect as exfoliated hBN, which restores almost intrinsic conditions[77]. The color scale of the data points in Fig. 2d represents $\Gamma_{2D}$ and averages to $28.8 \pm 3.6$ cm$^{-1}$ for graphene on BCB and $27.3 \pm 3.8$ cm$^{-1}$ for graphene on hBN. Both values indicate the decent quality of the graphene with little strain variations within the spot of the Raman excitation laser.

**Stacking of graphene layers by repeated transfer**. The recent discovery of specific twist angles between two graphene layers that give rise to astonishing properties such as unconventional superconductivity, correlated insulation, and magnetism[24–26] has spurred great interest in new methods for well-controlled fabrication of double-layer graphene. Furthermore, the stacking of two or multiple layers of 2D materials is of interest for controlling the stiffness of nanoelectromechanical devices[78,79].

We performed two consecutive transfers of monolayer (1-layer) graphene from their copper growth substrates to a 100-mm-diameter silicon wafer to demonstrate that our transfer methodology is suitable for high-yield formation of double-layer (2-layer) graphene on large areas (we avoid the term bilayer graphene to distinguish these films from Bernal stacked materials, see "Methods" for process details). First, we transferred CVD graphene from a quarter of a 100-mm copper foil to the center of the target wafer with BCB. For the second transfer, we used a full 100-mm copper foil with CVD graphene. To provide a reference for THz spectroscopy, we removed the graphene on parts of the growth substrate before transfer (left side of the wafer in Fig. 3a). The spatially resolved map of the sheet resistance ($R_{sh}$) in Fig. 3c reveals a uniform coverage of graphene on the wafer without major defects. Decreased $R_{sh}$ in the center clearly distinguishes 1- and 2-layer graphene on the target wafer and confirms the presence of double-layer graphene. We attribute the difference in the 1-layer graphene $R_{sh}$ between these experiments and the dataset presented in Fig. 2 to a dissimilarity in graphene quality before transfer. The graphene sheet in Fig. 3a that forms the 1-layer region on the target wafer was from a different batch than the graphene sheet used in Fig. 2. To characterize the 1- and 2-layer graphene on the silicon wafer, we extracted the position of the G and 2D peaks from Raman measurements and plotted the correlation map of $\omega_G$ and $\omega_{2D}$ (Fig. 3b). In both regions, the peak positions were clustered with little spread. The measurements indicate hole doping in the graphene, which is consistent with our measurements on graphene/hBN heterostructures in Fig. 2d. We found that the 1-layer graphene (triangles) experienced tensile strain, while the 2-layer graphene (circles) was strained compressively. However, the peak location for 2-layer graphene might also be influenced by weak interactions between the stacked layers. The full-width at half-maximum of the 2D peak ($\Gamma_{2D}$) (coloration of data points) is similar in both regions and averages to $\Gamma_{2D,\,1-layer} = 33.5 \pm 4.6$ cm$^{-1}$ and $\Gamma_{2D,\,2-layer} = 32.3 \pm 4.9$ cm$^{-1}$, suggesting minor variations in strain and comparably few defects, hinting at potentially high charge carrier mobility.

**Field-effect graphene devices**. We fabricated and evaluated top-gated field-effect devices on 100-mm-diameter silicon wafers using four photolithography layers and conventional processing

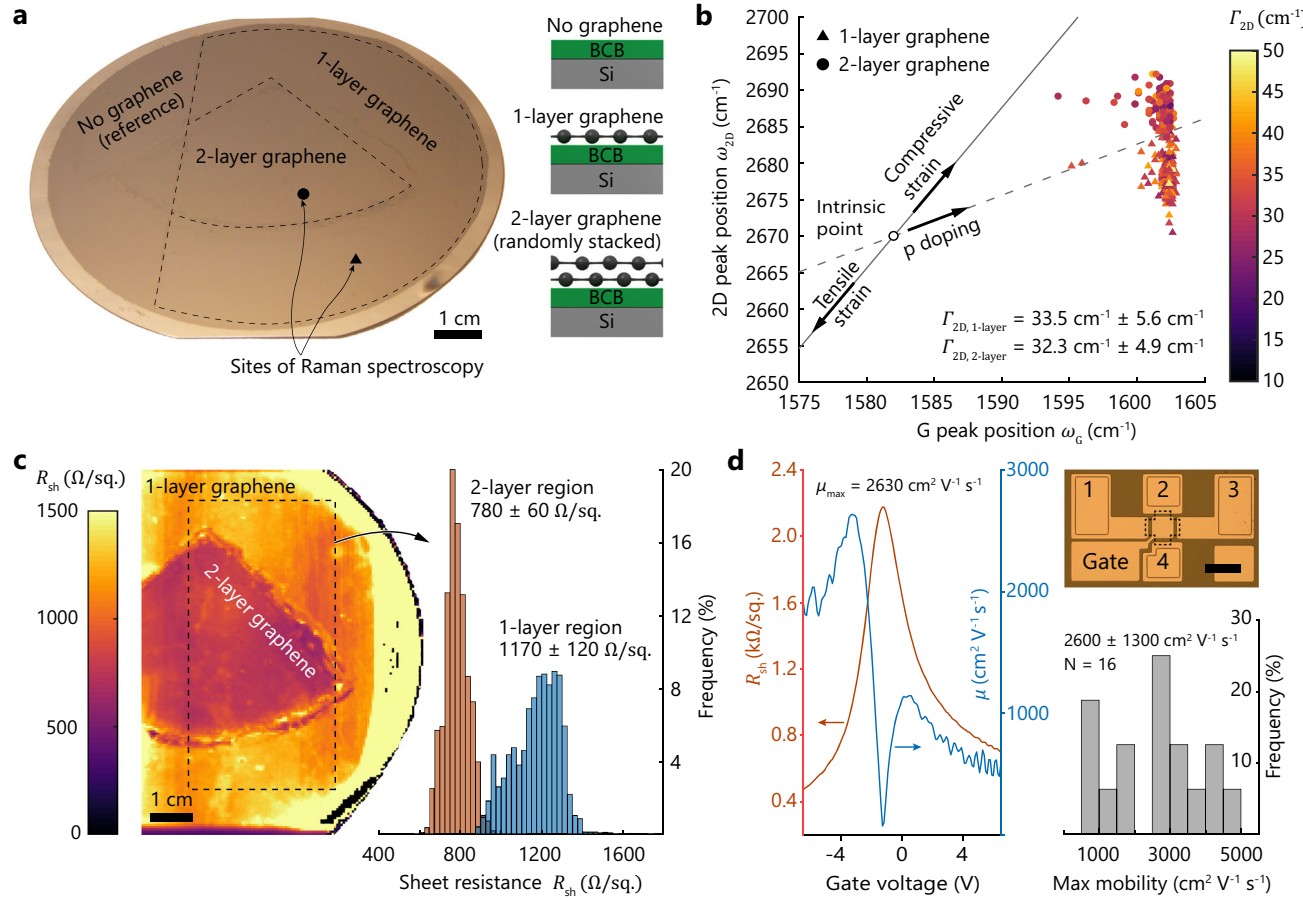

**Fig. 3 Characterization of double-layer (2-layer) graphene formed by two consecutive transfers of monolayer (1-layer) CVD graphene and wafer-level integration of top-gated field-effect graphene devices. a** Photograph of a high resistive silicon wafer with 100-mm diameter after graphene transfer. Dashed lines indicate regions of single-layer, double-layer, and no graphene (only bisbenzocyclobutene (BCB)). **b** Correlation map of Raman G and 2D peak positions ($\omega_G$ and $\omega_{2D}$, respectively), extracted from measurements in regions of 1-layer graphene (triangles) and 2-layer graphene (circles). The colormap represents the full-width at half-maximum of the 2D peak ($\Gamma_{2D}$) of single-layer graphene ($\Gamma_{2D,1\text{-layer}}$) and double-layer graphene ($\Gamma_{2D,2\text{-layer}}$). **c** Spatially resolved map of the graphene sheet resistance ($R_{sh}$) from noncontact terahertz near-field spectroscopy (left), clearly showing decreased $R_{sh}$ in the 2-layer graphene region. The histogram (right) represents the data inside the dashed rectangle (blue: 1-layer region; red: 2-layer region). **d** Top-gate voltage dependency of the graphene sheet resistance and the field-effect mobility in an integrated graphene device at room temperature (left) and histogram of the maximum filed-effect mobility ($\mu_{max}$) of 16 devices at room temperature (bottom right). Optical microscope picture of a fabricated field-effect graphene device (top right). The dashed line marks the graphene area. Electrodes 1 to 4 are used for electrical characterization by a 4-point probe van der Pauw method, while the gate voltage tuned the doping of the graphene. Scale bar: 100 μm.

technology (Fig. 3d, details see "Methods"). The extracted sheet resistance and charge carrier mobility as a function of the gate voltage of a typical device (Fig. 3d, left) show the expected characteristic behavior of gated graphene devices[80] (see "Methods" for measurement details), where the mobility is zero at the Dirac point due to a zero of the transconductance. At this point, the charge carriers convert from electrons to holes or vice versa. In our measurements, the mobility is not exactly zero because of the limited number of data points. The maximum field-effect mobility of holes from 16 devices averages to $2600 \pm 1300$ cm$^2$ V$^{-1}$ s$^{-1}$ with values up to $4520$ cm$^2$ V$^{-1}$ s$^{-1}$. The mobilities are in the same range as for identical reference devices with the same type of graphene that were fabricated on top of both BCB and quartz substrates ($2800 \pm 900$ and $1700 \pm 700$ cm$^2$ V$^{-1}$ s$^{-1}$, respectively) using conventional wet transfer (see Supplementary information). These results indicate that (1) the graphene transfer by adhesive wafer bonding using BCB yields similar final graphene quality as conventional wet transfer and that (2) BCB as substrate material yields similar graphene properties as common dielectric substrates

such as quartz. The mobilities of the field-effect graphene devices on BCB were comparable to the mobilities of the vdP devices shown in Fig. 2b, which were solely fabricated by noncontact processes to avoid potential material damage during device fabrication. This similarity suggests that the quality of the transferred graphene is preserved during device manufacturing with multiple processing steps such as high-temperature $Al_2O_3$ deposition, multiple lithographic layers, metal depositions, and etching processes. Consequently, these results suggest that our transfer methodology is compatible with large-scale manufacturing of graphene devices using conventional semiconductor process technologies.

**Transfer of MoS$_2$ and MoS$_2$/graphene heterostructures.** Our proposed transfer methodology is generic and applicable to a broad spectrum of 2D materials available on various types of growth substrates. We demonstrated this versatility by transferring a multilayer of CVD MoS$_2$ from a SiO$_2$/Si growth substrate to a 100-mm-diameter silicon wafer (see "Methods" for

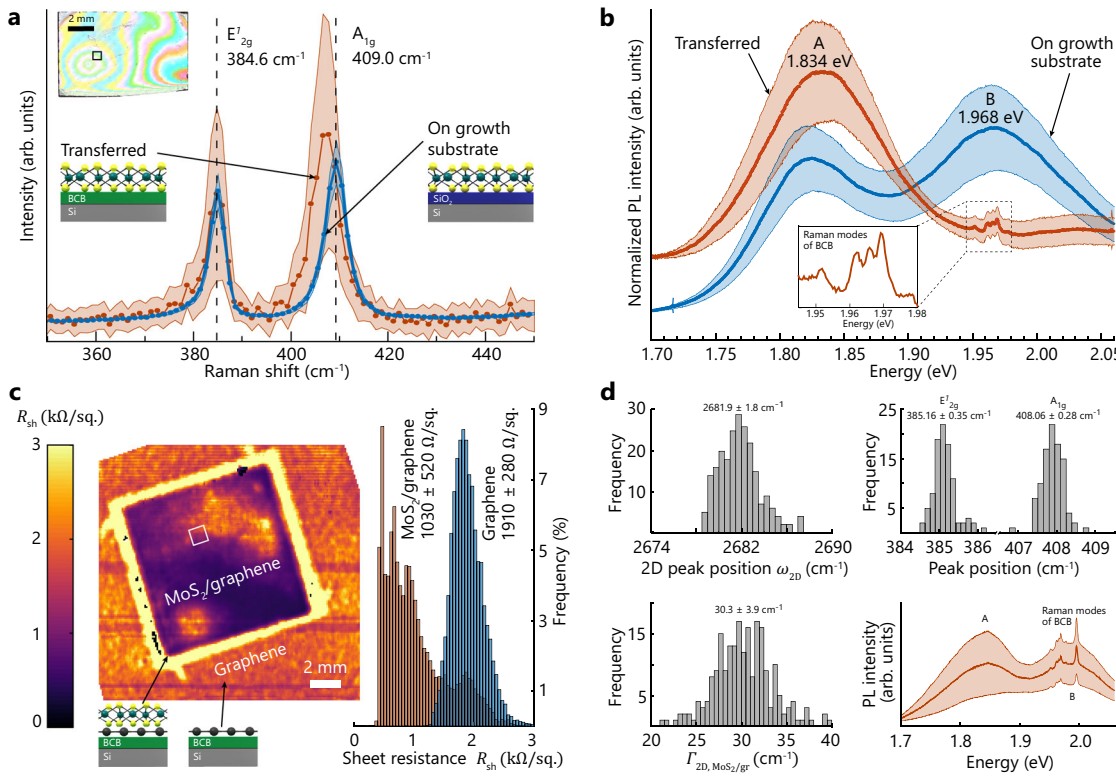

**Fig. 4 Transfer and characterization of molybdenum disulfide (MoS₂) and MoS₂/graphene heterostructures. a** Averaged Raman spectra of multilayer MoS₂ on the SiO₂/Si growth substrate before transfer (blue) and after transfer to the target wafer (red). The Raman spectra are averages of area scans with 8 × 10 measurements in 1 mm² areas (black rectangle in the photograph of transferred MoS₂) and were normalized to the intensity of the $E^1_{2g}$ mode. The shaded regions represent the standard deviation of the intensity. The vertical dashed lines indicate the peak positions of the $E^1_{2g}$ and $A_{1g}$ modes of the as-grown film before transfer. **b** Averages of 25 photoluminescence (PL) intensity spectra of pristine (blue) and transferred (red) multilayer MoS₂ with A and B excitonic transitions (shaded areas: standard deviation). The inset magnifies the region from 1.945 to 1.98 eV and shows the Raman modes of bisbenzocyclobutene (BCB), which are superimposed with the optical response of the MoS₂ layer. **c** Spatially resolved map (left) and histogram (right) of the graphene sheet resistance ($R_{sh}$) from terahertz near-field spectroscopy (resolution: 300 µm). The MoS₂/graphene heterostructure in the center (rectangular region) is surrounded by graphene, both resting on BCB. **d** Raman and PL spectroscopy of the MoS₂/graphene heterostructure, measured in a 1 × 1 mm² region on the same sample as in panel (**c**) (white rectangle). The Raman peak positions indicate the presence of both MoS₂ ($E^1_{2g}$ and $A_{1g}$ peaks) and graphene (2D peak position ($\omega_{2D}$) and the respective full-width at half-maximum ($\Gamma_{2D,MoS_2/gr}$). The averaged PL spectrum (bottom right) shows the A and B excitonic transitions of MoS₂ and the superimposed Raman modes of BCB (shaded area: standard deviation).

parameters of film growth and transfer). First, we bonded the centimeter-sized MoS₂/SiO₂/Si substrate to a BCB-coated 100-mm-diameter target wafer. Next, we submerged the bonded stack in a potassium hydroxide (KOH) solution, which permeated into the MoS₂/SiO₂ interface and detached the MoS₂ from its growth substrate. This process took tens of seconds, after which the MoS₂ remained transferred on top of the BCB layer of the target wafer. The surface roughness of the transferred MoS₂ (RMS: 1.9 nm; measurement area: 25 µm × 25 µm) remained low compared to the original surface roughness of the MoS₂ on the growth substrate (RMS: 0.8 nm; measurement area: 5 µm × 5 µm). Typical Raman spectra prior transfer (Fig. 4a) exhibited the characteristic $E^1_{2g}$ and $A_{1g}$ modes of pristine MoS₂ at 384.6 and 409.1 cm⁻¹, respectively, which are in good agreement with reported values for MoS₂[9,81–83]. After transfer, we found the peaks at 384.7 cm⁻¹ ($E^1_{2g}$) and 407.2 cm⁻¹ ($A_{1g}$), that is, without detectable shifts in the position of the $E^1_{2g}$ mode within the accuracy of our Raman setup. Since the $E^1_{2g}$ mode is highly sensitive to uniaxial strain[84], we conclude that the strain in the transferred MoS₂ film was the same as present on the growth substrate and was not significantly affected by our transfer methodology. In contrast, the $A_{1g}$ mode is sensitive to changes in the substrate material, which could cause

the observed shift of 1.8 cm⁻¹ [85,86]. To further investigate the quality of the transferred MoS₂ film, we compare the average of 25 normalized photoluminescence (PL) intensity spectra of pristine and transferred MoS₂ (Fig. 4b). Before transfer, we found two prominent peaks at 1.821 and 1.968 eV, which correspond to A and B excitonic transitions, respectively[87]. After transfer, we observed a drastic increase in the A/B intensity ratio since the B-excitation vanishes. A high A/B intensity ratio indicates a high quality of the MoS₂ film[88]. The A exciton transition energy is blue-shifted by 13 meV to 1.834 eV (see Supplementary Fig. 11 for statistics), which we attribute to the change of the substrate material from SiO₂ (before transfer) to BCB (after transfer)[86]. The additional features of the PL intensity between 1.96 and 1.97 eV are Raman modes of the underlying BCB, and hence do not belong to the excitonic transitions in the MoS₂ (see Supplementary information for the Raman spectrum of BCB). We formed MoS₂/graphene heterostructures by stacking a centimeter-sized layer of MoS₂ onto a sheet of graphene in two consecutive transfers (see "Methods"). The map of the graphene sheet resistance extracted from THz spectroscopy indicated the presence of graphene in both the MoS₂/graphene heterostructure and its surroundings (1030 ± 520 and 1910 ± 280 Ω sq⁻¹, respectively).

In the region of the $MoS_2$/graphene heterostructure, $R_{sh}$ is significantly reduced. Since the high sheet resistance of a sole $MoS_2$ layer is not directly detectable in this measurement, we speculate that interactions between graphene and $MoS_2$ caused this variation. Raman and PL spectroscopy of the $MoS_2$/graphene heterostructure confirmed the presence and integrity of both materials (Fig. 4d and Supplementary information). The position of the 2D peak of graphene ($\omega_{2D}$) averaged to $2681.9 \pm 1.8$ cm$^{-1}$ with a full-width at half-maximum ($\Gamma_{2D,MoS_2/gr}$) of $30.3 \pm 3.9$ cm$^{-1}$. This value is in good agreement with the results in Fig. 3, which hints at the integrity of the graphene even after stacking. The Raman $E^1_{2g}$ mode of $MoS_2$ exhibited a minor shift from $384.26 \pm 0.17$ to $385.16 \pm 0.35$ cm$^{-1}$, while the position of the $A_{1g}$ mode remained unchanged ($408.06 \pm 0.28$ cm$^{-1}$) (see Supplementary information on material characterization prior to transfer). The averaged PL spectrum featured the distinct peaks of A and B excitonic transitions in $MoS_2$ with high A/B ratio and superimposed Raman modes of BCB. We further demonstrated the versatility of the methodology by reversing the order of stacked materials, which formed graphene/$MoS_2$ heterostructures by successfully transferring graphene onto previously transferred $MoS_2$ layers (see Supplementary information). Taken together, the Raman and PL measurements confirm that the layer quality of $MoS_2$ is preserved during the transfer and the stacking to heterostructures. Hence, our methodology is potentially suitable for large-area transfer of $MoS_2$ films and their heterostructures.

## Discussion

We have demonstrated a generic methodology for transfer and large-area integration of 2D materials and their heterostructures on wafer level, which is compatible with conventional semiconductor fabrication lines. Our approach avoids manual handling of released layers and relies only on tools, processes, and materials that are already established in the semiconductor industry. To illustrate its applicability, we manufactured graphene devices on a 100-mm-diameter silicon, performed a statistical analysis of the material characteristics, and demonstrated that our transfer method is, in principle, compatible with the conventional process technology. THz near-field inspection confirmed the uniform coverage of graphene in large areas while the evaluation of vdP devices yielded a high charge carrier mobility of up to $2800 \pm 100$ cm$^2$ V$^{-1}$ s$^{-1}$, which is among the highest reported values for similar substrate sizes and graphene quality. As representatives for transition metal dichalcogenides, we demonstrated the transfer of $MoS_2$. Raman and PL spectroscopy indicated little strain in both transferred graphene and $MoS_2$. We fabricated top-gated field-effect graphene devices ($\mu = 2600 \pm 1300$ cm$^2$ V$^{-1}$ s$^{-1}$), which required multiple lithography steps in combination with conventional deposition and etching processes after the transfer. This result demonstrates that the proposed transfer process is compatible with large-scale device manufacturing without causing significant deterioration in the quality of the 2D material.

The transfer method replicates the surface topography of the growth substrate of the 2D material in an adhesive layer on the target wafer by molding at low viscosity of the adhesive. After removal of the growth substrate, the 2D material remains on the target wafer in the same shape as it was synthesized on the growth substrate. This feature minimizes defects or additional strain in the 2D material by avoiding deformation. In addition, it reduces wrinkles, which may result from excess material due to different surface topography of the growth substrate and the target wafer. We hypothesize that this feature contributes to preserving similar mechanical properties of the transferred 2D materials. The ability

of the adhesive layer to mold repeatedly allows the stacking of 2D materials to heterostructures. In this case, merely the bottommost 2D material is in contact with the adhesive layer. This ensures that the other 2D materials and the interfaces between layers are not contaminated by polymer residuals. To demonstrate this capability, we fabricated graphene/hBN heterostructures, double-layer graphene, and $MoS_2$/graphene heterostructures on large areas by consecutive transfers. In the resulting stacks, Raman, PL, and THz spectroscopy confirmed the uniform coverage and high quality of the graphene and $MoS_2$ layers.

We anticipate that our transfer methodology is applicable to 2D materials in general, independent of the size and the type of growth substrate. Consequently, the barrier is comparatively low for the industry to consider this methodology for the integration of 2D materials on top of conventional IC and microsystem device substrates in the back end of the line. Hence, we expect that our methodology has the potential to accelerate advancements in fundamental 2D material science, as well as in a wide range of application areas, spanning over electronics, sensing, and photonics.

## Methods

**Transfer of graphene**. First, a 2.5 μm-thick adhesive layer of BCB (Cyclotene 3022-46, Dow Inc.) was spin-coated at 5000 r.p.m. on the 100-mm target wafer. A softbake on a hot plate at 100 °C for 4 min removed solvents and solidified the adhesive layer. Next, the target wafer was brought in proximity to a 100-mm sheet of monolayer CVD graphene on copper foil (Graphenea Inc.) with the graphene facing the BCB layer. This stack was bonded in a commercial wafer bonder (SB8, SÜSS MicroTec SE) at 190 °C for 20 min using a bond force of 3 kN (bond pressure: 0.95 bar) in a nitrogen atmosphere. Etching the copper foil in FeCl$_3$ solution and rinsing in deionized water for 30 min uncovered the graphene, transferred to the target wafer.

**Formation of graphene/hBN heterostructures**. A 100-mm wafer was spin-coated with a 2.5-μm-thick layer of BCB (Cyclotene 3022-46, Dow Inc.; spinning speed: 5000 r.p.m., softbake: 100 °C for 4 min). A 2.5 cm × 2.5 cm sheet of multi-layer hBN (thickness: 2–5 nm) synthesized by CVD on copper foil (2D Semi-conductors Inc.) was placed on top of the target wafer (hBN facing the BCB layer) and bonded at 190 °C for 20 min at a bond force of 250 N (bond pressure: 4 bar) in a nitrogen atmosphere. Etching of the copper foil in FeCl$_3$ and subsequent rinsing in deionized water for 30 min uncovered the hBN, transferred to the target wafer. To form a graphene/hBN heterostructure, a 100-mm sheet of CVD graphene on copper foil was placed on top of the target wafer and bonded at 190 °C for 20 min (bond force: 3 kN, bond pressure: 0.95 bar, vacuum atmosphere). Etching of the copper foil in FeCl$_3$ and rinsing in deionized water for 30 min uncovered the graphene, which formed a graphene/hBN heterostructure with the previously transferred hBN.

**Formation of 2-layer graphene**. A 2.5-μm-thick adhesive layer of BCB (Cyclotene 3022-46, Dow Inc.) was spin-coated at 5000 r.p.m. on the 100-mm target wafer. Softbake on a hot plate at 100 °C for 4 min removed solvents and solidified the adhesive layer. The target wafer was brought in proximity to a quarter of a 100 mm sheet of monolayer CVD graphene on copper foil (Graphenea Inc.) with the graphene facing the BCB layer. This stack was bonded in a commercial wafer bonder (Suss-SB8) at 190 °C for 20 min using a bond force of 600 N (bond pressure: 3.1 bar) in a nitrogen atmosphere. Etching of the copper foil in FeCl$_3$ solution and rinsing in deionized water for 30 min uncovered the graphene, transferred to the target wafer. To form 2-layer graphene, a 100-mm sheet of CVD graphene on copper foil was placed on top of the target wafer and bonded at 190 °C for 20 min (bond force: 3 kN, bond pressure: 0.95 bar, vacuum atmosphere). Etching of the copper foil in FeCl$_3$ and rinsing in deionized water for 30 min uncovered the graphene, which formed 2-layer graphene with the previously transferred layer. Note, the bond force was adapted to account for the difference in size of the growth substrates. To form reference areas for THz spectroscopy, the 100-mm copper foil was cut into two pieces and graphene was partially removed from one piece by etching in O$_2$ plasma prior transfer. During transfer, both parts were placed side by side, covering the entire target wafer.

**Fabrication and evaluation of vdP devices**. A graphene/hBN heterostructure was formed on an oxidized 100-mm silicon wafer (100 nm SiO$_2$) (see above). Thermal evaporation of 20 nm Ti and 200 nm Au through a shadow mask formed electrodes (300 μm × 300 μm) on top of the transferred graphene layer. Ablation by a pulsed femtosecond laser electrically insulated individual vdP devices with an edge length of ~3 mm. These noncontact processes prevent potential material degradation by

lithographic layers in contact with the graphene. For device characterization, the wafer was resting on an electromagnet (Wuxue Wen Fang Electric Co. Ltd, model WF-P25/20), which was placed inside a probe station (Cascade Microtech Inc.). Four-point probe measurements of the voltage across the device edge (device current: 100 μA), and the Hall voltage (device current: 1 mA at 1.28 ± 0.17 V; magnetic flux: 28.8 mT) yielded the sheet resistance, carrier density, and mobility of the transferred graphene. Electrical measurements were performed under ambient conditions at room temperature, using a parameter analyzer (Keithley SCS4200).

**Fabrication and evaluation of top-gated field-effect devices**. Graphene-based top-gated field-effect devices were fabricated on a 100-mm silicon wafer. First, large-area graphene (10 cm × 10 cm) was transferred by adhesive wafer bonding (see above). Contacts to graphene were fabricated by sputter deposition of 25 nm Ni and e-beam evaporation of 135 nm Al, followed by a lift-off process in acetone (60 °C, 30 min) (1st photolithography layer). The graphene sheet was then patterned by etching in oxygen plasma (2nd photolithography layer)). As a top-gate dielectric, 40 nm $Al_2O_3$ was deposited by atomic layer deposition and top-gate electrodes were fabricated by e-beam evaporation of 10/150 nm Ti/Al (3rd photolithography layer). To access the graphene metal contacts under $Al_2O_3$, reactive ion etching with $SF_6$ and $O_2$ was used to open vias through the $Al_2O_3$ (4th photolithography layer). All resist layers were removed by immersion in acetone (60 °C, 30 min). In this process sequence, the first and second lithographic layers were in direct contact with the graphene, which may have degraded the graphene quality. However, encapsulation of the graphene before fabricating the contacts eliminates this potential source of degradation[89]. Electrical measurements were performed under ambient conditions at room temperature. The sheet resistance ($R_{sh}$) was measured by 4-probe vdP method with top gating. A current ($I_{12}$) was forced between electrodes 1 and 2 (Fig. 3d), while the voltage drop between electrodes 3 and 4 ($V_{34}$) was measured. The sheet resistance ($R_{sh}$) was calculated by[90]

$$R_{sh} = \frac{V_{34}}{I_{12}} \frac{\pi}{\ln 2} \qquad (1)$$

and the mobility ($\mu$) was then derived by

$$\mu = \frac{1}{C_G} \frac{dG_{sh}}{dV_G} \qquad (2)$$

where

$$G_{sh} = \frac{1}{R_{sh}} \qquad (3)$$

and $V_G$ and $C_G$ are the gate voltage and the capacitance of the dielectric per unit area ($k = 7$ for $Al_2O_3$), respectively.

**Raman spectroscopy of graphene**. Raman measurements were conducted using a confocal Raman microscope (alpha 300R, WITec GmbH), equipped with a 532-nm laser, which was coupled to the microscope using a single-mode optical fiber. A power meter in the optical path determined the laser power (0.6 mW), which ensured a high reproducibility of the experiments. The laser power was chosen to stay in the noninvasive regime[91]. The microscope was connected to a 600 mm ultrahigh-throughput spectrometer (UHTS 600) using a photonic fiber. An 1800 g mm$^{-1}$ grating was used for the dispersion and an EMCCD camera for the detection of the scattered light. To compensate the topography of the substrate, a ×50 objective with a NA of 0.5 was chosen for the measurements. Note that we have used a high-resolution 1800 g mm$^{-1}$ grating in combination with the 600 mm spectrometer to obtain more intense signals with a pixel resolution of 0.1 cm$^{-1}$ @ 2800 relative cm$^{-1}$. For each measurement site, the spectra were acquired in areas of 25 μm × 25 μm with 25 single spectra per line and 25 lines per site (integration time: 3 s). Peak positions were extracted by a Lorentzian fit. To ensure the validity of the extracted values, only data with a peak intensity exceeding a threshold of ten counts and sufficient separation to the noise level were considered.

**THz near-field spectroscopy**. We used THz-TDS in transmission mode to measure the sheet resistance of the transferred graphene on the target wafer. The samples were prepared on high-resistivity wafers with a resistivity of >10$^4$ Ω·cm, to avoid inadvertent absorption of the THz signal by the target wafer. For the measurements, we used a THz pump/probe setup equipped with a femtosecond fiber laser (generating pulses of 100 fs duration and 780 nm center wavelength) to pump a terahertz emitter and to gate a photoconductive near-field micro-probe detector (TeraSpike TD-800-X-HR-WT). By measuring the THz transmission through the sample, we extracted the local sheet resistance of the graphene[71,72]. Lateral scanning of the sample with the tip of the near-field detector held at close distance yielded a spatially resolved map of the graphene sheet resistance with a pixel size of a few hundred micrometer. See Supplementary information for a description of the setup and the estimation of the pixel size.

**Synthesis of MoS$_2$ monolayers**. The MoS$_2$ films were grown in a CVD furnace with a 40 mm-diameter quartz tube as the reactor, similar to that described by Luo et al.[92]. In short, the SiO$_2$/Si growth substrates were located 12–15 cm downstream from the center of the heated tube (500 mm). MoS$_2$ powder (Alfa Aeasar, 99%

purity) was loaded into a quartz boat and placed in the center of the quartz tube of the furnace. The furnace was purged by forming gas (Ar 95%/H$_2$ 5%). For the growth, the furnace temperature was ramped up to 950 °C within 30 min, which was then maintained for 60 min. After the growth, the quartz tube was cooled down within 4 h. The growth yielded quasi-continuous films of MoS$_2$ multilayers, with micrometer-sized domains.

**Transfer of MoS$_2$**. First, a 2.5-μm-thick adhesive layer of BCB (Cyclotene 3022-46, Dow Inc.) was spin-coated at 5000 r.p.m. on the 100-mm target wafer. A softbake on a hot plate at 100 °C for 4 min removed solvents and solidified the adhesive layer. Next, a multilayer of MoS$_2$, grown on SiO$_2$/Si chips (10 mm × 7 mm), was placed on top of the BCB layer (MoS$_2$ facing the target wafer) and surrounded by silicon dummy chips of the same height to ensure a uniform force distribution during the bonding process. This stack was bonded at 190 °C for 20 min using a bond force of 400 N (bond pressure: 2 bar) in a nitrogen atmosphere, which attached the MoS$_2$ layer on its growth substrate to the target wafer. Etching in O$_2$/SF$_6$ plasma cleaned the edges of the growth substrate from potential BCB residuals. Submersion of the bonded stack in KOH solution detached the growth substrate from the MoS$_2$ layer by permeation of KOH into the MoS$_2$/substrate interface within tens of seconds. The MoS$_2$ layer remained transferred on the target wafer.

**Formation of MoS$_2$/graphene heterostructures**. First, the silicon target wafer (diameter: 100 mm; resistivity: >10$^4$ Ω·cm) was spin-coated with a 2.5 μm-thick BCB layer (Cyclotene 3022-46, Dow Inc., spinning speed: 5000 r.p.m.; softbake: 100 °C for 4 min). A quarter of a 100-mm sheet of monolayer CVD graphene on copper foil (Graphenea Inc.) was bonded to the target wafer with the graphene facing the BCB layer (bond temperature: 190 °C; bond time: 30 min; bond force: 600 N with a resulting bond pressure of 3.1 bar, nitrogen atmosphere). Etching of the copper foil and rinsing in deionized water for 30 min uncovered the transferred graphene on top of the BCB on the target wafer. Next, a multilayer of MoS$_2$ on a SiO$_2$/Si chip (10 mm × 10 mm) was placed on top of the transferred graphene (MoS$_2$ facing the graphene) and surrounded by silicon dummy chips to ensure a uniform force distribution over the wafer during the following bonding process. Bonding at 190 °C for 30 min attached the MoS$_2$/SiO$_2$/Si chip to the target wafer (bond force: 750 N with a resulting bond pressure of 3 bar, vacuum atmosphere). Etching in O$_2$/SF$_6$ plasma cleaned the edges of the wafer while a resist mask protected the remaining surface of the target wafer. Submersion of the bonded stack in acetone stripped the resist mask and detached the growth substrate, leaving the MoS$_2$ film transferred on top of the graphene, forming a MoS$_2$/graphene heterostructure.

**Raman and PL measurements of MoS$_2$ films**. Raman and PL measurements were carried out in a confocal Raman system (alpha 300, WITec GmbH) in ambient conditions. For all measurements, an excitation laser with a wavelength of 532 nm and a 100× objective was used. For Raman measurements, the excitation power was 1 mW (1800 g mm$^{-1}$ grating), which achieves a spectral resolution of about 1.2 cm$^{-1}$. For PL measurements, the excitation power was 0.2 mW (600 g mm$^{-1}$ grating).

## Data availability
The data that support the findings of this study are available from the corresponding author upon reasonable request.

## Code availability
The code that supports the findings of this study is available from the corresponding author upon request.

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

## Acknowledgements

This project has received funding from the European Union's Horizon 2020 research and innovation program under grant agreement no. 825272 (ULISSES—www.ulisses-project.eu), the European Research Council through the Starting Grant M&M's (No. 277879), Graphene Flagship (785219, 881603), VINNOVA (2017-05108), VR (2015-05112), the Swedish Foundation for Strategic Research (SSF) (GMT14-0071), the German Ministry for Education and Research (BMBF, GIMMIK 03XP0210), the German Ministry for Economic Affairs and Energy (BMWi) and the European Social Fund in Germany (AachenCarbon, FKZ: 03EFLNW199), and the China Scholarship Council (CSC) through a scholarship grant. We thank Daniel Neumaier of AMO GmbH, Aachen for fruitful discussions. Graphene Flagship 2D Experimental Pilot Line (Acronym: 2D-EPL), Grant no. 952792.

## Author contributions

A.Q., X.W., K.B.G., N.R., G.S. and F.N. conceptualized the methodology. A.Q. performed significant part of the wafer bonding experiments, part of the white-light interferometric measurements, the entire fabrication and evaluation of the vdP devices, SEM inspection, X.W. performed part the wafer bonding experiments and white-light interferometric measurements. S.W. performed and A.Q. evaluated the Raman measurements of graphene/hBN heterostructures, 1-layer graphene and 2-layer graphene. S.S. performed and evaluated the THz spectroscopy and A.Q. analyzed the data statistically. B.U. and Z.W. designed, fabricated, and characterized top-gated field-effect devices from transferred graphene by A.Q. (bonding) and M.O. (wet transfer). S.L. synthesized the $MoS_2$, M.P. and O.H. performed all related Raman and PL measurements. M.P., O.H. and G.S.D. analyzed the spectroscopic measurements of $MoS_2$. M.L., K.B.G., N.R., G.S. and F.N. supervised the project. All authors contributed to drafting and revising the manuscript.

## Funding

## Competing interests

A.Q., X.W., K.B.G., N.R., G.S., and F.N. are co-inventors on a patent application describing a method for 2D material transfer (PCT/EP2020/051831). The other authors have no competing interests.
