## [Peer Review File · Nature Communications]

Reviewers' comments:

Reviewer #1 (Remarks to the Author):

Authors report a transfer method for graphene and 2D materials based on wafer bonding, where an adhesive layer (BCB) is deposited on the target substrate. By applying both pressure and heat, the 2D material, together with the substrate (e.g. Cu foil) where it has been grown are bonded to the target sample. The growth substrate is chemically removed, leaving the 2D material onto the CBC layer deposited on the target substrate. The process can be repeated to form stacks consisting of two or more 2D materials.

Whereas the electronic and structural properties of CVD-grown graphene are virtually indistinguishable from their mechanically exfoliated counterpart, a wafer-scale transfer process is still missing and is the major bottleneck for the integration of graphene and 2D materials onto semiconductor (back end of line) production. Therefore, the topic of the paper is highly relevant and timely.

The manuscript is well organized and properly communicated, however -in the view of this reviewer- not suitable for publication in Nature Communication in the present form due to the following reasons:

1. Graphene transfer: it is stated (lines 105-106) that "the adhesive layer molds into the surface topography of the growth substrate". If on one hand this is beneficial due to minimization of strain and wrinkling, on the other foils have a topography which is rarely flat. If the macroscopic surface roughness is "locked" into the adhesive tape it could affect not only the performance of the graphene but also and more significantly planar processing such a lithography. Authors should comment.

2. Graphene mobility: mobility and sheet resistance are very similar to conventional wet-transfer methods, which have same or even better scalability without requiring a bonding process and the associated equipment. Therefore, it is difficult to see the advantage of the method. Also, figure 2(a) shows a much larger number of devices compared to the 18 (graphene) + 9 (graphene/hBN) devices measured. It would have been useful to have a larger statistic as well as an idea of the scattering of mobility across the wafer surface.

3. Heterostructures: interface is the most critical part of any heterostructure. The manuscript neglects this aspect entirely, thus not providing any solid proof that the method is suitable for fabrication of heterostructures. Also, the poor mobility of graphene on hBN (worse than on the CBC film) raises additional concerns about the applicability of the method to form heterostructures. The only conclusion that one can take from this results are that either the hBN used is extremely poor or that the method -when applied to stack 2D materials- degrades their quality significantly.

Also, some of the statements in the manuscript are not correct, as follows:

- Lines 26- 27: "current transfer methods cause degradation of the 2D materials"
multiple studies have shown that the quality of 2D materials is fully "restored" if a suitable cleaning process is applied. For example, Purdie et al. showed (Nature Comm. 9, 5387) that graphene intentionally contaminated with solvent and polymers are indistinguishable from pristine sample if a suitable cleaning process is applied.

- Lines 28-29: "here, we report a generic methodology for dry-transfer and large-area integration of 2D materials and their heterostructures by adhesive wafer bonding"

The transfer method described cannot be considered dry and such statement is misleading. Indeed, similar to other transfer methods, the growth substrate is removed by wet etching (FeCl₃ in the

case graphene and hBN, KOH for MoS₂). For a review of wet and dry transfer, see for example De Fazio et al. 13, 8926

- Lines 298-299 "We also fabricated high mobility field-effect graphene devices"
The mobility achieved is very modest and the statement above is entirely misleading.

- Lines 314-315: "Consequently, the barrier is low for industry to incorporate this methodology into their manufacturing lines as a standard for back end of the line integration"

Incorporating graphene and 2D materials into semiconductor production lines is certainly not easy and the statement above is misleading. It would be correct to say that -if the method was providing good results, which unfortunately is not the case- it should be considered by industry as it is based on processes already available in the industry.

In summary, despite the approach is very interesting and provides a possible route towards wafer-scale integration of graphene and other 2D materials into semiconductor production lines, the results are very modest and similar to conventional wet-transfer methods. The industry requirements are significantly higher, e.g. carrier mobility $>10,000$ cm²/Vs at carrier density $\sim 10^{12}$ are essential for (opto)electronic devices (see for example Romagnoli et al, Nature Review Materials 3, 392).

I would strongly encourage working on the optimization of the method which -if results are significantly improved- can have a very strong impact on graphene and 2D materials technology. Unfortunately, in the view of this reviewer, the current results are very far from creating any significant improvement over conventional methods used for graphene transfer.

Reviewer #2 (Remarks to the Author):

The submitted work describes a method able to realize wafer level transfer and wafer level fabrication of two dimensional materials and heterostructures as well as devices thereon. The potential of the method is demonstrated using a 100 mm carrier wafer. Large area transfer of graphene/BCB, graphene/hBN, graphene/graphene and MoS₂/BCB was realised. Reasonable good material and devices properties are demonstrated. The drawback of the method is the large waviness of the surface after transfer. Nevertheless, due to the large potential of the method demonstrated, the paper is worth to be published in Nature Communications, if the authors will take into account the below listed remarks and improve the compelling of the results presented:

(1) Abstract line 31: Misleading statement. "We transferred monolayers of graphene and molybdenum disulphide (MoS₂) from their growth substrates to 100 mm diameter silicon wafers and fabricated field-effect graphene devices with high carrier mobility up to 2630 cm²/(Vs)." Firstly, it is unclear what dimensions the transferred graphene and MoS₂ had was it 4 inch or 100 x 100 nm². Secondly, please describe the type of graphene field effect transistor in a separate sentence to avoid misleading conclusions.

(2) Introduction: The introduction (first part) do not highlight the real advantages of 2D materials, give a comparison of Si properties and 2D properties when it is scaled to 10 nm.

(3) The understanding of the "Methodology ..." chapter is not easy.

(4) Figure 1: The Figure is designed in 3D style. It would be better to use the common 2D used in flow charts for semiconductor processing. When this is done in a proper way the process flow would be easier to understand.

(5) Line 81 to line 82: "All transferred layers and heterostructures were of high quality" The property of the quality is not specified!

(6) Line 110: "..., the chemical stability of the ... wafer processing." Is doubled in line 119. Please reformulate.

(7) The authors carry out a non-proper discussion of their sheet carrier resistance. The demonstrated values are not so exciting.

(8) One of the main statements of the paper is that the two dimensional materials are not contaminated by polymers used in the process sequence. The authors use BCB and transfer of the two dimensional materials onto BCB. So, at least the backside is contaminated with a polymer material leading to p-type doping of graphene. What is the advantage of BCB, a material developed for low-k applications? Later on, in the graphene FET section, the authors' uses standard photolithography technology to fabricate the devices. This thwarts the idea to avoid polymer contaminations or polymer free processing. To ensure that the authors have to developed a polymer contamination free process. Please give details for photoresist removal techniques and prove the contamination free processing.

(9) Abstract: The authors state that they developed an industrial compatible process. For our understanding this is a questionable statement, because shadow masks and femtosecond laser ablation for structuring (line 151-152) are not very common

(10) Page 6, line 146: The authors removed the copper foil with FeCl_3 and rinsed the in deionized water. What else was done to remove iron contaminations? Please list in methods section all used cleaning procedures used before and after the processing steps. Alternatively, they can be described in the body text of the publication.

(11) Please indicate the thickness of the multilayer hBN.

(12) Line 177-180: Tens of micrometres are not a microscopic resolution. Please delete the sentences related to this statement. And give the real spot size and resolution used for the mapping.

(13) Please do not use "wafer level transfer" especially for multilayer transfers, because it was not demonstrated. Transfer of "large areas" would be an example of a term closer to reality.

(14) Please comment on the differences in sheet carrier resistivity on Fig.2 and 3 for single layer graphene. The values differ by a factor of 4. The distribution of the resistance value is also much broader in Fig. 3.

(15) Line 248: The authors extract a maximum field-effect mobility of $2630 \text{ cm}^2 \text{ V}^{-1} \text{ s}^{-1}$ and state that the quality of the graphene is preserved after transfer. To prove this statement properties of the as grown graphene have to be given. Furthermore, it would be interesting to know how the values were extracted.

(16) Line 279: The authors mention "intercalation at the $\text{MoS}_2/\text{SiO}_2$ ". The term intercalation is not used properly. Intercalation means introduction of additional material (atoms, molecules) without lifting layers (see for example the overwhelming literature of graphite intercalation). Therefore, the question arises have the authors noticed intercalation effects as a consequence of their transfer process or processing sequences. This question is important, because intercalation (see

for example Ge intercalation of graphene published by Seyller) may change the electronic properties of the two dimensional materials.

(17) Can the authors add a comment on the shift of the A1g mode being approximately 1 cm⁻¹. Please comment also on the change in the peak heights. Furthermore, as in the case of the transferred graphene not only two spectra have to be shown, but an analysis over a mapping area (see for example Fig. 2d). This is required, because it was stated that large area single layer MoS₂ was grown and transferred. This has to be demonstrated and cannot be done with two single spectra.

(18) Yield data of the fabricated devices have to be added to the paper.

(19) Line 292: "... performed a statistical analysis of the resulting device and materials characteristics". Statistical analysis of transistor performance are not given in the paper. A comparison of charge carrier mobilities obtained in graphene FETs and vdP structures is not given.

(20) Please add the temperature of the electrical measurements.

(21) Fig. 3d: Why the mobility of the charge carriers shows a minimum, if it is assumed that the mobility scales with concentration lower charge carrier concentration lead to higher charge carrier mobility. Please comment

Reviewer #3 (Remarks to the Author):

Review of Large-Area Integration of Two-Dimensional Materials and Their Heterostructures by Wafer Bonding

Quellmalz et al. investigated a new 2D material transfer method which is large-area, allows stacking into heterostructures, does not require ultra-specialised equipment, and doesn't completely degrade the transport properties. Generally the work is well presented, appropriate methods were used for characterisation, and data is of high quality, and sufficient detail is given for others to reproduce the work. I believe the method is likely to be useful for the 2D materials community as well as industry, and that Nature Communications is an appropriate journal for the work. However, in my opinion the authors probably overstate how gentle their process is, at least based on the presented evidence. I also have a few other points which require clarification.

1. THz-TDS map from Figure 2c. Around the edges of the hBN square there are noticeably damaged areas on the graphene sheet- or at least areas which have sheet resistances about twice the rest of the sheet. What has caused this? When making heterostructures with your method is this interface degradation unavoidable?

2. For the hBN/graphene vs. graphene results, how can you tell that you are not just damaging the hBN during transfer and that is the reason why there is no increased device performance for the

3. Why no MoS₂ transistor measurements? Seeing as you are growing on Si, a comparison of devices made on the as-grown substrate vs transferred (using same fab methods, same substrate etc. just with BCB on transferred substrate) would be nice evidence that you aren't damaging the MoS₂ by transferring. Otherwise, for MoS₂ the evidence for large- scale transfer preserving the properties you present is a lot weaker in comparison to graphene, with single Raman/PL spectra- and no mapping. It is valuable to show the MoS₂ data, but statements such as: "Hence, our methodology is suitable for large area transfer of MoS₂ films." With no electrical measurements I think "potentially" is required, at least.

4. Several times throughout the paper is strong claims are made such as (line 106) ...substrate

without exerting excessive pressure on the 2D material, and hence without damaging, wrinkling or straining the 2D material. This is written as if no damage or wrinkling or strain occurs at all. I think these claims should at least include the word 'minimised', for instance. If you have Raman data of the graphene pre-transfer to suggest the strain is similar before your process? Then a similar claim with the word 'minimised' included would be appropriate.

Alternatively, you could transfer graphene, measure strain and d-peak via Raman (+THz-TDS), re-transfer the same graphene sheet and remeasure (Raman + THz), this would also provide very strong evidence of a lack of degradation due to processing. As presented, the data only suggest that the method is quite good at transferring graphene compared to other methods, but it is not possible to tell if it is being damaged or not- especially based on transport properties.

5. Re: data in Figure 2b. You measured 18 graphene and 9 graphene/hBN devices, which seems like a reasonable sample size. Can you mention the device yield? And if any devices failed, what do you attribute this to?

6. It is true that "neither the interface between the first and second 2D material nor the second 2D material are exposed to any polymer carrier nor adhesive that may potentially degrade the 2D material properties by contamination" and is a strength of your method. However, surface water will be trapped permanently. Could this have an effect on heterostructures? Can all surfaces be heated during the process?

7. Line 161 For the transferred graphene monolayer on BCB, the extracted mobilities are in the higher range of previously reported values for CVD graphene transferred on large areas by various methods, including wet transfer with polymeric carrier layers^{38,55}, face-to-face transfer by bubble formation⁵⁶lamination⁴⁶. In ref 46 by Shivayogimath et al (Fig 5b), the THz-TDS results for mobility of large-area transfer are similar to your results (slightly less) for etching transfer, but closer to 4000 cm²/Vs for PVA. The text can be updated so not to suggest your values are the highest reported for large-area transfer of graphene.

8. Line 152: materials by femtosecond-laser ablation to define van-der-Pauw (vdP) devices for characterization of the electrical properties of the 2D materials. Were you able to pattern the graphene without damaging the underlying BCB or substrate? if so, how did you determine this? If not, could this effect device properties?

9. Regarding THz-TDS maps, you use a near-field set up with spatial resolution down to 10 μm. However, at least for far-field set ups (with perhaps 100 μm spatial resolution) the actual transport length scale is ≈10-100 nm (see 2D Mater. 4 (2017) 042003 Table 1 and discussion on p18) and depend on graphene mobility for instance. I think the authors should state the transport length scale for their set up/graphene and make it clear for the reader that for THz: spatial resolution ≠ transport length. This will also lead to analysis of your data which supports the quality of your method, as it seems your sheet R values match well between vdP and THz, suggesting you do not have significant defects/inhomogeneity on the larger scale that can often effect sheet R in larger devices.

10. Line 185: Note, the THz near-field measurement in Figure 2c was performed on a different sample than the vdP measurements in Figure 2a and 2b. I presume this is because the metal contacts would saturate the THz-TDS signal? If so, that could be mentioned. Otherwise, with 10 μm resolution you should be able to get many pixels inside each vdP device for direct comparison.

11. Fig 3 a inset: I feel the 2-layer graphene inset gives the impression of Bernal stacking. Suggest changing or adding text for the readers who mainly look at figures.

12. Figure 3 results: Why is the sheet resistance for 1-layer graphene of this wafer so much higher than the Figure 2 results?

13. Fig 4: Good to see before and after transfer Raman of MoS₂, but why not the before and after transfer for photoluminescence measurements?

14. Hall measurements were performed at 1 mA. Which voltage was required to generate 1 mA?

15. Supp Info: For the THz resultsfor graphene/hBN, is it necessary to have a second reference area with no graphene but substrate+BCB+hBN?

Point-by-Point Response to the Reviewer Comments

We sincerely thank the reviewers for carefully reviewing our work and the constructive and insightful comments, which have helped us to improve the paper further. Our detailed responses to the reviewer's comments are provided in the following.

Reviewer #1 (Remarks to the Author):

Reviewer comment:

Authors report a transfer method for graphene and 2D materials based on wafer bonding, where an adhesive layer (BCB) is deposited on the target substrate. By applying both pressure and heat, the 2D material, together with the substrate (e.g. Cu foil) where it has been grown are bonded to the target sample. The growth substrate is chemically removed, leaving the 2D material onto the CBC layer deposited on the target substrate. The process can be repeated to form stacks consisting of two or more 2D materials.

Whereas the electronic and structural properties of CVD-grown graphene are virtually indistinguishable from their mechanically exfoliated counterpart, a wafer-scale transfer process is still missing and is the major bottleneck for the integration of graphene and 2D materials onto semiconductor (back end of line) production. Therefore, the topic of the paper is highly relevant and timely.

The manuscript is well organized and properly communicated, however -in the view of this reviewer- not suitable for publication in Nature Communication in the present form due to the following reasons:

Our response:

We thank the reviewer for this accurate summary, in particular the positive assessment of the timeliness and relevance of our work, its organization and the communication of the manuscript. We also thank the reviewer for the constructive criticism, which we have addressed thoroughly by the following responses to the individual comments.

Reviewer comment #1

1. Graphene transfer: it is stated (lines 105-106) that “the adhesive layer molds into the surface topography of the growth substrate”. If on one hand this is beneficial due to minimization of strain and wrinkling, on the other foils have a topography which is rarely flat. If the macroscopic surface roughness is “locked” into the adhesive tape it could affect not only the performance of the graphene but also and more significantly planar processing such a lithography. Authors should comment.

Our response:

We thank the reviewer for this valuable comment. We fully agree that the substrate topography is an essential property for device fabrication. In our method, the topography of the transferred 2D material on the target wafer is determined by the topography of the growth substrate.

The roughness of conventional Cu foils is about 100 – 250 nm root mean square (RMS). On graphene and hBN samples, we measured 176 nm and 264 nm RMS, respectively (Figure S3). However, additional smoothing before growth can reduce the surface roughness of Cu foils drastically¹. Alternative growth substrates such as Cu thin films on sapphire wafers or other epitaxial, single crystal metal films have dramatically reduced surface roughness down to the nanometer range. For other 2D materials that are grown on oxidized silicon wafers, the surface roughness is below ~5 nm. For example, for our MoS₂ samples, we measured 0.8 nm and 1.9 nm RMS before and after the transfer, respectively, thus demonstrating that our approach is generally applicable independent of the surface roughness of the donor substrate. However, if the topography of the 2D material surface on the target substrate is similar to the depth of focus (DoF) in a projection optical lithography process, this may limit the minimum feature size of patterns that can be reproduced on the target wafer. For typical i-line projection lithography (365 nm wavelength), the depth of focus is approximately 1 μm. For any of the growth substrates discussed in our manuscript, the roughness is well below this value; hence we do not expect any adverse effect in resolution for this type of lithography. For more modern excimer laser systems, the DoF decreases to 220 nm at 157 nm wavelength². Consequently, growth substrates with low surface roughness (e.g., epitaxial metal or epitaxial crystalline films on SiO₂ on Si) will be preferred for these types of very high-resolution lithography.

In the revised manuscript, we included a discussion of the surface roughness of transferred MoS₂. The related paragraphs read as follows:

(Wafer-level transfer of graphene and graphene/hBN heterostructures)

[...] Additionally, large topography of the transferred materials may potentially cause problems for very high-resolution lithography. Using growth substrates with lower surface topography may mitigate this issue^{3,4}. [...]

(Transfer of MoS₂)

[...] The surface roughness of the transferred MoS₂ (Rms: 1.9 nm) remained low compared to the original surface roughness of the MoS₂ on the growth substrate (Rms: 0.8 nm). [...]

Reviewer comment #2

2. *Graphene mobility: mobility and sheet resistance are very similar to conventional wet-transfer methods, which have same or even better scalability without requiring a bonding process and the associated equipment. Therefore, it is difficult to see the advantage of the method. Also, figure 2(a) shows a much larger number of devices compared to the 18 (graphene) + 9 (graphene/hBN) devices measured. It would have been useful to have a larger statistic as well as an idea of the scattering of mobility across the wafer surface.*

Our response:

We thank the reviewer for this comment. We agree that the material properties of our transferred graphene are similar to the material properties reported by other groups using various methods. This similarity is a strong indication that our transfer method can compete on achievable mobility and sheet resistances with those techniques. For this it is important to note that besides the transfer methods, the type and initial quality of the graphene before the transfer is also an important factor for the material properties obtained after transfer.

We would like to elaborate on the advantage of the proposed method in response to the statement “[...] *Therefore, it is difficult to see the advantage of the method.*”, with which we respectfully disagree. With our approach, we are targeting scalable 2D material integration in semiconductor and MEMS/NEMS manufacturing environments. The wafer bonding tool we use in our approach is a mature technology that is widely available and used in semiconductor foundries for substrate bonding in the manufacturing of extremely high volume products such as backside-illuminated (BSI) CMOS imaging sensors and MEMS accelerometers, with manufacturing volumes over several 100 million devices per year⁵. We believe that the availability of existing toolsets in the semiconductor manufacturing and supply chain is crucial for the price- and risk-sensitive semiconductor industry to incorporate an integration approach for 2D materials. While the reported methods for wet transfer of 2D materials may potentially be scalable to larger substrates, they certainly require unconventional or novel tools and methods which are not established in today’s semiconductor manufacturing facilities. In contrast, our transfer method relies solely on technologies and tools that are already available and used in the existing semiconductor infrastructure. We see this as an extremely strong or even decisive advantage of our approach over other methods.

Regarding Figure 2, it is correct that we measured a different amount of vdP devices with graphene and graphene/hBN. The growth substrates of hBN were smaller than the growth substrates of graphene (2.5 by 2.5 cm² and 100 mm diameter, respectively), as hBN growth lags behind graphene growth in maturity and scale. Therefore, the graphene/hBN heterostructure was only formed in the center of the wafer. Hence, there were less graphene/hBN vdP devices on the wafer. Nevertheless, to compare the extracted material properties of graphene devices with graphene/hBN devices, we measured a similar number of both types of devices and state the relative frequency. We emphasize that we report the results of ***all measured devices***, without excluding any defect or low-performance devices. The uniformity of the graphene on a similar sample is quantified in the map of the sheet resistance derived from THz spectroscopy (Figure 2c). This measurement has a resolution of ~500 μm and indicates a uniform coverage on the entire wafer, which is supported by a large statistic.

In the revised manuscript, we extended the discussion of the extracted material parameters, further clarified the unique advantages of our proposed transfer method, and state the resolution of the THz spectroscopy. The related paragraphs read as follows:

(Introduction)

[...] Here, we present a novel and versatile approach for the transfer and stacking of 2D materials to heterostructures by adhesive bonding with bisbenzocyclobutene (BCB) and commercial wafer bonding equipment. BCB is a hydroxyl-free dielectric material that was developed for the semiconductor industry and is being used as an interlayer dielectric, for heterogeneous system integration, and as an organic gate dielectric layer since it has low fixed-charge and trap densities⁶⁻¹⁰. Our approach avoids manual handling of the 2D material and does not require sacrificial carrier layers that may contaminate the surface of the 2D material. The proposed method only utilizes equipment, processes, and materials that are readily available in large-scale semiconductor production lines, which are crucial advantages for the integration into the semiconductor ecosystem^{11,12}. [...]

(Wafer-level transfer of graphene and graphene/hBN heterostructures)

[...] Four-point probe measurements of a total of 27 vdP devices yielded the graphene sheet resistance (R_{sh}), carrier density (n) and mobility of majority charge carrier (μ) of the transferred graphene layer, both for graphene resting on BCB and for graphene resting on hBN (Figure 2b) (see Methods). In both cases, all measured devices were functioning and we found a similar R_{sh} ($630 \pm 90 \Omega/sq.$ and $600 \pm 90 \Omega/sq.$, respectively) whereas the n and μ showed differences for graphene resting on BCB and on hBN, respectively. The carrier density in the graphene on BCB was lower than for graphene on hBN ($(3.7 \pm 0.7) \cdot 10^{12} \text{ cm}^{-2}$ and $(4.9 \pm 0.2) \cdot 10^{12} \text{ cm}^{-2}$), while the mobility was higher for graphene on BCB than for the graphene on hBN ($2800 \pm 100 \text{ cm}^2\text{V}^{-1}\text{s}^{-1}$ and $2000 \pm 200 \text{ cm}^2\text{V}^{-1}\text{s}^{-1}$). For the transferred graphene monolayer resting on BCB, the extracted mobilities are in the higher range of previously reported values for polycrystalline CVD graphene transferred on large areas from polycrystalline copper substrates by various methods, including wet transfer with polymeric carrier layers^{13,14}, face-to-face transfer by bubble formation¹⁵ and lamination¹⁶. The mobility in our graphene/hBN heterostructure is similar to values reported by Pandey et al. and Shautsova et al., who encapsulated graphene in CVD hBN using wet and dry transfers^{17,18}. In accordance with these works, our results indicate that current large-area CVD hBN substrates typically do not increase the mobility in polycrystalline graphene, thus contrasting several studies on single crystals of graphene and hBN¹⁹⁻²². We emphasize that for a meaningful comparison, the initial graphene quality before the transfer is of crucial importance. [...]

[...] For our measurements, we set the spatial resolution to 500 μm by adjusting the distance of the near-field detector to the sample and the pitch between acquisition sites. [...] The map of the graphene sheet resistance on a 100 mm diameter wafer after transfer and formation of the graphene/hBN heterostructure indicates a uniform coverage and successful transfer on the entire wafer (Figure 2c). The sheet resistance in the marked region averages to $450 \pm 50 \Omega/sq.$, and is similar for both underlying materials ($450 \pm 20 \Omega/sq.$ and $440 \pm 20 \Omega/sq.$ for BCB and hBN, respectively), which is in line with the measurements of the vdP devices shown in Figure 2b. These values are similar to the values reported for polycrystalline graphene that was transferred from

polycrystalline copper substrates using various methods^{13,16,17}. Rahimi et al. reported a higher mean value of 2600 Ω /sq. which originated from lower residual doping¹⁴. [...]

Reviewer comment #3

3. Heterostructures: interface is the most critical part of any heterostructure. The manuscript neglects this aspect entirely, thus not providing any solid proof that the method is suitable for fabrication of heterostructures. Also, the poor mobility of graphene on hBN (worse than on the CBC film) raises additional concerns about the applicability of the method to form heterostructures. The only conclusion that one can take from this results are that either the hBN used is extremely poor or that the method -when applied to stack 2D materials- degrades their quality significantly.

Our response:

We thank the reviewer for pointing out this important aspect. We agree that the interface between stacked layers is crucial for the properties of heterostructures.

It is correct that we measured a lower mobility in graphene on hBN than on BCB. We discuss this discrepancy in the revised manuscript and hypothesize that the differences in surface topography of the two growth substrates induces strain variations in the graphene transferred on top of the hBN which, in turn, may contribute to the lower charge carrier mobility.

Since the THz spectroscopy map of the sheet resistance of graphene on hBN and graphene on BCB (Figure 2c) shows a uniform coverage and similar sheet resistance in both regions, we believe that the poor quality of the hBN inhibits higher charge carrier mobility. We agree that this behavior is in clear contrast to observations on single crystals of graphene and hBN¹⁹⁻²². However, our results are in line with literature which shows that typical substrates of polycrystalline large-area CVD hBN that are available today do not increase the mobility in polycrystalline graphene^{17,18}. This situation can be expected to change if (or when) high quality hBN films become available but is not a general sign of shortcomings of the proposed wafer bonding technology. Generally, the issue of high topography may be mitigated when transferring and stacking layers from growth substrates with lower surface roughness (e.g. MoS₂ from Si/SiO₂ chips, graphene from Cu thin-films on sapphire, or graphene grown on single-crystalline metals).

We have identified residuals of copper and the copper etchant as potential contamination at the interface between layers. This is a general issue of graphene technology, although both may be reduced by improving the etching and cleaning procedures, as the reviewer correctly points out in comment #5. Both are studied intensely in the literature and may be combined with our transfer method. In addition, it should be noted that since we transfer the second layer in vacuum and at elevated temperature, the amount of water and gas molecules trapped between the layers may be lower than e.g. by two consecutive wet transfers.

In the revised manuscript, we further emphasize the discussion of the importance of the interface between layers and potential sources of contamination with our method. The related paragraphs read as follows:

(Introduction)

[...] Vertical stacking of 2D materials forms van der Waals heterostructures, a novel class of materials with new material properties that result from synergetic effects between the stacked layers at their interfaces²³⁻²⁸. [...]

(Wafer-level transfer of graphene and graphene/hBN heterostructures)

[...] Note that in the heterostructure, neither the graphene nor the top surface of the hBN was in contact with polymers or adhesives at any time. Moreover, the second transfer was performed in a vacuum and at elevated temperature, which reduces the amount of water and gas molecules that may get trapped between the layers. Residuals of copper and the copper etchant are still potential contaminants at the interlayer interface, which potentially can affect the properties of the heterostructure²⁹. [...]

Reviewer comment #4

Also, some of the statements in the manuscript are not correct, as follows:

- Lines 26- 27: "current transfer methods cause degradation of the 2D materials" multiple studies have shown that the quality of 2D materials is fully "restored" if a suitable cleaning process is applied. For example, Purdie et al. showed (Nature Comm. 9, 5387) that graphene intentionally contaminated with solvent and polymers are indistinguishable from pristine sample if a suitable cleaning process is applied.

Our response:

We thank the reviewer for this comment. We agree that there are cleaning procedures which restore the material properties of contaminated 2D materials. For example, Purdie et al. present a procedure to remove contaminants (blisters) from the interface of exfoliated materials in heterostructures assembled by a PDMS/PC stamp³⁰. The appropriate choice of the stamp material, the angle of the stamp relative to the target surface, stamping speed ($> 10 \mu\text{m s}^{-1}$) and temperature, mechanically manipulates the location of blisters in areas with edge lengths of about tens to 100 μm . The removal of PMMA residuals from single layer graphene encapsulated in hBN by this technique yields material properties which are similar to heterostructures which were not exposed to PMMA. Despite this progress, which we have now referenced in our manuscript, we would argue that the degradation of 2D materials due to transfer is not generally solved in all aspects.

In the revised manuscript, we corrected our statement in the abstract and included a statement on cleaning procedure in the introduction. The related paragraphs read as follows:

(Abstract)

[...] Current transfer methods often degrade material properties and are not compatible with industrial processing for large-scale manufacturing on wafer-level. [...]

(Introduction)

[...] On the micrometer scale, cleaning techniques can mechanically manipulate the location of encapsulated contaminants and restore intrinsic material properties³⁰. [...]

Reviewer comment #5

- Lines 28-29: “here, we report a generic methodology for dry-transfer and large-area integration of 2D materials and their heterostructures by adhesive wafer bonding”

The transfer method described cannot be considered dry and such statement is misleading. Indeed, similar to other transfer methods, the growth substrate is removed by wet etching (FeCl₃ in the case graphene and hBN, KOH for MoS₂). For a review of wet and dry transfer, see for example De Fazio et al. 13, 8926

Our response:

We thank the reviewer for this comment and for directing us to the review paper. De Fazio et al. indeed classify a transfer as “wet” if any surface of the 2D material is in contact with any liquids, etchants, or supporting layers during the process³¹. “Dry” transfers rely on peeling without any chemical etching. Following this definition, we agree that our transfer method is not a dry transfer process as our transferred 2D materials was in contact with liquids during the removal of the growth substrates. Our aim with the initially chosen terminology was to distinguish our method from conventional wet transfers with sacrificial layers and scooping of floating stacks of polymer-coated 2D materials from liquid surfaces. In response to this comment we have changed the terminology in the manuscript.

In the revised manuscript, we abstain from the term “dry transfer” to describe our integration approach and reference the suggested review. The revised paragraphs read as follows:

(Abstract)

[...] Here, we report a generic methodology for transfer and large-area integration of 2D materials and their heterostructures by adhesive wafer bonding. [...]

(Introduction)

[...] Commonly used wet transfer approaches rely on an intermediate polymeric carrier, typically poly(methyl methacrylate) (PMMA)^{31–33} or polycarbonate (PC)^{34,35}, which mechanically supports the 2D material during its removal from the growth substrate and transfer to the target substrate by scooping from the surface of a liquid. [...]

[...] Here, we present a novel and versatile approach for the transfer and stacking of 2D materials to heterostructures by adhesive bonding with bisbenzocyclobutene (BCB) and commercial wafer bonding equipment. BCB is a hydroxyl-free dielectric material that was developed for the semiconductor industry and is being used as an interlayer dielectric, for heterogeneous system integration, and as an organic gate dielectric layer since it has low fixed-charge and trap densities^{6–10}. Our approach avoids manual handling of the 2D material and does not require sacrificial carrier layers that may contaminate the surface of the 2D material. The proposed method only utilizes equipment, processes, and materials that are readily available in large-scale semiconductor

production lines, which are crucial advantages for the integration into the semiconductor ecosystem^{11,12}. [...]

(Conclusion)

[...] We have demonstrated a generic methodology for transfer and large-area integration of 2D materials and their heterostructures on wafer-level, which is compatible with conventional semiconductor fabrication lines. [...]

Reviewer comment #6

- Lines 298-299 *"We also fabricated high mobility field-effect graphene devices" The mobility achieved is very modest and the statement above is entirely misleading.*

Our response:

We agree with the reviewer that this comparative statement might have been misleading without putting our results into perspective. **In the revised manuscript**, we have reformulated our statement. The revised paragraph reads as follows:

(Conclusion)

[...] We fabricated top-gated field-effect graphene devices ($\mu = 2630 \text{ cm}^2\text{V}^{-1}\text{s}^{-1}$), which required multiple lithography steps in combination with conventional deposition and etching processes after the transfer. [...]

Reviewer comment #7

- Lines 314-315: *"Consequently, the barrier is low for industry to incorporate this methodology into their manufacturing lines as a standard for back end of the line integration"*

Incorporating graphene and 2D materials into semiconductor production lines is certainly not easy and the statement above is misleading. It would be correct to say that -if the method was providing good results, which unfortunately is not the case- it should be considered by industry as it is based on processed already available in the industry.

Our response:

We thank the reviewer for pointing this out. We agree that the integration of 2D materials into semiconductor production lines is a challenging task. One of the benefits of our methodology is that it uses conventional processes, equipment, and materials. Besides the necessity of high material quality, we see these factors as decisive when investigating integration approaches for industrial manufacturing (see also our response to Reviewer comment #2 above). We would also like to point out that achieving very high mobility in graphene may not be required by all potential applications, although it would obviously be a good indicator for high quality process technology.

In the revised manuscript, we have reformulated our statement in line with the suggestion. The related paragraph reads as follows:

(Conclusion)

[...] Consequently, the barrier is comparatively low for the industry to consider this methodology for the integration of 2D materials on top of conventional integrated circuit and microsystem device substrates in the back end of the line (BEOL). [...]

Reviewer comment #8

In summary, despite the approach is very interesting and provides a possible route towards wafer-scale integration of graphene and other 2D materials into semiconductor production lines, the results are very modest and similar to conventional wet-transfer methods. The industry requirements are significantly higher, e.g. carrier mobility >10,000 cm²/Vs at carrier density ~ 10¹² are essential for (opto)electronic devices (see for example Romagnoli et al, Nature Review Materials 3, 392).

I would strongly encourage working on the optimization of the method which -if results are significantly improved- can have a very strong impact on graphene and 2D materials technology. Unfortunately, in the view of this reviewer, the current results are very far from creating any significant improvement over conventional methods used for graphene transfer.

Our response:

We thank the reviewer for the positive assessment of the general interest in and timeliness of our approach and for the comments on our results. We do, however, not agree with some of the criticism.

It is correct that higher material quality is desired to leverage the full potential of 2D materials for industrial applications. The required specifications cited by the reviewer in the respective reference are indeed one such scenario, but by far not the only one. While it would be desirable to meet these targets today with a manufacturable method, such method is not available and instead, our paper discusses a promising route to follow in order to achieve this (or other) specification(s). In addition, the material quality as measured in a device is a combination of potential material degradation during transfer and the initial material quality. As we did not have ultra-high-quality material available for our experiments, we used widely available large-area standard-grade 2D materials grown by CVD. Considering the initial quality of 2D materials in our experiments, we believe that our results compare well to results achieved by other large-area transfer methods in published work. It is correct that the demonstrated material quality in our study does not outperform the material quality in studies using small flakes of single-crystalline graphene encapsulated in exfoliated, high-quality hBN. This is expected and does not diminish the relevance of our findings in any way. In our view, the clear benefit of our method over other published large-area transfer approaches is that it utilizes well-established commercial wafer bonding technology that is widely available in semiconductor foundries for the manufacturing of extremely high-volume products. Our method is versatile and can be extended to many different types of donor substrates and high-quality 2D materials.

To clarify these points in the revised manuscript, we have extended the discussion of the extracted material parameters, further clarified the unique advantages of our transfer method. The related paragraphs read as follows:

(Introduction)

[...] Most importantly, neither of the reported transfer methods is compatible with industry routines for large-scale manufacturing while still preserving the high quality of the 2D material as on the growth substrate, which is required for many applications³⁶. [...]

[...] Here, we present a novel and versatile approach for the transfer and stacking of 2D materials to heterostructures by adhesive bonding with bisbenzocyclobutene (BCB) and commercial wafer bonding equipment. BCB is a hydroxyl-free dielectric material that was developed for the semiconductor industry and is being used as an interlayer dielectric, for heterogeneous system integration, and as an organic gate dielectric layer since it has low fixed-charge and trap densities⁶⁻¹⁰. Our approach avoids manual handling of the 2D material and does not require sacrificial carrier layers that may contaminate the surface of the 2D material. The proposed method only utilizes equipment, processes, and materials that are readily available in large-scale semiconductor production lines, which are crucial advantages for the integration into the semiconductor ecosystem^{11,12}. [...]

(Wafer-level transfer of graphene and graphene/hBN heterostructures)

[...] Four-point probe measurements of a total of 27 vdP devices yielded the graphene sheet resistance (R_{sh}), carrier density (n) and mobility of majority charge carrier (μ) of the transferred graphene layer, both for graphene resting on BCB and for graphene resting on hBN (Figure 2b) (see Methods). In both cases, all measured devices were functioning and we found a similar R_{sh} ($630 \pm 90 \Omega/sq.$ and $600 \pm 90 \Omega/sq.$, respectively) whereas the n and μ showed differences for graphene resting on BCB and on hBN, respectively. The carrier density in the graphene on BCB was lower than for graphene on hBN ($(3.7 \pm 0.7) \cdot 10^{12} \text{ cm}^{-2}$ and $(4.9 \pm 0.2) \cdot 10^{12} \text{ cm}^{-2}$), while the mobility was higher for graphene on BCB than for the graphene on hBN ($2800 \pm 100 \text{ cm}^2\text{V}^{-1}\text{s}^{-1}$ and $2000 \pm 200 \text{ cm}^2\text{V}^{-1}\text{s}^{-1}$). For the transferred graphene monolayer resting on BCB, the extracted mobilities are in the higher range of previously reported values for polycrystalline CVD graphene transferred on large areas from polycrystalline copper substrates by various methods, including wet transfer with polymeric carrier layers^{13,14}, face-to-face transfer by bubble formation¹⁵ and lamination¹⁶. The mobility in our graphene/hBN heterostructure is similar to values reported by Pandey et al. and Shautsova et al., who encapsulated graphene in CVD hBN using wet and dry transfers^{17,18}. In accordance with these works, our results indicate that current large-area CVD hBN substrates typically do not increase the mobility in polycrystalline graphene, thus contrasting several studies on single crystals of graphene and hBN¹⁹⁻²². We emphasize that for a meaningful comparison, the initial graphene quality before the transfer is of crucial importance. [...]

(Conclusion)

[...] Consequently, the barrier is comparatively low for the industry to consider this methodology for the integration of 2D materials on top of conventional integrated circuit and microsystem device substrates in the back end of the line (BEOL). [...]

Reviewer #2 (Remarks to the Author):

The submitted work describes a method able to realize wafer level transfer and wafer level fabrication of two dimensional materials and heterostructures as well as devices thereon. The potential of the method is demonstrated using a 100 mm carrier wafer. Large area transfer of graphene/BCB, graphene/hBN, graphene/graphene and MoS₂/BCB was realised. Reasonable good material and devices properties are demonstrated. The drawback of the method is the large waviness of the surface after transfer. Nevertheless, due to the large potential of the method demonstrated, the paper is worth to be published in Nature Communications, if the authors will take into account the below listed remarks and improve the compelling of the results presented:

Our response:

We would like to thank the referee for the positive assessment of our work.

Reviewer comment #1

(1) Abstract line 31: Misleading statement. "We transferred monolayers of graphene and molybdenum disulphide (MoS₂) from their growth substrates to 100 mm diameter silicon wafers and fabricated field-effect graphene devices with high carrier mobility up to 2630 cm²/(Vs)." Firstly, it is unclear what dimensions the transferred graphene and MoS₂ had was it 4 inch or 100 x 100 nm². Secondly, please describe the type of graphene filed effect transistor in a separate sentence to avoid misleading conclusions.

Our response:

We thank the reviewer for this comment, which helped us to further improve the abstract and clarify the related text. We agree that the description of our experimental work in the abstract should have been stated more precisely.

In the revised manuscript, we stated the sample sizes and added information on the field-effect devices. The related paragraph reads as follows:

(Abstract)

[...] To demonstrate the utility of our approach, we transferred polycrystalline graphene monolayers with diameters of 100 mm from copper foils to 100 mm diameter silicon wafers, and centimeter-sized molybdenum disulfide (MoS₂) layers from SiO₂/Si growth substrates to 100 mm diameter silicon wafers. Additionally, we fabricated encapsulated field-effect graphene devices with an Al₂O₃ top-gate dielectric, featuring high carrier mobilities of up to 2630 cm²V⁻¹s⁻¹. [...]

Reviewer comment #2

(2) Introduction: The introduction (first part) do not highlight the real advantages of 2D materials, give a comparison of Si properties and 2D properties when it is scaled to 10 nm.

Our response:

We thank the reviewer for this comment. We agree that we did not mention the specific benefits of 2D materials for transistors in the introduction. Since this is an important application, we included a comparison between silicon and TMD transistors.

In the revised manuscript, we added a description of the scaling limits for silicon transistors below 10 nm gate length and compare it with the expected performance of MoS₂ transistors. The related paragraph reads as follows:

(Introduction)

[...] Transition metal dichalcogenides are often semiconducting and promise advancements for high- and low-power transistors³⁷, infrared photodetectors³⁸ and emerging device concepts such as memristors³⁹⁻⁴¹ and single-photon emitters for optical quantum communication⁴²⁻⁴⁴. The scaling limit of transistors is set by short channel effects that arise from intrinsic semiconductor properties. These effects degrade the off-state leakage current and are the limiting factor for silicon transistors with sub-10 nm gate length. On this scale, MoS₂ transistors are expected to achieve more than two orders of magnitude lower leakage current than silicon transistors⁴⁵. Furthermore, the carrier mobility in MoS₂ transistors degrades less with decreasing channel thickness than in silicon transistors^{46,47}. Hence, 2D semiconductors may be the ultimate channel material for ultra-scaled MOSFETs⁴⁸. [...]

Reviewer comment #3

(3) The understanding of the "Methodology ..." chapter is not easy.

Our response:

We thank the reviewer for this comment. We agree that parts of the method section were difficult to comprehend.

In the revised manuscript, we clarified the description of our method to improve the readability. The revised text passages read as follows:

(Methodology for wafer-level transfer of 2D materials and their heterostructures)

Our method for transferring 2D materials from their growth substrate to a target wafer comprises four consecutive steps (Figure 1): First, the target wafer is spin-coated with an adhesive layer of thermosetting bisbenzocyclobutene (BCB). A soft-bake removes solvents and solidifies the adhesive layer (Figure 1a, (1)). Next, the 2D material on its growth substrate is placed on top of the target wafer such that the 2D material is facing the adhesive layer (Figure 1a, (2)). The stack is then loaded in a commercial wafer bonder (Figure 1a, (3)). Inside the tool, heating temporarily decreases the viscosity of the adhesive layer while the bond chuck applies a uniform force to the wafer stack. Thus, the adhesive layer molds against the 2D material and forms a stable bond to the target wafer while replicating the surface topography of the growth substrate without exerting excessive pressure on the 2D material. This characteristic is beneficial, since it minimizes potential damage, wrinkles, or strain in the transferred 2D material. After wafer bonding, the growth substrate is removed (Figure 1a, (4)) by either etching, delamination, or permeation of liquids into the interface between the 2D material and the growth substrate, which leaves the 2D material transferred on the target wafer (Figure 1a, (I)). Since BCB is a thermosetting polymer, heating partially cross-links the polymer chains

in the adhesive layer, which form a network with high chemical stability. If the degree of cross-linking is kept low, by appropriate choice of bonding temperature and time, the BCB adhesive layer allows repeated molding. Hence, reusing the same target wafer for another transfer assembles heterostructures by vertical stacking of 2D materials – without coating any additional adhesive, simply by reusing the existing adhesive layer below the previously transferred 2D material (Figure 1b, (1-4) and (II)). Thus, van der Waals heterostructures form by repeating the three steps of material placement (Figure 1b, (2)), wafer bonding (Figure 1b, (3)), and removal of the growth substrate (Figure 1b, (4)). We note that in this process neither the interface between the first and second 2D material nor the surface of the second 2D material are exposed to any polymer carrier or adhesive that may potentially degrade the 2D material properties by contamination. Partially cross-linked BCB is chemically stable and sustains conventional wafer processing. This feature enables not only structuring of the 2D material to devices after the transfer (using the wafers (I) and (II) in Figure 1a and 1b, respectively), but also the integration of metal contacts on top of the BCB layer before transfer (Figure S1). These contacts could be used to interface integrated circuits (ICs) that may be embedded on the target wafer. Moreover, membranes of 2D materials can be suspended over cavities, which were etched into the adhesive layer before transfer (Figure S2). In both cases, the 2D material in the contact and membrane area is not in contact with the adhesive at any time (see Supplementary Information for demonstration).

Reviewer comment #4

(4) Figure 1: The Figure is designed in 3D style. It would be better to use the common 2D used in flow charts for semiconductor processing. When this is done in a proper way the process flow would be easier to understand.

Our response:

We thank the reviewer for this suggestion. We agree with the reviewer that 2D flow-charts generally are a suitable way to depict process flows. However, in the present case we believe that our 3D graphical approach communicates better, at a glance, the capability of our method for full wafer processing, which is a key feature of our work. We deem that this important aspect might be lost in a cross-sectional 2D flow chart. To further improve the presentation of the present Figure, we have further refined the illustration of the process for better clarity. However, if the journal editor feels that a 2D process illustration would be more appropriate for Figure 1, we are of course happy to change Figure 1 to a 2D illustration.

In the revised manuscript, the new figure and figure caption appears as follows:

(Methodology for wafer-level transfer of 2D materials and their heterostructures)

Figure 1: Schematics illustration of the methodology for wafer-level transfer of 2D materials (a) and formation of 2D material heterostructures (a and b).

a) (1) Spin-coating and soft-baking of thermosetting bisbenzocyclobutene (BCB) as an adhesive layer on the target wafer. **(2)** Placement of the 2D material on its growth substrate on top of the target wafer, with the 2D material facing the target wafer. **(3)** Adhesive wafer bonding by applying heat and force to the wafer stack using a commercial semiconductor wafer bonding tool, thereby forming a stable bond between the 2D material on its growth substrate and the target wafer. **(4)** Removal of the growth substrate. **(II)** Transferred 2D material on the target wafer. **b) (1)** Reusing the target wafer from step (I) without additional treatment. **(2)** Placement of a second 2D material on its growth substrate on top of the target wafer, with the 2D material facing the previously transferred 2D material. **(3)** Wafer bonding as in a) step (3). **(4)** Removing the growth substrate of the second 2D material. **(II)** Transferred 2D material heterostructure on the target wafer.

Reviewer comment #5

(5) Line 81 to line 82: "All transferred layers and heterostructures were of high quality" The property of the quality is not specified!

Our response:

We thank the reviewer for this comment which helped us to express this statement more precisely.

In the revised manuscript, we added the measured material properties and the related section reads as follows:

(Introduction)

[...] All transferred layers and heterostructures were of high quality, i.e. they featured uniform coverage and little strain in the transferred 2D materials. These findings indicate that our proposed integration approach preserves similar mechanical properties of the 2D materials as present on the growth substrate while minimizing degradation of the transferred layers through introduction of wrinkles or excessive strain. [...]

Reviewer comment #6

(6) Line 110: "..., the chemical stability of the ... wafer processing." Is doubled in line 119. Please reformulate.

Our response:

We thank the reviewer for this comment and agree that the wording in this section was repetitive.

In the revised manuscript, we removed redundancy in the description of our method. The related section reads as follows:

(Methodology for wafer-level transfer of 2D materials and their heterostructures)

[...] Since BCB is a thermosetting polymer, heating partially cross-links the polymer chains in the adhesive layer, which form a network with high chemical stability. If the degree of cross-linking is kept low, by appropriate choice of bonding temperature and time, the BCB adhesive layer allows repeated molding. Hence, reusing the same target wafer for another transfer assembles heterostructures by vertical stacking of 2D materials – without coating any additional adhesive, simply by reusing the existing adhesive layer below the previously transferred 2D material (Figure 1b, (1-4) and (II)). Thus, van der Waals heterostructures form by repeating the three steps of material placement (Figure 1b, (2)), wafer bonding (Figure 1b, (3)), and removal of the growth substrate (Figure 1b, (4)). We note that in this process neither the interface between the first and second 2D material nor the surface of the second 2D material are exposed to any polymer carrier or adhesive that may potentially degrade the 2D material properties by contamination. Partially cross-linked BCB is chemically stable and sustains conventional wafer processing. This feature enables not only structuring of the 2D material to devices after the transfer (using the wafers (I) and (II) in Figure 1a and 1b, respectively), but also the integration of metal contacts on top of the BCB layer before transfer (Figure S1). These contacts could be used to interface integrated circuits (ICs) that may be embedded on the target wafer. Moreover, membranes of 2D materials can be suspended over cavities, which were etched into the adhesive layer before transfer (Figure S2). In both cases, the 2D material in the contact and membrane area is not in contact with the adhesive at any time (see Supplementary Information for demonstration). [...]

Reviewer comment #7

(7) The authors carry out a non-proper discussion of their sheet carrier resistance. The demonstrated values are not so exciting.

Our response:

We thank the reviewer for pointing us to the insufficient discussion of the sheet resistance. We agree that our results from the van-der-Pauw and THz measurements should be contextualized. We would like to stress that we cannot expect to achieve record values with our proposed method, as we have explained in more detail in our reply to reviewer #1 (comment #2) and reviewer #3 (comment #2).

In the revised manuscript, we added a comparison of the sheet resistance with values reported in literature. The related paragraph reads as follows:

(Wafer-level transfer of graphene and graphene/hBN heterostructures)

[...] The map of the graphene sheet resistance on a 100 mm diameter wafer after transfer and formation of the graphene/hBN heterostructure indicates a uniform coverage and successful transfer on the entire wafer (Figure 2c). The sheet resistance in the marked region averages to $450 \pm 50 \Omega/\text{sq.}$, and is similar for both underlying materials ($450 \pm 20 \Omega/\text{sq.}$ and $440 \pm 20 \Omega/\text{sq.}$ for BCB and hBN, respectively), which is in line with the measurements of the vdP devices shown in Figure 2b. These values are similar to the values reported for polycrystalline graphene that was transferred from polycrystalline copper substrates using various methods^{13,16,17}. Rahimi et al. reported a higher mean value of $2600 \Omega/\text{sq.}$ which originated from lower residual doping¹⁴. [...]

Reviewer comment #8

(8) One of the main statements of the paper is that the two dimensional materials are not contaminated by polymers used in the process sequence. The authors use BCB and transfer of the two dimensional materials onto BCB. So, at least the backside is contaminated with a polymer material leading to p-type doping of graphene. What is the advantage of BCB, a material developed for low-k applications? Later on, in the graphene FET section, the authors' uses standard photolithography technology to fabricate the devices. This thwarts the idea to avoid polymer contaminations or polymer free processing. To ensure that the authors have to developed a polymer contamination free process. Please give details for photoresist removal techniques and prove the contamination free processing.

Our response:

We thank the reviewer for this comment and for the opportunity to further elaborate on these points. We agree, that for the simple case of graphene transfer onto BCB, the backside of the graphene is in direct contact with the BCB, which leads to p-type doping. This phenomenon is clearly observed in our Raman and Hall measurements.

However, when stacking layers to heterostructures, only the backside of the first material layer is in contact with the BCB. For subsequent transfers, no additional adhesive is applied between layers. Therefore, the second layer as well as the interface between the layers does not get in contact with a polymer at any time. The same holds true for two other scenarios which we demonstrate in the manuscript (see Supplementary Information): (1) Suspended membranes over cavities that were etched into the BCB before transfer. (2) Bottom contacts to metal electrodes that were structured on top of the BCB prior to transfer. In both cases, 2D material of the membrane and the contact area, respectively, is not in contact with the BCB.

The benefits of BCB as an adhesive layer are: (1) BCB has a low fixed-charge and trap density. (2) Partially cross-linked BCB bonds and reshapes in repeated transfers. This property enables the stacking of layers to heterostructures. (3) BCB is an established material in the semiconductor industry. (4) The high chemical stability of partially cured BCB allows conventional wafer processes before and after transfer to structure the adhesive, integrate contacts, pattern the transferred 2D material and fabricate devices.

We agree, that for the fabrication of the graphene field-effect devices, we use two lithographic layers which were in direct contact with the transferred graphene. These resist layers most likely contaminated the graphene by polymer residuals to some degree. However, it should be noted that our objective in realizing these devices was to demonstrate that the transferred 2D materials are compatible with continued semiconductor fabrication technologies such as lithography, etching and material deposition. In these experiments we did not aim at evaluating or demonstrating contamination-free graphene. The order of the evaluated semiconductor fabrication processes may be altered to prevent the 2D material from being exposed to the photoresist in the lithography process. For example, in an improved fabrication scheme, the passivating Al₂O₃ layer could be deposited in a first process step which could then serve as a hard mask during patterning of the graphene. In combination with bottom contacts (deposited before the graphene transfer) the graphene would not be in contact with either of the lithographic layers. Alternatively, fabrication schemes involving laser patterning could be used to avoid lithography with a photoresist during device fabrication [refs for litho-free graphene device fabrication?].

Inspired by this comment, **in the revised manuscript** we have clarified the related text passages to remove ambiguity about the polymer contamination, highlighted the benefits of BCB and added the process details of the resist removal. The related paragraphs read as follows:

(Introduction)

[...] Here, we present a novel and versatile approach for the transfer and stacking of 2D materials to heterostructures by adhesive bonding with bisbenzocyclobutene (BCB) and commercial wafer bonding equipment. BCB is a hydroxyl-free dielectric material that was developed for the semiconductor industry and is being used as an interlayer dielectric, for heterogeneous system integration, and as an organic gate dielectric layer since it has low fixed-charge and trap densities⁶⁻¹⁰. Our approach avoids manual handling of the 2D material and does not require sacrificial carrier layers that may contaminate the surface of the 2D material. The proposed method only utilizes equipment, processes, and materials that are readily available in large-scale semiconductor production lines, which are crucial advantages for the integration into the semiconductor ecosystem^{11,12}. We demonstrated the transfer of monolayer graphene from copper foils to 100 mm diameter silicon wafers, the transfer and stacking of multilayer hBN, and monolayer graphene to form graphene/hBN heterostructures, and the stacking of two graphene layers to form double-layer graphene. We also fabricated field-effect graphene devices to demonstrate the utility of our methodology for wafer-level device manufacturing with conventional semiconductor processes. [...]

(Methodology for wafer-level transfer of 2D materials and their heterostructures)

Our method for transferring 2D materials from their growth substrate to a target wafer comprises four consecutive steps (Figure 1): First, the target wafer is spin-coated with an adhesive layer of thermosetting bisbenzocyclobutene (BCB). A soft-bake removes solvents and solidifies the adhesive layer (Figure 1a, (1)). Next, the 2D material on its growth substrate is placed on top of the target wafer such that the 2D material is facing the adhesive layer (Figure 1a, (2)). The stack is then loaded in a commercial wafer bonder (Figure 1a, (3)). Inside the tool, heating temporarily decreases the viscosity of the adhesive layer while the bond chuck applies a uniform force to the wafer stack. Thus, the adhesive layer molds against the 2D material and forms a stable bond to the target wafer while replicating the surface topography of the growth substrate without exerting excessive pressure on the 2D material. This characteristic is beneficial, since it minimizes potential damage, wrinkles, or strain in the transferred 2D material. After wafer bonding, the growth substrate is removed (Figure 1a, (4)) by either etching, delamination, or permeation of liquids into the interface between the 2D material and the growth substrate, which leaves the 2D material transferred on the target wafer (Figure 1a, (I)). Since BCB is a thermosetting polymer, heating partially cross-links the polymer chains in the adhesive layer, which form a network with high chemical stability. If the degree of cross-linking is kept low, by appropriate choice of bonding temperature and time, the BCB adhesive layer allows repeated molding. Hence, reusing the same target wafer for another transfer assembles heterostructures by vertical stacking of 2D materials – without coating any additional adhesive, simply by reusing the existing adhesive layer below the previously transferred 2D material (Figure 1b, (1-4) and (II)). Thus, van der Waals heterostructures form by repeating the three steps of material placement (Figure 1b, (2)), wafer bonding (Figure 1b, (3)), and removal of the growth substrate (Figure 1b, (4)). We note that in this process neither the interface between the first and second 2D material nor the surface of the second 2D material are exposed to any polymer carrier or adhesive that may potentially degrade the 2D material properties by contamination. Partially cross-linked BCB is chemically stable and sustains conventional wafer processing. This feature enables not only structuring of the 2D material to devices after the transfer (using the wafers (I) and (II) in Figure 1a and 1b, respectively), but also the integration of metal contacts on top of the BCB layer before transfer (Figure S1). These contacts could be used to interface integrated circuits (ICs) that may be embedded on the target wafer. Moreover, membranes of 2D materials can be suspended over cavities, which were etched into the adhesive layer before transfer (Figure S2). In both cases, the 2D material in the contact and membrane area is not in contact with the adhesive at any time (see Supplementary Information for demonstration).

(Conclusion)

[...] The ability of the adhesive layer to mold repeatedly allows the stacking of 2D materials to heterostructures. In this case, merely the bottommost 2D material is in contact with the adhesive layer. This ensures that the other 2D materials and the interfaces between layers are not contaminated by polymer residuals. To demonstrate this capability, we fabricated graphene/hBN heterostructures and double-layer graphene on large-areas by consecutive transfers. [...]

(Methods)

Fabrication and evaluation of top-gated field-effect devices

Graphene based top-gated field-effect devices were fabricated on a 100 mm silicon wafer. First, large-area graphene (10 cm by 10 cm) was transferred by adhesive wafer bonding (see above). Contacts to graphene were fabricated by sputter deposition of 25 nm Ni and e-beam evaporation of 135 nm Al, followed by a lift-off process in acetone (60 °C, 30 min) (1st photolithography layer). The graphene sheet was then patterned by etching in oxygen plasma (2nd photolithography layer). As a top-gate dielectric, 40 nm Al₂O₃ was deposited by atomic layer deposition and top-gate electrodes were fabricated by e-beam evaporation of 10/150 nm Ti/Al (3rd photolithography layer). To access the graphene metal contacts under Al₂O₃, reactive ion etching with SF₆ and O₂ was used to open vias through the Al₂O₃ (4th photolithography layer). All resist layers were removed by immersion in acetone (60 °C, 30 min). In this process sequence, the first and second lithographic layers were in direct contact with the graphene, which may have degraded the graphene quality. However, encapsulation of the graphene before fabricating the contacts eliminates this potential source of degradation⁴⁹. [...]

Reviewer comment #9

(9) Abstract: The authors state that they developed an industrial compatible process. For our understanding this is a questionable statement, because shadow masks and femtosecond laser ablation for structuring (line 151-152) are not very common

Our response

We thank the reviewer for this comment. Our statement in the abstract was meant to refer to the methodology of transferring 2D materials and stacking of layers to heterostructures, which was not clearly stated. We agree that shadow mask deposition and femtosecond laser ablation are not common processes in industrial manufacturing facilities. We used those processes only to fabricate evaluation devices for this paper to characterize the effect of the transfer process on the transferred layers (without potentially affecting the material properties by a lithographic layer in direct contact with the 2D material). Our methodology of transferring 2D materials and stacking of layers to heterostructures requires only conventional semiconductor manufacturing equipment (wafer bonder) and a well-established adhesive material (BCB). This should make our approach suitable for industrial manufacturing, which we also demonstrated by fabricating top-gated field effect devices using standard semiconductor processes.

In the revised manuscript, we clarified our wording to avoid ambiguity. To further improve the readability of our manuscript, we moved the details of the fabrication to the Methods section. The related paragraphs read as follows:

(Wafer-level transfer of graphene and graphene/hBN heterostructures)

[...] Next, we integrated Ti/Au electrodes and patterned the 2D material heterostructure to van-der-Pauw (vdP) devices, which allow electrical material characterization on the target wafer (Figure 2a). [...]

(Methods)

[...] Thermal evaporation of 20 nm Ti and 200 nm Au through a shadow mask formed electrodes (300 μm by 300 μm) on top of the transferred graphene layer. Ablation by a pulsed femtosecond-laser electrically insulated individual van-der-Pauw devices with an edge length of about 3 mm. These non-contact processes prevent potential material degradation by lithographic layers in contact with the graphene. [...]

Reviewer comment #10

(10) Page 6, line 146: The authors removed the copper foil with FeCl_3 and rinsed the in deionized water. What else was done to remove iron contaminations? Please list in methods section all used cleaning procedures used before and after the processing steps. Alternatively, they can be described in the body text of the publication.

Our response:

We thank the reviewer for this question. In our experiments, we did not use any cleaning step other than intense rinsing in deionized water. We acknowledge that the proper cleaning of graphene and 2D materials to obtain front end of the line manufacturing specifications regarding contamination levels remains a general challenge, independent of the transfer process.

In the revised manuscript, we added the rinsing time in the process description. The related paragraphs read as follows:

(Methods)

Transfer of graphene

[...] Etching the copper foil in FeCl_3 solution and rinsing in deionized water for 30 min uncovered the graphene, transferred to the target wafer. [...]

Formation of graphene/hBN heterostructures

[...] Etching of the copper foil in FeCl_3 and subsequent rinsing in deionized water for 30 min uncovered the hBN, transferred to the target wafer. [...] Etching of the copper foil in FeCl_3 and rinsing in deionized water for 30 min uncovered the graphene, which formed a graphene/hBN heterostructure with the previously transferred hBN.

Formation of 2-layer graphene

Etching of the copper foil in FeCl_3 solution and rinsing in deionized water for 30 min uncovered the graphene, transferred to the target wafer. [...] To form 2-layer graphene, a 100-millimeter sheet of CVD graphene on copper foil was placed on top of the target wafer and bonded at 190 $^\circ\text{C}$ for 20 min (bond force: 3 kN, bond pressure: 0.95 bar, vacuum atmosphere). Etching of the copper foil in FeCl_3 and rinsing in deionized water for 30 min uncovered the graphene, which formed 2-layer graphene with the previously transferred layer. [...]

Reviewer comment #11

(11) Please indicate the thickness of the multilayer hBN.

Our response:

We thank the reviewer for the suggestion to add this important detail.

In the revised manuscript, we noted the thickness of the hBN. The related paragraph read as follows:

(Methods)

Formation of graphene/hBN heterostructures

[...] A 2.5 cm x 2.5 cm sheet of multilayer hBN (thickness: 2-5 nm) synthesized by CVD on copper foil (2D Semiconductors Inc.) was placed on top of the target wafer (hBN facing the BCB layer) and bonded at 190 °C for 20 min at a bond force of 250 N (bond pressure: 4 bar) in nitrogen atmosphere. [...]

Reviewer comment #12

(12) Line 177-180: Tens of micrometres are not a microscopic resolution. Please delete the sentences related to this statement. And give the real spot size and resolution used for the mapping.

Our response:

We thank the reviewer for this comment. We agree that this formulation was misleading.

In the revised manuscript, we corrected this statement and clearly specified the measurement resolution. The related paragraph read as follows:

(Wafer-level transfer of graphene and graphene/hBN heterostructures)

[...] Scanning across the sample surface yields spatially resolved images of the graphene sheet resistance on wafer-scale. For our measurements, we set the spatial resolution to 500 μm by adjusting the distance of the near-field detector to the sample and the pitch between acquisition sites. [...]

Reviewer comment #13

(13) Please do not use “wafer level transfer” especially for multilayer transfers, because it was not demonstrated. Transfer of “large areas” would be an example of a term closer to reality.

Our response:

We thank the reviewer for this comment. We agree that we demonstrated the stacking of layers only in centimeter sized areas on 100 mm diameter target wafers.

For the experiments to realize 2-layer graphene, we deliberately chose these dimensions to clearly visualize the difference between 1-layer and 2-layer regions by THz spectroscopy. For the graphene/hBN heterostructures, the size was imposed by the availability of the hBN on Cu foil. Based on the observation of the uniform coverage of transferred 1-layer graphene on full wafer scale and the large area ($\sim 15 \text{ cm}^2$) of the formed 2-layer graphene, we are confident that both processes also work on full wafer size.

In the revised manuscript, we changed our terminology related to the stacked layers as suggested. The related paragraph read as follows:

(Abstract)

[...] Furthermore, we form 2D material heterostructures by stacking of CVD grown hexagonal boron nitride and graphene layers on centimeter-sized areas. [...]

(Wafer-level transfer of graphene and graphene/hBN heterostructures)

[...] To demonstrate the viability of our methodology for wafer-level transfer of 2D materials and formation of large-area van der Waals heterostructures, we integrated graphene/hBN heterostructures (see Methods for details of the process and materials). [...]

[...] We used Terahertz time-domain-spectroscopy (THz-TDS) to further characterize the electrical properties and homogeneity of the wafer-level transferred graphene and large-area graphene/hBN heterostructure. [...]

(Repeated transfer of graphene)

[...] We performed two consecutive transfers of monolayer (1-layer) graphene from their copper growth substrates to a 100 mm diameter silicon wafer to demonstrate that our transfer methodology is suitable for high-yield formation of double-layer (2-layer) graphene on large areas (we avoid the term “bilayer graphene” to distinguish these films from Bernal stacked materials, see Methods for process details). [...]

Reviewer comment #14

(14) Please comment on the differences in sheet carrier resistivity on Fig.2 and 3 for single layer graphene. The values differ by a factor of 4. The distribution of the resistance value is also much broader in Fig. 3.

Our response:

We thank the reviewer for pointing us at this observation. We attribute the discrepancy in sheet resistance to a difference in graphene quality before transfer. The process parameters to form 2-layer graphene and to form the graphene/hBN heterostructure were essentially the same.

For the 2-layer graphene sample, the second graphene sheet that forms the 1-layer region on the target wafer, was from another batch and considerably older (by mistake) than the other sheets used in this work. We assume that the quality of the graphene on that growth substrate was substantially lower. Unfortunately, we discovered this disparity at a late stage of drafting this manuscript. However, we decided to present the full dataset of Raman and THz spectroscopy measured on this sample, since its purpose is to demonstrate the formation of 2-layer graphene by repeated transfer and not benchmark the graphene quality itself.

In the revised manuscript, we have added a sentence to clarify this disparity. The related paragraph reads as follows:

(Repeated transfer of graphene)

[...] We attribute the difference in the 1-layer graphene R_{sh} between these experiments and the dataset presented in Figure 2 to a dissimilarity in graphene quality before transfer. The graphene sheet in Figure 3a that forms the 1-layer region on the target wafer was from a different batch than the graphene sheet used in Figure 2. [...]

Reviewer comment #15

(15) Line 248: The authors extract a maximum field-effect mobility of $2630 \text{ cm}^2 \text{ V}^{-1} \text{ s}^{-1}$ and state that the quality of the graphene is preserved after transfer. To prove this statement properties of the as grown graphene have to be given. Furthermore, it would be interesting to know how the values were extracted.

Our response:

We thank the reviewer for this comment. We agree that we have not been sufficiently clear in the formulation of this statement. It is correct that we did not quantitatively characterize the quality of the graphene before transfer (e.g. by Raman spectroscopy) since it is beyond the capability of our measurement setup. However, we extracted the mobility from two different types of devices: (1) The van-der-Pauw devices in Figure 2b ($2800 \pm 100 \text{ cm}^2 \text{ V}^{-1} \text{ s}^{-1}$), which were intentionally fabricated by non-contact methods (shadow-mask evaporation and laser ablation) to avoid potential graphene degradation during the device fabrication. (2) The top-gated field-effect device ($2630 \text{ cm}^2 \text{ V}^{-1} \text{ s}^{-1}$), which were fabricated by standard processing techniques (high temperature Al_2O_3 deposition, multiple lithographic layers, metal depositions and etching processes). Since the mobility is similar in both cases, we conclude that the fabrication of the field-effect devices did not affect the graphene quality. This is important since it demonstrates that our method is compatible with conventional processing technology.

To further clarify this point **in the revised manuscript**, we have revised our statement regarding the preserved graphene quality and extended the description and discussion of the electrical measurements and mobility extraction. The related paragraphs read as follows:

(Field-effect devices)

We fabricated and evaluated top-gated field-effect devices on 100 mm diameter silicon wafers using four photolithography layers and conventional processing technology (inset in Figure 3d, details see Methods). The extracted sheet resistance and charge carrier mobility as a function of the gate voltage of a typical device in Figure 3d resemble the characteristic behavior of gated graphene

devices⁵⁰ (see Methods for measurement details), where the mobility is zero at the Dirac point due to a zero of the transconductance. At this point, the charge carriers convert from electrons to holes or vice versa. In our measurements, the mobility is not exactly zero because of the limited number of data points. For this device, the maximum field-effect mobility of holes is $2630 \text{ cm}^2\text{V}^{-1}\text{s}^{-1}$, which is similar to the values extracted from Hall-measurements of the vdP devices in Figure 2b. This agreement demonstrates that the high quality of the transferred graphene is preserved during device manufacturing with multiple processing steps such as high temperature Al_2O_3 deposition, multiple lithographic layers, metal depositions and etching processes. Consequently, these results support the claim that our transfer methodology is compatible with for large-scale manufacturing of graphene devices using conventional semiconductor process technology. The two-point transfer characteristics of additional devices are shown in Figure S5.

(Methods)

[...] Electrical measurements were performed under ambient conditions at room temperature. The sheet resistance (R_{sh}) was measured by 4-probe van-der-Pauw method with top-gating. A current (I_{12}) was forced between electrode 1 and 2 (Figure 3d), while the voltage drop between electrode 3 and 4 (V_{34}) was measured. The sheet resistance (R_{sh}) was calculated by⁵¹

$$R_{\text{sh}} = \frac{V_{34}}{I_{12}} \frac{\pi}{\ln 2}$$

and the mobility (μ) was then derived by

$$\mu = \frac{1}{C_G} \frac{dG_{\text{sh}}}{dV_G}$$

where

$$G_{\text{sh}} = \frac{1}{R_{\text{sh}}}$$

and V_G and C_G are the gate voltage and the capacitance of the dielectric per unit area ($k = 7$ for Al_2O_3), respectively.

Reviewer comment #16

(16) Line 279: The authors mention “intercalation at the MoS2/SiO2”. The term intercalation is not used properly. Intercalation means introduction of additional material (atoms, molecules) without lifting layers (see for example the overwhelming literature of graphite intercalation). Therefore, the question arises have the authors noticed intercalation effects as a consequence of their transfer process or processing sequences. This question is important, because intercalation (see for example Ge intercalation of graphene published by Seyller) may change the electronic properties of the two dimensional materials.

Our response

We thank the reviewer for this comment and clarification. We agree that our terminology was not adequate to describe the detachment of the MoS₂ layer from the growth substrate. Since the distance between the MoS₂ layer and the substrate may change, we believe that the term “permeation into the interface” is a more suitable expression. In our experiments, we could not attribute any change in electronic properties to intercalation effects of molecules into the MoS₂.

In the revised manuscript, we corrected our description of the transfer of MoS₂. The related paragraphs read as follows:

(Methodology for wafer-level transfer of 2D materials and their heterostructures)

[...] After wafer bonding, the growth substrate is removed (Figure 1a, (4)) by either etching, delamination, or permeation of liquids into the interface between the 2D material and the growth substrate, which leaves the 2D material transferred on the target wafer (Figure 1a, (I)). [...]

(Transfer of MoS₂)

[...] Next, we submerged the bonded stack in a potassium hydroxide (KOH) solution, which permeated into the MoS₂/SiO₂ interface and detached the MoS₂ from its growth substrate. This process took tens of seconds, after which the MoS₂ remained transferred on top of the BCB layer of the target wafer. [...]

(Methods)

[...] Submersion of the bonded stack in KOH solution detached the growth substrate from the MoS₂ layer by permeation of KOH into the MoS₂/substrate interface within tens of seconds. [...]

Reviewer comment #17

(17) Can the authors add a comment on the shift of the A1g mode being approximately 1 cm⁻¹. Please comment also on the change in the peak heights. Furthermore, as in the case of the transferred graphene not only two spectra have to be shown, but an analysis over a mapping area (see for example Fig. 2d). This is required, because it was stated that large area single layer MoS₂ was grown and transferred. This has to be demonstrated and cannot be done with two single spectra.

Our response:

We thank the reviewer for this excellent suggestion that triggered us to perform additional measurements and analysis. We agree that the analysis and the extent of the Raman measurements on MoS₂ were less comprehensive than of graphene.

In the revised manuscript, we extended the Raman spectroscopy of MoS₂ by additional area mappings and detailed statistical analysis. We normalized the spectra to the intensity of the E_{2g}¹ mode to clarify the comparison of the peak positions before and after transfer. Our new measurements support our claim of a successful large-area transfer. Besides, from the statistical analysis we concluded that the MoS₂ is a multilayer instead of a single layer, which we corrected throughout the manuscript. The

related paragraph and figures in the main text as well as the Supplementary information are presented as follows:

(Transfer of MoS₂)

Figure 4: Transfer and characterization of MoS₂. **a)** Averaged Raman spectra of multilayer MoS₂ on the SiO₂/Si growth substrate before transfer (blue) and after transfer to the target wafer (red). The Raman spectra are averages of area scans with 8 × 10 measurements in 1 mm² areas (black rectangle in the photograph of transferred MoS₂) and were normalized to the intensity of the E¹_{2g} mode. The shaded regions represent the standard deviation of the intensity. The vertical dashed lines indicate the peak positions of the E¹_{2g} and A_{1g} modes of the as-grown film before transfer. **b)** Averages of 25 photoluminescence (PL) intensity spectra of pristine (blue) and transferred (red) multilayer MoS₂ with A and B excitonic transitions (shaded areas: standard deviation). The inset magnifies the region from 1.945 eV to 1.98 eV and shows the Raman modes of BCB, which are superimposed with the optical response of the MoS₂ layer.

(Transfer of MoS₂)

[...] Typical Raman spectra prior transfer (Figure 4a) exhibited the characteristic E¹_{2g} and A_{1g} modes of pristine MoS₂ at 384.6 cm⁻¹ and 409.1 cm⁻¹, respectively, which are in good agreement with reported values for MoS₂^{37,52–54}. After transfer, we found the peaks at 384.7 cm⁻¹ (E¹_{2g}) and 407.2 cm⁻¹ (A_{1g}), i.e. without detectable shifts in the position of the E¹_{2g} mode within the accuracy of our Raman setup. Since the E¹_{2g} mode is highly sensitive to uniaxial strain⁵⁵, we conclude that the strain in the transferred MoS₂ film was the same as present on the growth substrate and was not significantly affected by our transfer methodology. In contrast, the A_{1g} mode is sensitive to changes in the substrate material, which could cause the observed shift of 1.8 cm⁻¹^{56,57}. [...]

(Supplementary Information)

6. Analysis of Raman spectroscopy of MoS₂ and BCB

Figure S6: a) Histograms of the E_{2g}^1 mode position of pristine (blue) and transferred (red) MoS_2 films, extracted from area scans with 8×10 measurements in 1 mm^2 areas. Gaussian fits yield mode positions of 384.6 cm^{-1} and 384.7 cm^{-1} for pristine and transferred MoS_2 , respectively. The broadening of the E_{2g}^1 position distribution is due to lower signal intensity which results in higher uncertainty of the fitting quality of the pseudo-Voigt function to the data. **b)** Histograms of the A_{1g} mode position of pristine (blue) and transferred (red) MoS_2 films, extracted from the same area scans as in a). Gaussian fits yield mode positions of 409.1 cm^{-1} and 407.2 cm^{-1} for pristine and transferred MoS_2 , respectively. The A_{1g} mode is sensitive to changes in the substrate material which could cause the overserved shift of 1.9 cm^{-1} ^{56,57}.

Reviewer comment #18

(18) Yield data of the fabricated devices have to be added to the paper.

Our response:

We thank the reviewer for this comment. We agree that a yield data of the fabricated devices would be very insightful. Due to access restrictions to our laboratories during the COVID-19 outbreak, it is difficult for us to perform supplementary characterization or fabrication experiments. Nevertheless, we were able to extract additional data, which shows characterization of several devices. As shown below, 7 devices were measured with 2-point top gate transfer characterization, which we have included in the Supplementary information of the revised manuscript. These devices all show similar behavior in terms of current and doping level. As a general note, the main focus of our paper is on presenting the novel transfer method. A proper yield investigation requires characterization of a large number of devices on the wafer, which was not planned for when we prepared the manuscript. Instead, the purpose of the FET fabrication was to show that our proposed transfer method is in principle compatible with conventional semiconductor process technology.

In the revised manuscript, we included a figure with the two-point I-V characteristic in the Supplementary Information and a related reference in the main text. The paragraph and figure are presented as follows:

(Field-effect devices)

[...] The two-point transfer characteristics of additional devices are shown in Figure S5. [...]

(Supplementary Information)

5. Two-point transfer characteristic of top-gated field effect devices

Figure S5: Two-point transfer characteristic of seven top-gated field-effect devices at room temperature.

Reviewer comment #19

(19) Line 292: "... performed a statistical analysis of the resulting device and materials characteristics". Statistical analysis of transistor performance are not given in the paper. A comparison of charge carrier mobilities obtained in graphene FETs and vdP structures is not given.

Our response:

We thank the reviewer for pointing this out and we apologize for the oversight. It is correct that the statistical analysis of resulting field-effect devices is not given in the manuscript. While we provide statistical data for the van-der-Pauw structures in Figure 2b, we do not have statistical data for the FET devices as discussed in our response to comment #18 of reviewer #2. The measurements shown in Figure 2b and Figure 3d are measured using the 4-point van der Pauw method, since it provides measurement results which are not affected by contact resistance. Therefore, both measurements are contact resistance free and thus, comparable. This comparison has been already shown in our response to comment #15 of reviewer #2.

In the revised manuscript, we clarified our statement and added a comparison between the mobility in FETs and vdP devices. The related paragraphs state as follows:

(Conclusion)

[...] To illustrate its applicability, we manufactured graphene devices on 100 mm diameter silicon, performed a statistical analysis of the material characteristics, and demonstrated that our transfer method is, in principle, compatible with the conventional process technology. [...]

(Field-effect devices)

[...] For this device, the maximum field-effect mobility of holes is $2630 \text{ cm}^2\text{V}^{-1}\text{s}^{-1}$, which is similar to the values extracted from Hall-measurements of the vdP devices in Figure 2b. This agreement demonstrates that the high quality of the transferred graphene is preserved during device manufacturing with multiple processing steps such as high temperature Al_2O_3 deposition, multiple lithographic layers, metal depositions and etching processes. [...]

Reviewer comment #20

(20) Please add the temperature of the electrical measurements.

Our response:

We thank the reviewer for pointing at this important detail. All electrical measurements were performed at room temperature.

In the revised manuscript, we added the temperature of the electrical measurements. The related paragraphs read as follows:

(Methods)

Fabrication and evaluation of van-der-Pauw devices

[...] Electrical measurements were performed under ambient conditions at room temperature, using a parameter analyzer (Keithley SCS4200). [...]

(Caption Figure 2)

[...] **b)** Extracted sheet resistance (R_{sh}), carrier density (n), and charge carrier mobility (μ) from Hall-measurements of vdP devices at room temperature (Number of devices: 18 graphene; 9 graphene/hBN). [...]

(Caption Figure 3)

[...] **d)** Top-gate voltage dependency of the graphene sheet resistance and the field-effect mobility in an integrated graphene device at room temperature. [...]

(Supplementary Information)

[...] To characterize the fabricated devices, we measured the electrical 2-terminal resistances between the outer electrodes of 202 devices at room temperature using a parameter analyzer (Keithley SCS4200, Tektronix, Inc.) (Figure S1c).

Reviewer comment #21

(21) Fig. 3d: Why the mobility of the charge carriers shows a minimum, if it is assumed that the mobility scales with concentration lower charge carrier concentration lead to higher charge carrier mobility. Please comment

Our response:

We thank the reviewer for this question and the opportunity to further clarify this point. Indeed, the minimum of the mobility should not be a limited value as shown in the figure, but rather be zero. This point corresponds to the zero transconductance (derivative in transfer curve) at the Dirac point. At this point the carrier type converts from electrons to holes or vice versa. However, since the sweep of the gate voltage has a limited number of data points, unfortunately, our calculations do not show a zero-mobility at this transition.

The statement “*lower charge carrier concentration lead to higher charge carrier mobility*”, is only correct when we assume that the conductivity is a constant, according to the Drude formula for the conductivity $\sigma = en\mu$ (e : elementary charge, n : charge carrier concentration and μ : charge carrier mobility). This scaling behavior of the mobility shown in Figure 3d is typical for graphene-based devices (see for example Figure S2a and S2b in Uzlu et al.⁵⁰).

In the revised manuscript, we clarified the interpretation of the mobility and included a reference to the characteristic behavior of the mobility for gated graphene devices. The related paragraph reads as follows:

(Field-effect devices)

[...] The extracted sheet resistance and charge carrier mobility as a function of the gate voltage of a typical device in Figure 3d resemble the characteristic behavior of gated graphene devices⁵⁰ (see Methods for measurement details), where the mobility is zero at the Dirac point due to a zero of the transconductance. At this point, the charge carriers convert from electrons to holes or vice versa. In our measurements, the mobility is not exactly zero because of the limited number of data points. [...]

Reviewer #3 (Remarks to the Author)

Our reviewer comment:

Review of Large-Area Integration of Two-Dimensional Materials and Their Heterostructures by Wafer Bonding

Quellmalz et al. investigated a new 2D material transfer method which is large-area, allows stacking into heterostructures, does not require ultra-specialised equipment, and doesn't completely degrade the transport properties. Generally the work is well presented, appropriate methods were used for characterisation, and data is of high quality, and sufficient detail is given for others to reproduce the work. I believe the method is likely to be useful for the 2D materials community as well as industry, and that Nature Communications is an appropriate journal for the work. However, in my opinion the authors probably overstate how gentle their process is, at least based on the presented evidence. I also have a few other points which require clarification.

Our response:

We thank the reviewer for the positive assessment of our work and for pointing out aspects that can be improved.

Reviewer comment #1

1. THz-TDS map from Figure 2c. Around the edges of the hBN square there are noticeably damaged areas on the graphene sheet- or at least areas which have sheet resistances about twice the rest of the sheet. What has caused this? When making heterostructures with your method is this interface degradation unavoidable?

Our response:

We thank the reviewer for this observation which indeed deserves an explanation in the manuscript.

In this specific transfer of hBN, the contact area of the bond chuck with the substrate stack is slightly smaller than the donor substrate (2.5 cm x 2.5 cm copper foil). During bonding, the center region of the donor substrate was pressed uniformly against the target wafer while the rim of the foil did not experience any force. This mismatch of force in the center and the rim caused the edge of the foil to bend upwards, which, consequently, did not completely attach to the target wafer and caused inhomogeneities in the hBN layer and the BCB layer. In the following transfer of graphene, those inhomogeneities locally affected the bond quality, which, in turn, affected the graphene sheet resistance. This effect may be prevented by carefully adjusting the contact area of the bond chuck with the substrate stack to the size of the copper foil. Thus, the lateral interface between regions with different layer stacks may be improved.

In the revised manuscript, we included a discussion of the cause of the interface degradation and a plausible measure of how to improve the interface. The related paragraph reads as follows:

(Wafer-level transfer of graphene and graphene/hBN heterostructures)

[...] At the lateral transition between the graphene/hBN heterostructure and the monolayer graphene (i.e. the outer edges of the heterostructure) the graphene sheet resistance is increased and less uniform. This variation was most likely caused by surface inhomogeneities in the BCB layer that resulted from a partly detached edge region of the hBN on copper foil in the first transfer. This effect is known to occur in wafer bonding of compliant substrates where the contact area of the wafer chuck is smaller than the substrate. Precise adjustment of the contact area between the wafer bonder and the donor substrate to the size of the copper foil should improve the lateral interface between regions with different layer stacks.

Reviewer comment #2

2. For the hBN/graphene vs. graphene results, how can you tell that you are not just damaging the hBN during transfer and that is the reason why there is no increased device performance for the

Our response:

We thank the reviewer for posing this very relevant question. We agree that damages in the hBN encapsulation are likely to diminish the beneficial effect of hBN on the transport properties of the graphene. In our case, we use CVD grown hBN, which is known to have inherent defects between individual crystalline grains⁵⁸. Since the grain size of both graphene and hBN are much smaller (micrometer range) than the size of our van-der-Pauw devices (millimeter range), our results are average values of the material properties over the entire device area, comprising a multitude of grains, grain boundaries and grain defects. Our observation that CVD hBN does not improve the charge carrier mobility in polycrystalline graphene agrees with work by Pandey et al.¹⁷ and Shautsova et al.¹⁸. Taken together, these are four reports of different types of transfer processes (2 wet transfers, 1 dry transfer and our bond transfer) that present similar results. Based on these published results and our data, we believe that our observations are mainly caused by the quality of currently available large-area CVD hBN substrates, rather than the transfer processes.

In the revised manuscript, we clarified our comparison with the above-mentioned studies and included a note of the used transfer methods. The related paragraph reads as follows:

(Wafer-level transfer of graphene and graphene/hBN heterostructures)

[...] The mobility in our graphene/hBN heterostructure is similar to values reported by Pandey et al. and Shautsova et al., who encapsulated graphene in CVD hBN using wet and dry transfers^{17,18}. In accordance with these works, our results indicate that current large-area CVD hBN substrates typically do not increase the mobility in polycrystalline graphene, thus contrasting several studies on single crystals of graphene and hBN^{19–22}. We emphasize that for a meaningful comparison, the initial graphene quality before the transfer is of crucial importance. [...]

Reviewer comment #3

3. Why no MoS₂ transistor measurements? Seeing as you are growing on Si, a comparison of devices made on the as-grown substrate vs transferred (using same fab methods, same substrate etc. just with BCB on transferred substrate) would be nice evidence that you aren't damaging the MoS₂ by transferring. Otherwise, for MoS₂ the evidence for large-scale transfer preserving the properties you present is a lot weaker in comparison to graphene, with single Raman/PL spectra- and no mapping. It is valuable to show the MoS₂ data, but statements such as: "Hence, our methodology is suitable for large area transfer of MoS₂ films." With no electrical measurements I think "potentially" is required, at least.

Our response:

We thank the reviewer for this excellent comment and suggestion. We agree that we presented less evidence for the large-area transfer of MoS₂ than for graphene.

In the revised manuscript, we added a new set of PL and Raman measurements from area scans of MoS₂ on BCB, which provide further evidence of a successful transfer MoS₂ over large areas. Unfortunately, we were not able to perform the suggested experiments to demonstrate MoS₂ devices due to access restrictions to our laboratories during the COVID-19 outbreak, and instead follow the reviewer's suggestion to reformulate our concluding sentence. The revised paragraph and figure in the main text and the additional figures in the Supplementary Information are presented as follows:

(Transfer of MoS₂)

Figure 4: Transfer and characterization of MoS₂. **a)** Averaged Raman spectra of multilayer MoS₂ on the SiO₂/Si growth substrate before transfer (blue) and after transfer to the target wafer (red). The Raman spectra are averages of area scans with 8 × 10 measurements in 1 mm² areas (black rectangle in the photograph of transferred MoS₂) and were normalized to the intensity of the E_{2g}¹ mode. The shaded regions represent the standard deviation of the intensity. The vertical dashed lines indicate the peak positions of the E_{2g}¹ and A_{1g} modes of the as-grown film before transfer. **b)** Averages of 25 photoluminescence (PL) intensity spectra of pristine (blue) and transferred (red) multilayer MoS₂ with A and B excitonic transitions (shaded areas: standard deviation). The inset magnifies the region from 1.945 eV to 1.98 eV and shows the Raman modes of BCB, which are superimposed with the optical response of the MoS₂ layer.

(Transfer of MoS₂)

[...] Typical Raman spectra prior transfer (Figure 4a) exhibited the characteristic E_{2g}¹ and A_{1g} modes of pristine MoS₂ at 384.6 cm⁻¹ and 409.1 cm⁻¹, respectively, which are in good agreement with reported values for MoS₂^{37,52-54}. After transfer, we found the peaks at 384.7 cm⁻¹ (E_{2g}¹) and 407.2 cm⁻¹ (A_{1g}), i.e. without detectable shifts in the position of the E_{2g}¹ mode within the accuracy of our Raman setup. Since the E_{2g}¹ mode is highly sensitive to uniaxial strain⁵⁵, we conclude that the strain in the transferred MoS₂ film was the same as present on the growth substrate and was not significantly affected by our transfer methodology. In contrast, the A_{1g} mode is sensitive to changes in the substrate material, which could cause the observed shift of 1.8 cm⁻¹^{56,57}. To further investigate the quality of the transferred MoS₂ film, we compare the average of 25 normalized photoluminescence (PL) intensity spectra of pristine and transferred MoS₂ (Figure 4b). Before transfer, we found two prominent peaks at 1.821 eV and 1.968 eV, which correspond to A and B excitonic transitions, respectively⁵⁹. After transfer, we observed a drastic increase in the A/B intensity ratio since the B-excitation vanishes. A high A/B intensity ratio indicates a high quality of the MoS₂ film⁶⁰. The A-exciton transition energy is blue-shifted by 13 meV to 1.834 eV (see Figure S9 for statistics), which we attribute to the change of the substrate material from SiO₂ (before transfer) to BCB (after transfer)⁵⁷. The additional features of the PL intensity between 1.96 eV and 1.97 eV are Raman modes of the underlying BCB, hence, do not belong to the excitonic transitions in the MoS₂ (see Supplementary Information for the Raman spectrum of BCB). Taken together, the Raman and PL measurements confirm that the layer quality is preserved during the transfer process. Hence, our methodology is potentially suitable for large-area transfer of MoS₂ films.

(Supplementary Information)

6. Analysis of Raman spectroscopy of MoS₂ and BCB

Figure S6: a) Histograms of the E_{2g}¹ mode position of pristine (blue) and transferred (red) MoS₂ films, extracted from area scans with 8 × 10 measurements in 1 mm² areas. Gaussian fits yield mode positions of 384.6 cm⁻¹ and 384.7 cm⁻¹ for pristine and transferred MoS₂, respectively. The broadening of the E_{2g}¹ position distribution is due to lower signal intensity which results in higher uncertainty of the fitting quality of the pseudo-Voigt function to the data. **b)** Histograms of the A_{1g}

mode position of pristine (blue) and transferred (red) MoS₂ films, extracted from the same area scans as in a). Gaussian fits yield mode positions of 409.1 cm⁻¹ and 407.2 cm⁻¹ for pristine and transferred MoS₂, respectively. The A_{1g} mode is sensitive to changes in the substrate material which could cause the overserved shift of 1.9 cm⁻¹ ^{56,57}.

(Supplementary Information)

7. Analysis of photoluminescence spectroscopy of MoS₂

Figure S9: Histogram of the peak positions of the A-excitonic transition in MoS₂ before and after transfer. After transfer, the mean position is blue shifted by 13 meV.

Reviewer comment #4

4. Several times throughout the paper is strong claims are made such as (line 106) ...substrate without exerting excessive pressure on the 2D material, and hence without damaging, wrinkling or straining the 2D material. This is written as if no damage or wrinkling or strain occurs at all. I think these claims should at least include the word 'minimised', for instance. If you have Raman data of the graphene pre-transfer to suggest the strain is similar before your process? Then a similar claim with the word 'minimised' included would be appropriate.

Alternatively, you could transfer graphene, measure strain and d-peak via Raman (+THz-TDS), re-transfer the same graphene sheet and remeasure (Raman + THz), this would also provide very strong evidence of a lack of degradation due to processing. As presented, the data only suggest that the method is quite good at transferring graphene compared to other methods, but it is not possible to tell if it is being damaged or not- especially based on transport properties.

Our response:

We thank the reviewer for pointing at the imprecise expressions and the suggestion. We agree that we have left room for misinterpretation and may have overstated our results.

Unfortunately, we do not have Raman measurements of graphene on copper foil before transfer. Our Raman system is equipped with a green laser (532 nm wavelength) which creates a strong luminescence background signal in the copper substrate and makes the characterization of graphene on copper foil challenging. The interested reader might refer to work by Costa et al.⁶¹. However, we performed Raman spectroscopy of the MoS₂ layer before and after transfer and found the peaks of the E_{2g}¹ and A_{1g} mode at the same position within the accuracy of our Raman setup (spectral resolution: 1.2 cm⁻¹). Since the E_{2g}¹ position is highly sensitive to strain, we see this result as indication that the strain in the transferred MoS₂ layer is similar as on the growth substrate and that our transfer process minimizes degradation by additional strain.

We also agree that the suggested sequence of repeated transfer, characterization and transfer of the same graphene sheet would provide convincing evidence of possible degradation during processing. However, due to the chemical stability of the partially cured BCB, we believe that it is not possible to remove the adhesive BCB layer with conventional wet-etching (e.g. in acetone or mesitylene) or dry plasma etching without affecting the properties of the graphene. For this reason, we did not attempt to evaluate the proposed transfer and characterization procedure.

As the reviewer pointed out correctly, our data suggests that our transfer yields good material properties compared to other transfer methods. As suggested by the reviewer we have clarified our claims and statements regarding possible damage and wrinkling introduced in the transferred layers by our method.

In the revised manuscript, the related paragraphs read as follows:

(Introduction)

[...] All transferred layers and heterostructures were of high quality, i.e. they featured uniform coverage and little strain in the transferred 2D materials. These findings indicate that our proposed integration approach preserves similar mechanical properties of the 2D materials as present on the growth substrate while minimizing degradation of the transferred layers through introduction of wrinkles or excessive strain. [...]

(Methodology for wafer-level transfer of 2D materials and their heterostructures)

[...] Thus, the adhesive layer molds against the 2D material and forms a stable bond to the target wafer while replicating the surface topography of the growth substrate without exerting excessive pressure on the 2D material. [...]

(Conclusion)

[..] After removal of the growth substrate, the 2D material remains on the target wafer in the same shape as it was synthesized on the growth substrate. This feature minimizes defects or additional strain in the 2D material by avoiding deformation. Additionally, it reduces wrinkles, which may result from excess material due to different surface topography of the growth substrate and the target wafer. We hypothesize that this feature contributes to preserving similar mechanical properties of the transferred 2D materials. [...]

Reviewer comment #5

5. Re: data in Figure 2b. You measured 18 graphene and 9 graphene/hBN devices, which seems like a reasonable sample size. Can you mention the device yield? And if any devices failed, what do you attribute this to?

Our response:

Thank you for the positive comment on the sample size. In total, we measured 27 devices of which 18 comprised graphene on BCB and 9 graphene/hBN on BCB. The entire dataset is presented in the paper, without excluding any device.

In the revised manuscript, we included a note on the total number of devices and the yield of the vdP measurements. The related paragraphs read as follows:

(Wafer-level transfer of graphene and graphene/hBN heterostructures)

[...] Four-point probe measurements of a total of 27 vdP devices yielded the graphene sheet resistance (R_{sh}), carrier density (n) and mobility of majority charge carrier (μ) of the transferred graphene layer, both for graphene resting on BCB and for graphene resting on hBN (Figure 2b) (see Methods). In both cases, all measured devices were functioning and we found a similar R_{sh} ($630 \pm 90 \Omega/sq.$ and $600 \pm 90 \Omega/sq.$, respectively) whereas the n and μ showed differences for graphene resting on BCB and on hBN, respectively. [...]

Reviewer comment #6

6. It is true that “neither the interface between the first and second 2D material nor the second 2D material are exposed to any polymer carrier nor adhesive that may potentially degrade the 2D material properties by contamination” and is a strength of your method. However, surface water will be trapped permanently. Could this have an effect on heterostructures? Can all surfaces be heated during the process?

Our response:

We thank the reviewer for pointing out the importance of residuals and trapped molecules between layers in heterostructures. We agree, since the interface in van der Waals heterostructures is of utmost importance, impurities between layers will strongly affect their properties. Our method reduces the risk of trapped water since it is carried out in vacuum and at elevated temperature.

In our method, the second transfer (graphene onto hBN) was performed in vacuum atmosphere ($2 \cdot 10^{-4}$ mBar) which is considerably below the vapor pressure of water at room temperature (31 mBar). Hence, we expect residual water to desorb from the surfaces and not to be trapped between layers. Additionally, the entire bond stack was heated up to 190 °C which further increases the desorption of molecules from all surfaces.

In the revised manuscript, we included this aspect. The related paragraphs read as follows:

(Wafer-level transfer of graphene and graphene/hBN heterostructures)

[...] First, we attached the hBN on copper foil (2.5 cm by 2.5 cm) to the adhesive on the spin-coated target wafer by wafer bonding. After etching away the copper in FeCl_3 solution and rinsing in deionized water, we placed the monolayer graphene on copper foil (100-millimeter diameter) on top of the target wafer and performed a second bonding process in a vacuum atmosphere. [...] Note that in the heterostructure, neither the graphene nor the top surface of the hBN was in contact with polymers or adhesives at any time. Moreover, the second transfer was performed in a vacuum and at elevated temperature, which reduces the amount of water and gas molecules that may get trapped between the layers. [...]

Reviewer comment #7

7. Line 161 For the transferred graphene monolayer on BCB, the extracted mobilities are in the higher range of previously reported values for CVD graphene transferred on large areas by various methods, including wet transfer with polymeric carrier layers^{38,55}, face-to-face transfer by bubble formation⁵⁶lamination⁴⁶. In ref 46 by Shivayogimath et al (Fig 5b), the THz-TDS results for mobility of large-area transfer are similar to your results (slightly less) for etching transfer, but closer to 4000 cm^2/Vs for PVA. The text can be updated so not to suggest your values are the highest reported for large-area transfer of graphene.

Our response:

We thank the reviewer for pointing out this ambiguous comparison of our results. We agree that higher mobility values have been reported by other groups. However, the type and initial quality of graphene is also an important factor for its transport properties after transfer. Therefore, it is crucial to consider those parameters when comparing between different transfer methods. Shivayogimath et al.⁶² extracted the mobility of graphene in two sets of experiments with different types of graphene and methods:

Set (1): Polycrystalline graphene grown by CVD on commercial Cu foil (8 cm × 8 cm). The field-effect mobility was extracted by back-gating vdP devices (5 mm × 5 mm) and was found to be in the range of 350–800 $\text{cm}^2\text{V}^{-1}\text{s}^{-1}$.

Set (2): Single-crystal monolayer graphene grown by CVD on single-crystal Cu(111) foil (2 cm × 3 cm). The mobility was extracted by THz-TDS (Average of Gaussian fit: 3707 $\text{cm}^2\text{V}^{-1}\text{s}^{-1}$).

In our experiments, we use polycrystalline graphene grown on polycrystalline Cu foil, and we extracted the mobility by Hall-measurements of vdP devices (3 mm × 3 mm). These conditions are similar to the first set of experiments by Shivayogimath et al.. Therefore, we believe a comparison to those results is more justified and meaningful.

In the revised manuscript, we clarified the comparison of our results and included a sentence on the importance of the initial graphene quality. The related paragraphs read as follows:

(Wafer-level transfer of graphene and graphene/hBN heterostructures)

[...] For the transferred graphene monolayer resting on BCB, the extracted mobilities are in the higher range of previously reported values for polycrystalline CVD graphene transferred on large areas from polycrystalline copper substrates by various methods, including wet transfer with polymeric carrier layers^{13,14}, face-to-face transfer by bubble formation¹⁵ and lamination¹⁶. The mobility in our graphene/hBN heterostructure is similar to values reported by Pandey et al. and Shautsova et al., who encapsulated graphene in CVD hBN using wet and dry transfers^{17,18}. In accordance with these works, our results indicate that current large-area CVD hBN substrates typically do not increase the mobility in polycrystalline graphene, thus contrasting several studies on single crystals of graphene and hBN^{19–22}. We emphasize that for a meaningful comparison, the initial graphene quality before the transfer is of crucial importance. [...]

Reviewer comment #8

8. Line 152: materials by femtosecond-laser ablation to define van-der-Pauw (vdP) devices for characterization of the electrical properties of the 2D materials. Were you able to pattern the graphene without damaging the underlying BCB or substrate? if so, how did you determine this? If not, could this affect device properties?

Our response:

We thank the reviewer for this question. Since the ablation by ultrashort laser pulses relies on multi-photon absorption, hBN may be ablated during the process, albeit the photon energy in our system (2.38 eV) is lower than the band gap in hBN (~6 eV⁶³). To ensure a proper insulation of the vdP devices, we chose a laser power which exceeded the minimum power setting to ablate graphene. Therefore, we believe that the hBN as well as the surface of the BCB are likely to be damaged during the process. However, we do not expect any influence on the device properties by the patterned hBN nor potential damages on the surface of the BCB layer. Both hBN and BCB are electrically insulating, hence, neither should contribute to the electrical measurement signal by a relevant current flow through the layer.

However, the strong light absorption in graphene contrasts the absorption in hBN. Therefore, we hypothesize that it may be feasible to find a set of parameters which allows selective patterning of graphene without damaging hBN. This possibility might be of interest for fully encapsulated hBN/graphene/hBN heterostructures for instance. We will leave the investigation of this hypothesis to the interested reader and future work.

In the revised manuscript, we stated more precisely that the entire heterostructure is patterned. The related paragraph read as follows:

(Wafer-level transfer of graphene and graphene/hBN heterostructures)

[...] Next, we integrated Ti/Au electrodes and patterned the 2D material heterostructure to van-der-Pauw (vdP) devices, which allow electrical material characterization on the target wafer (Figure 2a). [...]

Reviewer comment #9

9. Regarding THz-TDS maps, you use a near-field set up with spatial resolution down to 10 μm . However, at least for far-field set ups (with perhaps 100 μm spatial resolution) the actual transport length scale is $\approx 10\text{-}100$ nm (see 2D Mater. 4 (2017) 042003 Table 1 and discussion on p18) and depend on graphene mobility for instance. I think the authors should state the transport length scale for their set up/graphene and make it clear for the reader that for THz: spatial resolution \neq transport length. This will also lead to analysis of your data which supports the quality of your method, as it seems your sheet R values match well between vdP and THz, suggesting you do not have significant defects/inhomogeneity on the larger scale that can often effect sheet R in larger devices.

Our response:

We thank the reviewer for this profound comment. We agree that a detailed background for interpreting the THz measurements was missing in the manuscript.

In the revised manuscript, we clarified the difference between spatial resolution and the transport length which we also estimate for our sample and setup. The related paragraphs read as follows:

(Wafer-level transfer of graphene and graphene/hBN heterostructures)

[...] We used Terahertz time-domain-spectroscopy (THz-TDS) to further characterize the electrical properties and homogeneity of the wafer-level transferred graphene and large-area graphene/hBN heterostructure. The THz-TDS measurements are recorded in transmission mode using a photoconductive near-field detector^{64,65} (see Supplementary Information), which records the amplitude and phase information of a THz signal traveling through the sample. Scanning across the sample surface yields spatially resolved images of the graphene sheet resistance on wafer-scale. For our measurements, we set the spatial resolution to 500 μm by adjusting the distance of the near-field detector to the sample and the pitch between acquisition sites. The spatial resolution is significantly larger than the intrinsic length scale that is probed by the THz-TDS. This transport length is estimated by the distance l_D , which a charge carrier transverses during one cycle of the alternating THz field⁶⁶. For our sample and setup, we estimate $l_D \approx 60\text{--}100$ nm (See Supplementary Information for calculation). [...]

(Supplementary Information)

In this work, we did not operate the THz TDS near-field setup at its resolution limit of a few micrometers^{64,65}. Instead, data acquisition was done with a step-size of a few 100 μm . Also, the distance of the near-field detector to the sample was set to a similar value. Thus, the probing area of each pixel was a few 100 μm^2 , which is clearly larger than the intrinsic length scale that is probed by the THz spectroscopy. The intrinsic length scale is estimated by the distance a carrier moves during one cycle of the alternating THz field $l_D \sim \sqrt{D/\omega}$ where D is the diffusion coefficient and $\omega = 2\pi f$ the angular frequency of the THz pulse⁶⁶. The THz-spectrum of our setup covers the frequency range from $f = 0.1 - 3$ THz. In the following calculation we use the range $f = 0.5\text{--}1.5$ THz and the results from the vdP measurements to estimate the relevant length scales for the THz spectroscopy. Assuming similar mobility and charge carrier doping as extracted by the vdP measurements

($\mu = 2800 \text{ cm}^2\text{V}^{-1}\text{s}^{-1}$ and $n = 3.7 \cdot 10^{12} \text{ cm}^{-2}$, respectively), we calculated the conductivity with the Drude formula

$$\sigma = en\mu = 1.66 \text{ mS} \quad (1),$$

and a mean free path of

$$l_{\text{mfp}} = \sigma h / (2e^2 \sqrt{\pi n}) \approx 63 \text{ nm} \quad (2)^{49}.$$

Accordingly, the scattering time of carriers is

$$\tau = l_{\text{mfp}} / v_{\text{F}} \approx 63 \text{ fs} \quad (3),$$

where $v_{\text{F}} \approx 10^6 \text{ m/s}$ is the Fermi velocity⁶⁶. Since the scattering time is smaller than the duration of the THz pulse ($\sim 1 \text{ ps}$), the movement of charge carriers is diffusive with a diffusion coefficient of

$$D = v_{\text{F}} l_{\text{mfp}} / 2 \approx 314 \text{ cm}^2/\text{s} \quad (4)^{67}.$$

Hence, THz spectroscopy at 0.5–1.5 THz is sensitive to the average of the microscopic conductivity over the length scale

$$l_{\text{D}} = \sqrt{D/\omega} \approx 60\text{--}100 \text{ nm} \quad (5).$$

Since both l_{D} and the l_{mfp} are smaller than the grain size, which is in the micrometer range, most carriers do not scatter on grain boundaries. Hence, our results from THz spectroscopy are a reasonable estimation of the average of the sheet conductivity in the area that is probed by the near-field detector. [...]

Reviewer comment #10

10. Line 185: Note, the THz near-field measurement in Figure 2c was performed on a different sample than the vdP measurements in Figure 2a and 2b. I presume this is because the metal contacts would saturate the THz-TDS signal? If so, that could be mentioned. Otherwise, with 10 μm resolution you should be able to get many pixels inside each vdP device for direct comparison.

Our response:

We thank the reviewer for this question. It is correct, that the metal electrodes would affect the THz signal and that the device size would allow to measure in between the contacts. However, for this sample, the doping of the Si substrate is the reason why we cannot perform this measurement. High doping of the Si wafer reduces the transparency of the substrate to THz radiation. The samples for THz TDS were prepared on high resistive silicon wafers with a resistivity of $> 10000 \Omega \cdot \text{cm}$. The Si substrate of the vdP devices has a resistivity of $5 - 10 \Omega \cdot \text{cm}$, which prevents the extraction of quantitative results by THz TDS.

In the revised manuscript, we clarified our choice of the used Si substrates in the main text and in the methods section. The related paragraphs read as follows:

(Wafer-level transfer of graphene and graphene/hBN heterostructures)

[...] Note that the THz near-field measurement in Figure 2c was performed on a different sample with high-resistivity Si than the vdP measurements in Figures 2a and 2b to avoid absorption of the THz signal by the substrate. [...]

(Methods)

THz near-field spectroscopy

We used terahertz (THz) time-domain spectroscopy (TDS) in transmission mode to measure the sheet resistance of the transferred graphene on the target wafer. The samples were prepared on high-resistivity wafers with a resistivity of $> 10^4 \Omega\cdot\text{cm}$, to avoid inadvertent absorption of the THz signal by the target wafer. For the measurements, we used a THz pump/probe setup equipped with a femtosecond fiber-laser (generating pulses of 100 fs duration and 780 nm center wavelength) to pump a terahertz emitter and to gate a photoconductive near-field micro-probe detector (TeraSpike TD-800-X-HR-WT). By measuring the THz transmission through the sample, we extracted the local sheet resistance of the graphene^{64,65}. Lateral scanning of the sample with the tip of the near-field detector held at close distance yielded a spatially resolved map of the graphene sheet resistance with a pixel size of a few hundred micrometer. See Supplementary Information for details of the setup and evaluation.

Reviewer comment #11

11. Fig 3 a inset: I feel the 2-layer graphene inset gives the impression of Bernal stacking. Suggest changing or adding text for the readers who mainly look at figures.

Our response:

We thank the reviewer for this comment. We agree that our previous figure was ambiguous and that the schematic of the 2-layer graphene could have been interpreted as Bernal stacked. It is correct, that the graphene layer in our 2-layer graphene samples is randomly stacked since the grains on the polycrystalline Cu growth substrates are randomly oriented.

In the revised manuscript, we added a label “randomly stacked” to the schematic of the 2-layer graphene on BCB. The new figure is shown below:

(Repeated transfer of graphene)

Figure 3: Characterization of double-layer (2-layer) graphene formed by two consecutive transfers of monolayer (1-layer) CVD graphene and wafer-level integration of top-gated field-effect graphene device. **a)** Photograph of a high resistive silicon wafer with 100 mm diameter after graphene transfer. Dashed lines indicate regions of single-layer, double-layer, and no graphene. **b)** Correlation map of Raman G and 2D peak positions (ω_G and ω_{2D} , respectively), extracted from measurements in regions of 1-layer graphene (triangles) and 2-layer graphene (circles). The colormap represents the full width at half maximum of the 2D peak (Γ_{2D}). **c)** Spatially resolved map of the graphene sheet resistance (R_{sh}) from non-contact THz near-field spectroscopy (left), clearly showing decreased R_{sh} in the 2-layer graphene region. The histogram (right) represents the data inside the dashed rectangle. **d)** Top-gate voltage dependency of the graphene sheet resistance and the field-effect mobility in an integrated graphene device at room temperature. Inset: Optical microscope picture of a fabricated field-effect graphene device. The dashed line marks the graphene area. Electrodes 1 to 4 are used for electrical characterization by a 4-point probe van-der-Pauw method while the gate voltage tuned the doping of the graphene.

Reviewer comment #12

12. Figure 3 results: Why is the sheet resistance for 1-layer graphene of this wafer so much higher than the Figure 2 results?

Our response:

We thank the reviewer for pointing at this observation. We attribute the discrepancy in sheet resistance to a difference in graphene quality before transfer. The process parameters to form 2-layer graphene and to form the graphene/hBN heterostructure were essentially the same.

For the 2-layer graphene sample, the second graphene sheet that forms the 1-layer region on the target wafer, was considerably older (by mistake) than the other sheets used in this work. We assume that the quality of the graphene on that growth substrate was substantially lower. Unfortunately, we discovered this disparity at a late stage of drafting this manuscript. However, we decided to present the full dataset of Raman and THz spectroscopy measured on this sample, since its purpose is to demonstrate the formation of 2-layer graphene by repeated transfer and not benchmark the graphene quality itself.

In the revised manuscript, we have added a sentence to clarify this disparity. The related paragraph reads as follows:

(Repeated transfer of graphene)

[...] We attribute the difference in the 1-layer graphene R_{sh} between these experiments and the dataset presented in Figure 2 to a dissimilarity in graphene quality before transfer. The graphene sheet in Figure 3a that forms the 1-layer region on the target wafer was from a different batch than the graphene sheet used in Figure 2. [...]

Reviewer comment #13

13. Fig 4: Good to see before and after transfer Raman of MoS₂, but why not the before and after transfer for photoluminescence measurements?

Our response:

We thank the reviewer for pointing out this inconsistency in our experimental design.

In the revised manuscript, we added the results and statistical evaluation of photoluminescence measurements before transfer. The revised figures in the main text and in the Supplementary Information as well as the extended paragraph in the main text are as follows:

(Transfer of MoS₂)

Figure 4: Transfer and characterization of MoS₂. **a)** Averaged Raman spectra of multilayer MoS₂ on the SiO₂/Si growth substrate before transfer (blue) and after transfer to the target wafer (red). The Raman spectra are averages of area scans with 8 × 10 measurements in 1 mm² areas (black rectangle in the photograph of transferred MoS₂) and were normalized to the intensity of the E_{2g}¹ mode. The shaded regions represent the standard deviation of the intensity. The vertical dashed lines indicate the peak positions of the E_{2g}¹ and A_{1g} modes of the as-grown film before transfer. **b)** Averages of 25 photoluminescence (PL) intensity spectra of pristine (blue) and transferred (red) multilayer MoS₂ with A and B excitonic transitions (shaded areas: standard deviation). The inset magnifies the region from 1.945 eV to 1.98 eV and shows the Raman modes of BCB, which are superimposed with the optical response of the MoS₂ layer.

(Transfer of MoS₂)

[...] To further investigate the quality of the transferred MoS₂ film, we compare the average of 25 normalized photoluminescence (PL) intensity spectra of pristine and transferred MoS₂ (Figure 4b). Before transfer, we found two prominent peaks at 1.821 eV and 1.968 eV, which correspond to A and B excitonic transitions, respectively⁵⁹. After transfer, we observed a drastic increase in the A/B intensity ratio since the B-excitation vanishes. A high A/B intensity ratio indicates a high quality of the MoS₂ film⁶⁰. The A-exciton transition energy is blue-shifted by 13 meV to 1.834 eV (see Figure S9 for statistics), which we attribute to the change of the substrate material from SiO₂ (before transfer) to BCB (after transfer)⁵⁷. The additional features of the PL intensity between 1.96 eV and 1.97 eV are Raman modes of the underlying BCB, hence, do not belong to the excitonic transitions in the MoS₂ (see Supplementary Information for the Raman spectrum of BCB).

(Supplementary Information)

7. Analysis of photoluminescence spectroscopy of MoS₂

Figure S9: Histogram of the peak positions of the A-excitonic transition in MoS₂ before and after transfer. After transfer, the mean position is blue shifted by 13 meV.

Reviewer comment #14

14. Hall measurements were performed at 1 mA. Which voltage was required to generate 1 mA?

Our response:

We thank the reviewer for this relevant question. The average voltage to force a current of 1 mA through the 27 vdP devices was 1.28 ± 0.17 V.

In the revised manuscript, we added the average voltage in the methods section. The related paragraph reads as follows:

(Methods):

[...] For device characterization, the wafer was resting on an electromagnet (Wuxue Wen Fang Electric Co. Ltd, model WF-P25/20), which was placed inside a probe station (Cascade Microtech Inc.). Four-point probe measurements of the voltage across the device edge (device current: 100 μ A), and the Hall voltage (device current: 1 mA at 1.28 ± 0.17 V, magnetic flux: 28.8 mT) yielded the sheet resistance, carrier density and mobility of the transferred graphene. Electrical measurements were performed under ambient conditions at room temperature, using a parameter analyzer (Keithley SCS4200).

Reviewer comment #15

15. *Supp Info: For the THz results for graphene/hBN, is it necessary to have a second reference area with no graphene but substrate+BCB+hBN?*

Our response:

We thank the reviewer for this question. It is fundamentally correct, that the extraction of quantitative graphene properties requires a reference measurement of the same substrate without graphene layer. However, the hBN layer is electrically insulating and has a thickness of a few nanometer. Hence, it does not absorb the THz radiation significantly. Therefore, the BCB coated Si wafer is an adequate reference for both graphene and the graphene/hBN heterostructure.

In the revised manuscript, we added a comment on the reference area. The related paragraph reads as follows:

(Supplementary Information)

[...] Note that thin layers of electrically insulating materials, such as nanometer-thick hBN, do not absorb THz radiation significantly and reference measurements with and without hBN layer did not show any difference. Hence, adding these layers to the substrate stack does not require a new reference measurement. [...]

References

1. Griep, M. H. *et al.* Enhanced Quality CVD-Grown Graphene via a Double-Plateau Copper Surface Planarization Methodology. *Cryst. Growth Des.* **17**, 5725–5731 (2017).
2. Nishi, Y., Doering, R. & Doering, R. *Handbook of Semiconductor Manufacturing Technology*. (CRC Press, 2017). doi:10.1201/9781420017663.
3. Sutter, P. W., Flege, J.-I. & Sutter, E. A. Epitaxial graphene on ruthenium. *Nat. Mater.* **7**, 406–411 (2008).
4. Yoshii, S. *et al.* Suppression of Inhomogeneous Segregation in Graphene Growth on Epitaxial Metal Films. *Nano Lett.* **11**, 2628–2633 (2011).
5. Pizzagalli, A. Bonding and Lithography Equipment Market for More than Moore Devices. *Yole Dev.* (2018).
6. Burdeaux, D., Townsend, P., Carr, J. & Garrou, P. Benzocyclobutene (BCB) dielectrics for the fabrication of high density, thin film multichip modules. *J. Electron. Mater.* **19**, 1357–1366 (1990).
7. Mills, M. E., Townsend, P., Castillo, D., Martin, S. & Achen, A. Benzocyclobutene (DVS-BCB) polymer as an interlayer dielectric (ILD) material. *Microelectron. Eng.* **33**, 327–334 (1997).
8. Niklaus, F., Enoksson, P., Kälvesten, E. & Stemme, G. Low-temperature full wafer adhesive bonding. *J. Micromechanics Microengineering* **11**, 100 (2001).
9. Chua, L.-L. *et al.* General observation of n-type field-effect behaviour in organic semiconductors. *Nature* **434**, 194–199 (2005).
10. Chua, L.-L., Ho, P. K. H., Sirringhaus, H. & Friend, R. H. High-stability ultrathin spin-on benzocyclobutene gate dielectric for polymer field-effect transistors. *Appl. Phys. Lett.* **84**, 3400–3402 (2004).

11. Quellmalz, A. *et al.* Wafer-Scale Transfer of Graphene by Adhesive Wafer Bonding. in *2019 IEEE 32nd International Conference on Micro Electro Mechanical Systems (MEMS)* 257–259 (2019). doi:10.1109/MEMSYS.2019.8870682.
12. Quellmalz, A. *et al.* Large-Scale Integration of 2D Material Heterostructures by Adhesive Bonding. in *2020 IEEE 33rd International Conference on Micro Electro Mechanical Systems (MEMS)* 943–945 (IEEE, 2020). doi:10.1109/MEMS46641.2020.9056203.
13. Leong, W. S. *et al.* Paraffin-enabled graphene transfer. *Nat. Commun.* **10**, 1–8 (2019).
14. Rahimi, S. *et al.* Toward 300 mm Wafer-Scalable High-Performance Polycrystalline Chemical Vapor Deposited Graphene Transistors. *ACS Nano* **8**, 10471–10479 (2014).
15. Gao, L. *et al.* Face-to-face transfer of wafer-scale graphene films. *Nature* **505**, 190–194 (2014).
16. Shivayogimath, A. *et al.* Do-It-Yourself Transfer of Large-Area Graphene Using an Office Laminator and Water. *Chem. Mater.* **31**, 2328–2336 (2019).
17. Pandey, H. *et al.* All CVD Boron Nitride Encapsulated Graphene FETs With CMOS Compatible Metal Edge Contacts. *IEEE Trans. Electron Devices* **65**, 4129–4134 (2018).
18. Shautsova, V., Gilbertson, A. M., Black, N. C. G., Maier, S. A. & Cohen, L. F. Hexagonal Boron Nitride assisted transfer and encapsulation of large area CVD graphene. *Sci. Rep.* **6**, 30210 (2016).
19. Banszerus, L. *et al.* Ballistic Transport Exceeding 28 μm in CVD Grown Graphene. *Nano Lett.* **16**, 1387–1391 (2016).
20. Banszerus, L. *et al.* Ultrahigh-mobility graphene devices from chemical vapor deposition on reusable copper. *Sci. Adv.* **1**, e1500222 (2015).
21. Petrone, N. *et al.* Chemical Vapor Deposition-Derived Graphene with Electrical Performance of Exfoliated Graphene. *Nano Lett.* **12**, 2751–2756 (2012).
22. Calado, V. E. *et al.* Ballistic transport in graphene grown by chemical vapor deposition. *Appl. Phys. Lett.* **104**, 023103 (2014).

23. Song, J. C. W., Shytov, A. V. & Levitov, L. S. Electron Interactions and Gap Opening in Graphene Superlattices. *Phys. Rev. Lett.* **111**, 266801 (2013).
24. Dean, C. R. *et al.* Hofstadter's butterfly and the fractal quantum Hall effect in moiré superlattices. *Nature* **497**, 598–602 (2013).
25. Ponomarenko, L. A. *et al.* Cloning of Dirac fermions in graphene superlattices. *Nature* **497**, 594–597 (2013).
26. Cao, Y. *et al.* Correlated insulator behaviour at half-filling in magic-angle graphene superlattices. *Nature* (2018) doi:10.1038/nature26154.
27. Cao, Y. *et al.* Unconventional superconductivity in magic-angle graphene superlattices. *Nature* (2018) doi:10.1038/nature26160.
28. Sharpe, A. L. *et al.* Emergent ferromagnetism near three-quarters filling in twisted bilayer graphene. *Science* **365**, 605–608 (2019).
29. Lupina, G. *et al.* Residual Metallic Contamination of Transferred Chemical Vapor Deposited Graphene. *ACS Nano* **9**, 4776–4785 (2015).
30. Purdie, D. G. *et al.* Cleaning interfaces in layered materials heterostructures. *Nat. Commun.* **9**, 5387 (2018).
31. De Fazio, D. *et al.* High-Mobility, Wet-Transferred Graphene Grown by Chemical Vapor Deposition. *ACS Nano* **13**, 8926–8935 (2019).
32. Li, X. *et al.* Transfer of Large-Area Graphene Films for High-Performance Transparent Conductive Electrodes. *Nano Lett.* **9**, 4359–4363 (2009).
33. Suk, J. W. *et al.* Transfer of CVD-grown monolayer graphene onto arbitrary substrates. *ACS Nano* **5**, 6916–6924 (2011).
34. Lin, Y. C. *et al.* Clean transfer of graphene for isolation and suspension. *ACS Nano* **5**, 2362–2368 (2011).

35. Wood, J. D. *et al.* Annealing free, clean graphene transfer using alternative polymer scaffolds. *Nanotechnology* **26**, 055302 (2015).
36. Romagnoli, M. *et al.* Graphene-based integrated photonics for next-generation datacom and telecom. *Nat. Rev. Mater.* **3**, 392 (2018).
37. Gatensby, R., Hallam, T., Lee, K., McEvoy, N. & Duesberg, G. S. Investigations of vapour-phase deposited transition metal dichalcogenide films for future electronic applications. *Solid-State Electron.* **125**, 39–51 (2016).
38. Yim, C. *et al.* Wide Spectral Photoresponse of Layered Platinum Diselenide-Based Photodiodes. *Nano Lett.* **18**, 1794–1800 (2018).
39. Sangwan, V. K. *et al.* Gate-tunable memristive phenomena mediated by grain boundaries in single-layer MoS₂. *Nat. Nanotechnol.* **10**, 403–406 (2015).
40. Zhang, F. *et al.* Electric-field induced structural transition in vertical MoTe₂- and Mo_{1-x}W_xTe₂-based resistive memories. *Nat. Mater.* **18**, 55–61 (2019).
41. Belete, M. *et al.* Nonvolatile Resistive Switching in Nanocrystalline Molybdenum Disulfide with Ion-Based Plasticity. *ArXiv191106032 Phys.* (2019).
42. He, Y.-M. *et al.* Single quantum emitters in monolayer semiconductors. *Nat. Nanotechnol.* **10**, 497–502 (2015).
43. Mak, K. F. & Shan, J. Photonics and optoelectronics of 2D semiconductor transition metal dichalcogenides. *Nat. Photonics* **10**, 216–226 (2016).
44. Wu, W. *et al.* Locally defined quantum emission from epitaxial few-layer tungsten diselenide. *Appl. Phys. Lett.* **114**, 213102 (2019).
45. Desai, S. B. *et al.* MoS₂ transistors with 1-nanometer gate lengths. *Science* **354**, 99–102 (2016).
46. Cao, W., Kang, J., Sarkar, D., Liu, W. & Banerjee, K. 2D Semiconductor FETs—Projections and Design for Sub-10 nm VLSI. *IEEE Trans. Electron Devices* **62**, 3459–3469 (2015).

47. Schmidt, M. *et al.* Mobility extraction in SOI MOSFETs with sub 1nm body thickness. *Solid-State Electron.* **53**, 1246–1251 (2009).
48. Akinwande, D. *et al.* Graphene and two-dimensional materials for silicon technology. *Nature* **573**, 507–518 (2019).
49. Wang, L. *et al.* One-Dimensional Electrical Contact to a Two-Dimensional Material. *Science* **342**, 614–617 (2013).
50. Uzlu, B. *et al.* Gate-tunable graphene-based Hall sensors on flexible substrates with increased sensitivity. *Sci. Rep.* **9**, 18059 (2019).
51. Enderling, S. *et al.* Suspended Greek cross test structures for measuring the sheet resistance of non-standard cleanroom materials. in *Proceedings of the 2005 International Conference on Microelectronic Test Structures, 2005. ICMTS 2005.* 1–4 (IEEE, 2005). doi:10.1109/ICMTS.2005.1452202.
52. Lee, K., Gatensby, R., McEvoy, N., Hallam, T. & Duesberg, G. S. High-Performance Sensors Based on Molybdenum Disulfide Thin Films. *Adv. Mater.* **25**, 6699–6702 (2013).
53. Li, H. *et al.* From Bulk to Monolayer MoS₂: Evolution of Raman Scattering. *Adv. Funct. Mater.* **22**, 1385–1390 (2012).
54. Lee, C. *et al.* Anomalous Lattice Vibrations of Single- and Few-Layer MoS₂. *ACS Nano* **4**, 2695–2700 (2010).
55. Wang, Y., Cong, C., Qiu, C. & Yu, T. Raman Spectroscopy Study of Lattice Vibration and Crystallographic Orientation of Monolayer MoS₂ under Uniaxial Strain. *Small* **9**, 2857–2861 (2013).
56. Rahaman, M. *et al.* Highly Localized Strain in a MoS₂/Au Heterostructure Revealed by Tip-Enhanced Raman Spectroscopy. *Nano Lett.* **17**, 6027–6033 (2017).
57. Buscema, M., Steele, G. A., van der Zant, H. S. J. & Castellanos-Gomez, A. The effect of the substrate on the Raman and photoluminescence emission of single-layer MoS₂. *Nano Res.* **7**, 561–571 (2014).

58. Gibb, A. L. *et al.* Atomic Resolution Imaging of Grain Boundary Defects in Monolayer Chemical Vapor Deposition-Grown Hexagonal Boron Nitride. *J. Am. Chem. Soc.* **135**, 6758–6761 (2013).
59. Splendiani, A. *et al.* Emerging Photoluminescence in Monolayer MoS₂. *Nano Lett.* **10**, 1271–1275 (2010).
60. McCreary, K. M., Hanbicki, A. T., Sivaram, S. V. & Jonker, B. T. A- and B-exciton photoluminescence intensity ratio as a measure of sample quality for transition metal dichalcogenide monolayers. *APL Mater.* **6**, 111106 (2018).
61. Costa, S. D. *et al.* Resonant Raman spectroscopy of graphene grown on copper substrates. *Solid State Commun.* **152**, 1317–1320 (2012).
62. Shivayogimath, A. *et al.* Do-It-Yourself Transfer of Large-Area Graphene Using an Office Laminator and Water. *Chem. Mater.* **31**, 2328–2336 (2019).
63. Watanabe, K., Taniguchi, T. & Kanda, H. Direct-bandgap properties and evidence for ultraviolet lasing of hexagonal boron nitride single crystal. *Nat. Mater.* **3**, 404–409 (2004).
64. Wächter, M., Nagel, M. & Kurz, H. Tapered photoconductive terahertz field probe tip with subwavelength spatial resolution. *Appl. Phys. Lett.* **95**, 041112 (2009).
65. Nagel, M., Matheisen, C. & Kurz, H. 12 - Novel techniques in terahertz near-field imaging and sensing. in *Handbook of Terahertz Technology for Imaging, Sensing and Communications* (ed. Saeedkia, D.) 374–402 (Woodhead Publishing, 2013). doi:10.1533/9780857096494.2.374.
66. Bøggild, P. *et al.* Mapping the electrical properties of large-area graphene. *2D Mater.* **4**, 042003 (2017).
67. Rengel, R. & Martín, M. J. Diffusion coefficient, correlation function, and power spectral density of velocity fluctuations in monolayer graphene. *J. Appl. Phys.* **114**, 143702 (2013).

REVIEWER COMMENTS

Reviewer #1 (Remarks to the Author):

I genuinely believe the method has good potential to mechanize (and possibly even automate) transfer of graphene and 2D materials from their growth substrate and have a strong impact on graphene technology.

Authors have carefully revised the manuscript and thoroughly responded to all the points raised. The following points, however, have not be fully addressed:

- Roughness: my comment was referring to macroscopic roughness, in particular for materials grown on thin foils. I presume the values provided refer to microscopic roughness, i.e. obtained by AFM or similar over small areas. On a wafer scale, however, such values can be orders of magnitude larger, so it should be specified over which area such roughness refer to.

- Advantage of the method: the results in terms of doping and mobility are rather modest and make it hard to justify the complexity of using a wafer bonder over a simple wet-transfer setup. I appreciate Authors' point that the starting material -and not the process- may be the reason of the performance, reason for which in my previous review I encouraged further experiments to prove this important point, which unfortunately does not seem to have been done. It should also be noted that conventional wet-transfer can be used at any stage of device fabrication, allowing placing of the 2D materials directly on semiconductors or pre-fabricated contacts, whereas the proposed method require a non-removable CBC layer which would require further processing to connect the 2D material(s) with any foundry-made underlying structure (e.g. CMOS).

- Heterostructure fabrication: I agree with Authors that quality of multilayer large area hBN is far from its exfoliated counterpart, so I won't be too focused on such results. It is therefore not possible to benchmark the quality of the heterostructure produced by the method and conclude whether it works better or worse than other methods (e.g. wet transfer). As the study also included MoS₂ films, I would recommend investigating a graphene/TMD interface which could give further insight on the capability of the method, including the formation of bubbles at the interface.

- Graphene mobility: similar to previous comment, given the uncertainty on the quality of the starting material, it is difficult to conclude whether performance is limited by the method (e.g. example by charge trapped at the graphene/BCB interface or in the insulator itself) or by the materials itself.

In summary, I believe Authors have developed an interesting and potentially very useful method, however the real "quality" of the materials it can produce (particularly heterostructures) has still to be fully addressed.

Reviewer #2 (Remarks to the Author):

After a careful study of the submitted revised version of the publication, the following conclusion has to be drawn: The original version contained incorrect statements not supported by experiments and careful metrological investigations. For this reason, the publication should be rejected as it does not meet the scientific and ethical standards of Nature Communications.

Reviewer #3 (Remarks to the Author):

All of my points have been addressed. I Am happy for the manuscript to be published as is.

Point-by-Point Response to the Reviewer Comments

We sincerely thank the reviewers for carefully reviewing our work and the constructive and insightful comments, which have helped us to further improve the paper by including new experimental results and additional discussions. Our detailed responses to the reviewer's comments are provided in the following.

Reviewer #1 (Remarks to the Author):

I genuinely believe the method has good potential to mechanize (and possibly even automate) transfer of graphene and 2D materials from their growth substrate and have a strong impact on graphene technology.

Authors have carefully revised the manuscript and thoroughly responded to all the points raised. The following points, however, have not be fully addressed:

Our response:

We thank the reviewer for recognizing the potential impact of our integration approach on 2D material technology, and for the additional helpful comments. We have addressed the remaining question in the detailed response below:

Reviewer comment #1

- Roughness: my comment was referring to macroscopic roughness, in particular for materials grown on thin foils. I presume the values provided refer to microscopic roughness, i.e. obtained by AFM or similar over small areas. On a wafer scale, however, such values can be orders of magnitude larger, so it should be specified over which area such roughness refer to.

Our response:

We thank the reviewer for this clarification and apologize for our misunderstanding. It is correct, that the values provided in the manuscript refer to the microscopic roughness which are obtained by AFM and white light interferometry.

In the revised manuscript, we clarified that our values are microscopic roughness and state the measurement area. The related paragraphs read as follows:

(Wafer-level transfer of graphene and graphene/hBN heterostructures)

[...] For our materials, the microscopic surface roughness of hBN on Cu foil is about 50% higher than for graphene on Cu foil (Rms: 264 nm and 176 nm, respectively, see Figure S3). [...]

(Supplementary Information)

Figure S3: White-light interferometric measurements of the microscopic surface topography of copper (Cu) substrates after material growth of **a)** graphene (measurement area: 63 μm by 47 μm) and **b)** hBN (measurement area: 125 μm by 94 μm).

(Transfer of MoS_2)

[...] The surface roughness of the transferred MoS_2 (Rms: 1.9 nm, measurement area: 25 μm by 25 μm) remained low compared to the original surface roughness of the MoS_2 on the growth substrate (Rms: 0.8 nm, measurement area: 5 μm by 5 μm) [...]

Reviewer comment #2

- Advantage of the method: the results in terms of doping and mobility are rather modest and make it hard to justify the complexity of using a wafer bonder over a simple wet-transfer setup. I appreciate Authors' point that the starting material -and not the process- may be the reason of the performance, reason for which in my previous review I encouraged further experiments to prove this important point, which unfortunately does not seem to have been done. It should also be noted that conventional wet-transfer can be used at any stage of device fabrication, allowing placing of the 2D materials directly on semiconductors or pre-fabricated contacts, whereas the proposed method require a non-removable CBC layer which would require further processing to connect the 2D material(s) with any foundry-made underlying structure (e.g. CMOS).

Our response:

We thank the reviewer for this comment. We agree that based on our previous experiments it was not possible to clearly separate the effects on the device performance of the material quality, on the one hand, and of the transfer process, on the other hand.

To demonstrate that the material quality dominates the device performance, we fabricated new reference samples with the same device design on (1) a quartz substrate and (2) a BCB coated silicon wafer, using a conventional wet transfer method. The remaining process steps were identical to the fabrication of the field-effect devices made by transferring graphene using wafer bonding with bisbenzocyclobutene (BCB). Since all graphene samples were sourced from the same supplier at the same time, we assume that the graphene quality in all experiments was similar. We also fabricated new substrates with field-effect devices using our bond transfer with BCB, to extend the total number of devices in our analysis, and found a maximum field-effect mobility of $2600 \pm 1300 \text{ cm}^2\text{V}^{-1}\text{s}^{-1}$ ($n=16$). The mobilities are in the same range as for the reference devices on both BCB and quartz substrates ($2800 \pm 900 \text{ cm}^2\text{V}^{-1}\text{s}^{-1}$ and $1700 \pm 700 \text{ cm}^2\text{V}^{-1}\text{s}^{-1}$, respectively), fabricated using conventional wet transfer. These results indicate that (1) the transfer by adhesive wafer bonding yields at least similar graphene quality in the final device as do conventional wet transfer methods and that (2) BCB may have the potential to compete with common dielectrics as device substrate. In addition to these new experiments, we would like to point out that Uzlu et al. reported similar mobilities in 54 graphene devices fabricated on polyimide substrates using a wet transfer process, but otherwise of the same design and using the same type of graphene (see Figure S2c)¹.

In the revised manuscript, we included the new characterization of the graphene field-effect devices on BCB and quartz substrates. The updated figure in the main text with the histogram of the mobility and the new section in the Supplementary Information appear as follows:

(Wafer-level transfer of graphene and graphene/hBN heterostructures)

Figure 3: Characterization of double-layer (2-layer) graphene formed by two consecutive transfers of monolayer (1-layer) CVD graphene and wafer-level integration of top-gated field-effect graphene device. **a)** Photograph of a high resistive silicon wafer with 100 mm diameter after graphene transfer. Dashed lines indicate regions of single-layer, double-layer, and no graphene. **b)** Correlation map of Raman G and 2D peak positions (ω_G and ω_{2D} , respectively), extracted from measurements in regions of 1-layer graphene (triangles) and 2-layer graphene (circles). The colormap represents the full width at half maximum of the 2D peak (Γ_{2D}). **c)** Spatially resolved map of the graphene sheet resistance (R_{sh}) from non-contact THz near-field spectroscopy (left), clearly showing decreased R_{sh} in the 2-layer graphene region. The histogram (right) represents the data inside the dashed rectangle. **d)** Top-gate voltage dependency of the graphene sheet resistance and the field-effect mobility in an integrated graphene device at room temperature (left) and histogram of the maximum field-effect mobility of 16 devices at room temperature (bottom right). Optical microscope picture of a fabricated field-effect graphene device (top right). The dashed line marks the graphene area.

Electrodes 1 to 4 are used for electrical characterization by a 4-point probe van-der-Pauw method while the gate voltage tuned the doping of the graphene. Scale bar: 100 μm .

(Field-effect graphene devices)

We fabricated and evaluated top-gated field-effect devices on 100 mm diameter silicon wafers using four photolithography layers and conventional processing technology (Figure 3d, details see Methods). The extracted sheet resistance and charge carrier mobility as a function of the gate voltage of a typical device (Figure 3d left) show the expected characteristic behavior of gated graphene devices¹ (see Methods for measurement details), where the mobility is zero at the Dirac point due to a zero of the transconductance. At this point, the charge carriers convert from electrons to holes or vice versa. In our measurements, the mobility is not exactly zero because of the limited number of data points. The maximum field-effect mobility of holes from 16 devices averages to $2600 \pm 1300 \text{ cm}^2\text{V}^{-1}\text{s}^{-1}$ with values up to $4520 \text{ cm}^2\text{V}^{-1}\text{s}^{-1}$. The mobilities are in the same range as for identical reference devices with the same type of graphene that were fabricated on top of both BCB and quartz substrates ($2800 \pm 900 \text{ cm}^2\text{V}^{-1}\text{s}^{-1}$ and $1700 \pm 700 \text{ cm}^2\text{V}^{-1}\text{s}^{-1}$, respectively) using conventional wet transfer (see Supplementary Information). These results indicate that (1) the graphene transfer by adhesive wafer bonding using BCB yields similar final graphene quality as conventional wet transfer and that (2) BCB as substrate material yields similar graphene properties as common dielectric substrates such as quartz. The mobilities of the field-effect graphene devices on BCB were comparable to the mobilities of the vdP devices shown in Figure 2b, which were solely fabricated by non-contact processes to avoid potential material damage during device fabrication. This similarity suggests that the quality of the transferred graphene is preserved during device manufacturing with multiple processing steps such as high temperature Al_2O_3 deposition, multiple lithographic layers, metal depositions and etching processes. Consequently, these results suggest that our transfer methodology is compatible with large-scale manufacturing of graphene devices using conventional semiconductor process technologies.

(Supplementary Information)

5. Mobility and sheet resistance of integrated field-effect graphene devices

Figure S5-1: Field-effect-graphene devices on BCB substrate fabricated by wafer bonding: Top-gate voltage dependency of the graphene sheet resistance of 16 graphene devices at room temperature (same data set as for the histogram of the field-effect mobility in Figure 3d). The graphene was transferred by adhesive wafer bonding and is resting on the BCB layer in the final device (see Methods for details of the device fabrication and measurements). The sheet resistance reached a maximum at the bias voltage $V_{dirac,bond}$.

Figure S5-2: Field-effect-graphene devices on BCB substrate fabricated by conventional wet transfer. **a)** Histogram of the maximum field-effect mobility of 17 field-effect graphene devices at room temperature. **b)** Top-gate voltage dependency of the graphene sheet resistance (same data set as for the histogram in a)). The graphene was transferred by a wet transfer technique¹ and is resting on the BCB layer in the final device (see Methods for details of the device fabrication and measurements). The sheet resistance reaches a maximum at the bias voltage $V_{dirac,wet}$.

Figure S5-3: Field-effect-graphene devices on quartz substrate fabricated by conventional wet transfer: Two-probe field-effect mobility of six top-gated devices on a quartz substrate at room temperature ($1700 \pm 700 \text{ cm}^2\text{V}^{-1}\text{s}^{-1}$). The graphene was transferred by a conventional wet transfer technique¹ and is resting on the quartz substrate in the final device (see Methods for details of the device fabrication and measurements)

Furthermore, we agree that, using conventional wet transfer methods, it is possible to place the 2D material directly onto pre-processed semiconductors and contacts on an underlying electronic (e.g. CMOS) wafer. Using our method, we have demonstrated the transfer of graphene directly on top of bottom contacts, which were integrated on the BCB layer before the transfer of graphene (see Supplementary Information “1. Integration of bottom electrodes to transferred graphene”). Such contacts could be used to interface integrated circuits that are embedded on the target wafer. In this case, the necessary process steps to connect the 2D material to foundry-made structures on the target wafer after the material transfer are minimized.

In the revised manuscript, we added a sentence which mentions the possibility of using wet-transfer of 2D materials at different stages of the device fabrication. The related paragraph reads as follows:

(Introduction)

[...] Commonly used wet transfer approaches rely on an intermediate polymeric carrier, typically poly(methyl methacrylate) (PMMA)²⁻⁴ or polycarbonate (PC)^{5,6}, which mechanically supports the 2D material during its removal from the growth substrate and the transfer to the target substrate by scooping from the surface of a liquid. This process may be implemented at various stages of the device fabrication and allows the placement of the 2D material directly on a target substrate of choice, including complementary metal oxide semiconductor (CMOS) electronic wafers. [...]

Finally, the reviewer states that “*the results [...] make it hard to justify the complexity of using a wafer bonder over a simple wet-transfer setup.*”. We agree with this statement in the context of experimental work on individual substrates and of very-low volume device manufacturing. However, in the context of medium and high-volume semiconductor manufacturing, it is important to note that wafer bonding is a mature technology that is widely available in semiconductor foundries, and that is customarily used for manufacturing of components at extremely high volumes of several 100 million devices per year⁷. We believe that the availability of existing toolsets in semiconductor manufacturing facilities and their supply chain is a crucial prerequisite for the price- and risk-sensitive semiconductor industry, to incorporate any integration approach for 2D materials. Our transfer method relies only on such established technology and tools, which we see as an extremely strong, even as a decisive advantage of our approach over other methods.

Reviewer comment #3

- Heterostructure fabrication: I agree with Authors that quality of multilayer large area hBN is far from its exfoliated counterpart, so I won't be too focused on such results. It is therefore not possible to benchmark the quality of the heterostructure produced by the method and conclude whether it works better or worse than other methods (e.g. wet transfer). As the study also included MoS₂ films, I would recommend investigating a graphene/TMD interface which could give further insight on the capability of the method, including the formation of bubbles at the interface.

Our response:

We agree with the reviewer on the interpretation of our results, which include a graphene/hBN heterostructure. We also thank the reviewer for the constructive suggestion to investigate a graphene/TMD heterostructure to further demonstrate the capabilities of our method. We have taken this advice and successfully stacked MoS₂ (grown by CVD on SiO₂/Si chips) and graphene (grown on copper foil) by consecutive transfers to MoS₂/graphene and graphene/MoS₂ heterostructures.

In the revised manuscript, we added the description and analysis of MoS₂ heterostructures. The related paragraph reads, and figures are presented, as follows:

(Transfer of MoS₂ and MoS₂/graphene heterostructures)

Figure 4: Transfer and characterization of MoS₂ and MoS₂/graphene heterostructures. **a)** Averaged Raman spectra of multilayer MoS₂ on the SiO₂/Si growth substrate before transfer (blue) and after transfer to the target wafer (red). The Raman spectra are averages of area scans with 8×10 measurements in 1 mm^2 areas (black rectangle in the photograph of transferred MoS₂) and were normalized to the intensity of the E_{12g} mode. The shaded regions represent the standard deviation of the intensity. The vertical dashed lines indicate the peak positions of the E_{12g} and A_{1g} modes of the as-grown film before transfer. **b)** Averages of 25 photoluminescence (PL) intensity spectra of pristine (blue) and transferred (red) multilayer MoS₂ with A and B excitonic transitions (shaded areas: standard deviation). The inset magnifies the region from 1.945 eV to 1.98 eV and shows the Raman modes of BCB, which are superimposed with the optical response of the MoS₂ layer. **c)** Spatially resolved map (left) and histogram (right) of the graphene sheet resistance (R_{sh}) from THz near-field spectroscopy (resolution: $300 \mu\text{m}$). The MoS₂/graphene heterostructure in the center (rectangular region) is surrounded by graphene, both resting on BCB. **d)** Raman and PL spectroscopy of the MoS₂/graphene heterostructure, measured in a $1 \text{ by } 1 \text{ mm}^2$ region on the same sample as in (c) (white rectangle). The Raman peak positions indicate the presence of both graphene (2D peak position ω_{2D} and the respective full width at half maximum $\Gamma_{2D, \text{MoS}_2/\text{gr}}$) and MoS₂ (E_{12g} and A_{1g} peaks). The averaged PL spectrum (bottom right) shows the A and B excitonic transitions of MoS₂ and the superimposed Raman modes of BCB (shaded area: standard deviation).

[...] We formed MoS₂/graphene heterostructures by stacking a centimeter-sized layer of MoS₂ onto a sheet of graphene in two consecutive transfers (see methods). The map of the graphene sheet resistance extracted from THz spectroscopy indicated the presence of graphene in both the MoS₂/graphene heterostructure and its surroundings ($1030 \pm 520 \text{ } \Omega/\text{sq.}$ and $1910 \pm 280 \text{ } \Omega/\text{sq.}$, respectively). In the region of the MoS₂/graphene heterostructure, R_{sh} is significantly reduced. Since the high sheet resistance of a sole MoS₂ layer is not directly detectable in this measurement, we speculate that interaction between graphene and MoS₂ caused this variation. Raman and PL spectroscopy of the MoS₂/graphene heterostructure confirmed the presence and integrity of both materials (Figure 4d and Supplementary Information). The position of the 2D peak of graphene ($\omega_{2\text{D}}$) averaged to $2681.9 \pm 1.8 \text{ cm}^{-1}$ with a full width at half maximum ($\Gamma_{2\text{D},\text{MoS}_2/\text{gr}}$) of $30.3 \pm 3.9 \text{ cm}^{-1}$. This value is in good agreement with the results in Figure 3, which hints at the integrity of the graphene even after stacking. The Raman E_{2g}¹ mode of MoS₂ exhibited a minor shift from $384.26 \pm 0.17 \text{ cm}^{-1}$ to $385.16 \pm 0.35 \text{ cm}^{-1}$, while the position of the A_{1g} mode remained unchanged ($408.06 \pm 0.28 \text{ cm}^{-1}$) (See Supplementary Information on material characterization prior to transfer). The averaged PL spectrum featured the distinct peaks of A and B excitonic transitions in MoS₂ with high A/B ratio and superimposed Raman modes of BCB. We further demonstrated the versatility of the methodology by reversing the order of stacked materials, which formed graphene/MoS₂ heterostructures by successfully transferring graphene onto previously transferred MoS₂ layers (see Supplementary Information). Taken together, the Raman and PL measurements confirm that the layer quality of MoS₂ is preserved during the transfer and the stacking to heterostructures. Hence, our methodology is potentially suitable for large-area transfer of MoS₂ films and their heterostructures.

(Methods)

Formation of MoS₂/graphene heterostructures

First, the silicon target wafer (diameter: 100 mm, resistivity: $> 10^4 \text{ } \Omega\text{-cm}$) was spin-coated with a 2.5 μm thick layer of BCB (CYCLOTENE 3022-46 Dow Inc., spinning speed: 5000 rpm, soft-bake: 100 °C for 4 min). A quarter of a 100 millimeter sheet of monolayer CVD graphene on copper foil (Graphenea Inc.) was bonded to the target wafer with the graphene facing the BCB layer (bond temperature: 190 °C, bond time: 30 min, bond force: 600 N with a resulting bond pressure of 3.1 bar, nitrogen atmosphere). Etching of the copper foil and rinsing in deionized water for 30 min uncovered the transferred graphene on top of the BCB on the target wafer. Next, a multilayer of MoS₂ on a SiO₂/Si chip (10 mm by 10 mm) was placed on top of the transferred graphene (MoS₂ facing the graphene) and surrounded by silicon dummy chips to ensure a uniform force distribution over the wafer during the following bonding process. Bonding at 190 °C for 30 min attached the MoS₂/SiO₂/Si chip to the target wafer (bond force: 750 N with a resulting bond pressure of 3 bar, vacuum atmosphere). Etching in O₂/SF₆ plasma cleaned the edges of the chip while a resist mask protected the remaining surface of the target wafer. Submersion of the bonded stack in acetone stripped the resist mask and detached the growth substrate, leaving the MoS₂ film transferred on top of the graphene, forming a MoS₂/graphene heterostructure.

(Supplementary Information)

6. Raman and photoluminescence spectroscopy of MoS₂/graphene heterostructures

Figure S8-1: Stitched white-light microscopy image with a MoS₂/graphene heterostructure inside the dashed rectangle. The solid rectangle represents the acquisition site for Raman and photoluminescence (PL) spectroscopy. Particles on the growth substrate of the MoS₂ film caused partially bonded regions.

Figure S8-2: Averaged Raman spectrum of graphene in the MoS₂/graphene heterostructure (black) with standard deviation (blue). The 225 individual spectra were acquired in an area of 1 mm² (same dataset as for the histograms in Figure 4d).

Figure S8-3: Histogram of the peak positions of the A-excitonic transition in MoS₂ in the MoS₂/graphene heterostructure after transfer. The 100 individual spectra were acquired in an area of 1 mm² (same dataset as for the histograms in Figure 4d).

Figure S8-4: Averaged Raman spectrum of MoS₂ in the MoS₂/graphene heterostructure. The 100 individual spectra were acquired in an area of 1 mm² (same dataset as for the histograms in Figure 4d) (shaded area: standard deviation).

Figure S8-5: Raman spectroscopy of MoS₂ on the growth substrate (SiO₂/Si chip) before forming the MoS₂/graphene heterostructure. **a)** Averaged spectrum (shaded area: standard deviation). **b)** Histograms of the E¹_{2g} mode position. **c)** Histograms of the A_{1g} mode position.

Figure S8-6: Photoluminescence spectroscopy of MoS₂ on the growth substrate (SiO₂/Si chip) before forming the MoS₂/graphene heterostructure. **a)** Averaged PL spectrum (shaded area: standard deviation). **b)** Histogram of the peak positions of the A-excitonic transition in MoS₂.

9. Formation and analysis of graphene/MoS₂ heterostructures

First, the silicon target wafer (diameter: 100 mm, resistivity: $> 10^4 \Omega\text{-cm}$) was spin-coated with a 2.5 μm thick layer of BCB (CYCLOTENE 3022-46 Dow Inc., spinning speed: 5000 rpm, soft-bake: 100 °C for 4 min). A MoS₂ film on a SiO₂/Si chip (10 mm by 10 mm, 2D Semiconductors Inc.) was placed on top of the adhesive layer (MoS₂ facing the BCB) and surrounded by silicon dummy chips to ensure a wider distribution of the forces over the wafer during the following bonding process. Bonding at 190 °C for 30 min attached the MoS₂/SiO₂/Si chip to the target wafer (bond force: 750 N with a resulting bond pressure of 3 bar, nitrogen atmosphere). Etching in O₂/SF₆ plasma cleaned the edges

of the chip while a resist mask protected the remaining surface of the target wafer (Spin-coating: MICROPOSIT SPR700-1.2 positive tone photoresist (Dow Inc.) at 4000 rpm, soft-bake: 100 °C for 1 min, exposure: proximity lithography, development: MICROPOSIT MF CD-26 (Dow Inc.)). Submersion of the bonded stack in acetone stripped the resist mask and subsequent submersion in KOH solution (2M) detached the growth substrate (SiO₂/Si chip) from the MoS₂ layer by permeation of KOH into the MoS₂/substrate interface. The MoS₂ layer remained transferred on top of the BCB on the target wafer. Next, a sheet of monolayer CVD graphene on copper foil (2 cm by 1 cm, Graphenea Inc.) was bonded to the target wafer with the graphene facing the MoS₂ layer (bond temperature: 190 °C, bond time: 30 min, bond force: 750 N resulting in a bond pressure of 2.9 bar, vacuum atmosphere). Etching of the copper foil and rinsing in deionized water for 30 min uncovered the transferred graphene, forming a graphene/MoS₂ heterostructure on top of the BCB on the target wafer.

Figure S9-1: Averaged Raman spectrum of the graphene/MoS₂ heterostructure, showing both the characteristic peaks of MoS₂ (E_{2g}^1 and A_{1g}) and graphene (2D) (shaded area: standard deviation).

Figure S9-2: Histograms of **a)** the E_{2g}^1 and **b)** the A_{1g} mode position of MoS₂ in the graphene/MoS₂ heterostructure.

Figure S9-3: Histogram of the peak positions of the A-excitonic transition in MoS₂ in the graphene/MoS₂ heterostructure.

Figure S9-4: Raman spectroscopy of MoS₂ on the growth substrate (SiO₂/Si chip) before forming the graphene/MoS₂ heterostructure. **a)** Averaged Raman spectrum (shaded area: standard deviation). **b)** Histogram of the E'_{2g} mode position. **c)** Histogram of and A_{1g} mode position. The measurement captured two different regions of MoS₂, which resulted from a partly patchy MoS₂ growth. A double gaussian fit accounts for the different peak positions in these regions.

Figure S9-5: Photoluminescence spectroscopy of MoS₂ on the growth substrate (SiO₂/Si chip) before forming the graphene/MoS₂ heterostructure.

Reviewer comment #4

- *Graphene mobility: similar to previous comment, given the uncertainty on the quality of the starting material, it is difficult to conclude whether performance is limited by the method (e.g. example by charge trapped at the graphene/BCB interface or in the insulator itself) or by the materials itself.*

Our response:

We thank the reviewer for this comment. We agree that based on our previous experiments it was not possible to clearly separate the effects on the device performance, of the material quality on one hand, and the transfer process on the other hand. To demonstrate the point that the material quality dominates the device performance, we fabricated new reference samples with the same device design on (1) a quartz substrate and (2) a BCB coated silicon wafer, using a conventional wet transfer method. We found that the transfer by adhesive wafer bonding yields at least similar graphene qualities as conventional wet transfer methods and that BCB may have the potential to compete with common dielectrics as device substrate. For a detailed response and changes in the manuscript, we refer to the reviewer's comment #2 above.

Reviewer comment #5

In summary, I believe Authors have developed an interesting and potentially very useful method, however the real "quality" of the materials it can produce (particularly heterostructures) has still to be fully addressed.

Our response:

We thank the reviewer for seeing the potential in our work and the valuable comments, which helped us to substantially improve the manuscript. We believe that we have addressed all comments in the responses above and that the revised work clearly demonstrates the capability of our method.

Reviewer #2 (Remarks to the Author):

After a careful study of the submitted revised version of the publication, the following conclusion has to be drawn: The original version contained incorrect statements not supported by experiments and careful metrological investigations. For this reason, the publication should be rejected as it does not meet the scientific and ethical standards of Nature Communications.

Our response:

We thank the reviewer for the insightful suggestions in the first revision of our manuscript which helped us to improve this work. However, we firmly disagree with the reviewer's opinion that our work does not meet the scientific and ethical standards of *Nature Communications*.

Reviewer #3 (Remarks to the Author):

All of my points have been addressed. I Am happy for the manuscript to be published as is.

Our response:

We thank the reviewer for the positive assessment and for the insightful suggestions in the first revision which helped us to improve this work.

References

1. Uzlu, B. *et al.* Gate-tunable graphene-based Hall sensors on flexible substrates with increased sensitivity. *Sci. Rep.* **9**, 18059 (2019).
2. Li, X. *et al.* Transfer of Large-Area Graphene Films for High-Performance Transparent Conductive Electrodes. *Nano Lett.* **9**, 4359–4363 (2009).
3. Suk, J. W. *et al.* Transfer of CVD-grown monolayer graphene onto arbitrary substrates. *ACS Nano* **5**, 6916–6924 (2011).
4. De Fazio, D. *et al.* High-Mobility, Wet-Transferred Graphene Grown by Chemical Vapor Deposition. *ACS Nano* **13**, 8926–8935 (2019).
5. Lin, Y. C. *et al.* Clean transfer of graphene for isolation and suspension. *ACS Nano* **5**, 2362–2368 (2011).
6. Wood, J. D. *et al.* Annealing free, clean graphene transfer using alternative polymer scaffolds. *Nanotechnology* **26**, 055302 (2015).
7. Pizzagalli, A. Bonding and Lithography Equipment Market for More than Moore Devices. *Yole Dev.* (2018).

REVIEWERS' COMMENTS

Reviewer #1 (Remarks to the Author):

Authors carefully addressed all the points raised in the review. In particular, I appreciate the additional experiments performed and the honest comparison with wet transfer, which is very helpful for the reader to contextualize the results presented. I would therefore recommend publication of the manuscript in Nature Comm.

Point-by-Point Response to the Reviewer Comments

Reviewer #1 (Remarks to the Author):

Authors carefully addressed all the points raised in the review. In particular, I appreciate the additional experiments performed and the honest comparison with wet transfer, which is very helpful for the reader to contextualize the results presented. I would therefore recommend publication of the manuscript in Nature Comm.

Our response:

We sincerely thank the reviewer for carefully reviewing our work and the constructive and insightful comments, which have helped us to improve the manuscript substantially.